# Introducing SlideforMAP; a probabilistic finite slope approach for modelling shallow landslide probability in forested situations

Feiko Bernard van Zadelhoff[1], Adel Albaba[1], Denis Cohen[2], Chris Phillips[3], Bettina Schaefli[4], Lucas Karel Agnes Dorren[1,5], and Massimiliano Schwarz[1,5]

[1]Bern University of Applied Sciences - HAFL, Länggasse 85, CH-3052 Zollikofen, Switzerland
[2]COSCI Ltd.
[3]Manaaki Whenua - Landcare Research, Lincoln, New Zealand
[4]Institute of Geography (GIUB) & Oeschger Centre for Climate Change Research (OCCR), University of Bern, 3012 Bern, Switzerland
[5]Int. ecorisQ Association, P.O. Box 2348, 1211 Geneva 2, Switzerland
**Correspondence:** Van Zadelhoff F.B. (feiko.vanzadelhoff@bfh.ch)

**Abstract.** Shallow landslides pose a risk to infrastructure and residential areas. Therefore, we developed SlideforMAP, a probabilistic model that allows for a regional assessment of shallow landslide probability while considering the effect of different scenarios of forest cover, forest management and rainfall intensity. SlideforMAP uses a probabilistic approach by distributing hypothetical landslides to uniformly randomized coordinates in a 2D space. The surface areas for these hypothetical landslides are derived from a distribution function calibrated on observed events. For each generated landslide, SlideforMAP calculates a factor of safety using the limit equilibrium approach. Relevant soil parameters are assigned to the generated landslides from log-normal distributions based on mean and standard deviation values representative for the study area. The computation of the degree of soil saturation is implemented using a stationary flow approach and the topographic wetness index. The root reinforcement is computed by root proximity and root strength derived from single tree detection data. The ratio of unstable landslides to the number of generated landslides, per raster cell, is calculated and used as an index for landslide probability. We performed a calibration of SlideforMAP for three test areas in Switzerland with a reliable landslide inventory, by randomly generating 1000 combinations of model parameters and then maximising the Area Under the Curve (AUC) of the Receiver Operation Curve. The test areas are located in mountainous areas ranging from $0.5 - 7.5$ km$^2$ with mean slope gradients from 18 - 28°. The density of inventoried historical landslides varies from $5 - 59$ slides/km$^2$. AUC values between 0.64 and 0.93 with the implementation of single-tree detection indicated a good model performance. A qualitative sensitivity analysis indicated that the most relevant parameters for accurate modeling of shallow landslide probability are the soil thickness, soil cohesion and the precipitation intensity/transmissivity ratio. Furthermore, we show that the inclusion of single tree detection improves overall model performance compared to assumptions of uniform vegetation. In conclusion, our study shows that the approach used in SlideforMAP can reproduce observed shallow landslide occurrence at a catchment scale.

keywords: mountain forest, shallow landslide probability, probabilistic modelling, single tree detection, root reinforcement

## 1 Introduction

Landslides pose serious threats to inhabited areas world-wide. They are the cause of 17% of the fatalities due to natural hazards in the period of 1994–2013 (Kjekstad and Highland, 2009). Average annual monetary losses over the period of 2010–2019 are approximately 25 billion US dollars (Munich RE, 2018). In addition, Swiss Re Institute (2019) notes a significant increase in damages by hydrologically related natural hazards over the past 5 years, including hydrologically-triggered shallow landslides. This has been attributed to increased urbanization in risk-prone areas and to an increase in heavy rainfall events. Furthermore, Swiss Re Institute (2019) notes that the modelling of shallow landslides is underdeveloped compared to the severity of the danger they pose. In mountainous regions, landsliding is a prominent natural hazard. For instance, in the Alpine parts of Switzerland, 74 people have died as a result of landslide events between 1946 and 2015 (Badoux et al., 2016). The annual cost of landslide protective measures alone is approximately 15 million CHF each year (Dorren and Sandri, 2009). No distinction is made between deep-seated and shallow landslides in these numbers. Rain induced shallow landslides are one of the most important and dangerous types of mass movement in mountainous regions (Varnes, 1978). Shallow landslides are defined as translational mass movement with a maximum soil thickness of 2 m and are the main focus in this paper. Fortunately, improvements in hazard assessment have significantly decreased the number of shallow landslide related deaths over the past decades (Badoux et al., 2016). This general trend is also supported by long-term data (Munich RE, 2018). The fatality decrease is related to better organizational measures regarding hazards, such as warning based evacuations and road closures. Biological measures, such as management of protection forests, also play a role in mitigation of natural hazards. The latter role is especially important for (shallow) landslides, rockfall, snow avalanches and debris flows (Corominas et al., 2014).

Modelling of shallow landslide triggering has been an ongoing process. Shallow landslide probability has been modelled mostly using a deterministic approach (Corominas et al., 2014). The deterministic approach is defined by using average values of risk components and resulting in a univariate result (Corominas et al., 2014). An example of a deterministic approach in this sense is the SHALSTAB model of Dietrich and Montgomery (1998). Other contemporary examples are TRIGRS (Baum et al., 2002) and SLIP (Montrasio et al., 2011), the latter showing good results in assessing soil saturation in a spatially heterogeneous way. In a comparative research it was noted that the SHALSTAB approach was not representative for the spatial variability of the parameters at a small scale (Cervi et al., 2010). In recent decades, the development of probabilistic models and statistical methods has improved model performance for quantifying landslide probability and the interpretation of their results (Corominas et al., 2014). In statistical methods (e.g. Baeza and Corominas, 2001), there is no explicit accounting of physical processes. Probabilistic methods could take physical processes into account and additionally quantify the reliability of the results considering the probability distribution of values of one or more input parameters (Salvatici et al., 2018). The output is a probability rather than a univariate result. A prime example of a probabilistic model in SINMAP (Pack et al., 1998). Generally, these models perform better than deterministic ones (Park et al., 2013; Zhang et al., 2018), likely due to

natural landslides having a mode of movement significantly controlled by internal inhomogeneities and discontinuities in the soil (Varnes, 1978). These control mechanisms are unpredictable at small-scales, making it hard for deterministic models to identify exact locations of instabilities and adjust the heterogeneous parametrization accordingly. Below we go into more detail on the initiation of shallow landslides.

Initiation of instability is a process that combines mechanical and hydrological processes on different spatial and temporal scales and can thereby be very localized, with successive movement increasing the magnitude of the event (Varnes, 1978). In alpine environments, instabilities are typically triggered by rainfall, leading to soil wetting and ensuing increase of pore pressure, which destabilizes the soil and can then initiate soil movement. An increase in pore pressure can build up in minutes to months following a rainfall event (Bordoni et al., 2015; Lehmann et al., 2013), where rapid pore pressure changes are attributed to macropore flow and slow pore pressure changes to the matrix water flow. The higher the horizontal hydraulic conductivity of the soil, the faster pore pressure changes can develop (Iverson, 2000). The reaction of pore pressure to rainfall is variable and highly dependent on soil type. A key experimental study is the work of Bordoni et al. (2015) in which in-situ measurements were taken on a slope with clayey–sandy silt and clayey–silty sand soils that experienced a shallow landslide. It showed that intense rainfall and a rapid increase of pore pressure were the triggering factors of the landslide. Over the duration of the measurements, comparable saturation degrees have been reached both during prolonged and intense rainfall events. Prolonged rainfall did not result in the pore pressure required to trigger a shallow landslide. Similar behaviour has been observed in an artificially triggered landslide in Switzerland (Askarinejad et al., 2012; Lehmann et al., 2013; Askarinejad et al., 2018). In the first wetting phase (year 2008), homogeneously induced rainfall with a duration of 3 days, an accumulated rainfall of 1700 mm and an intensity of 35 mm/hr, induced a maximum pore water pressure of 2 kPa at 1.2 m soil depth, resulting in no landslide. In the second phase of the experiment (year 2009), the rainfall was heterogeneous, with a maximum intensity of 50 mm/hr in the upper part of the slope that induced an increase of pore water pressure up to 5 kPa at 1.2 m soil depth, resulting in the triggering of a shallow landslide. The triggering was reached after 15 hours with a cumulative rainfall of 150 mm. In addition, a computational study by Li et al. (2013) showed that at a high rainfall intensity (80 mm/hr), the pore water pressure at a depth of 1 m reached a constant value within 1 hour. For a lower intensity of 20 mm/hr, this took approximately 3 hours. This shows that landslide triggering is related to a fast build up of pore water pressure proportional to rainfall intensity. The work of Wiekenkamp et al. (2016) suggests that preferential flow dominates the runoff in a heterogeneous catchment during extreme precipitation events. Water can move downslope very rapidly through macropores (in experimental conditions) under both saturated and unsaturated conditions (Mosley, 1982). The role of macropores can be important in a closed soil structure or in the presence of a shallow impermeable bedrock, where they control the soil hydrological behavior. Further examples of the influence of macropores on hillslope hydrology in various soil types are presented in the work of Weiler and Naef (2003) and Bodner et al. (2014). Additionally, Torres et al. (1998) demonstrates the strong role of macropore in preferential flow paths for landslide triggering in an artificial rain experiment in a loamy sandy soil. Montgomery et al. (2002) and Montgomery and Dietrich (2004) also underline the importance of macropore flow, but state that the vertical flow governs response time and build up of pore pressure rather than the lateral flow in their study areas.

The mechanical aspect of shallow landslide initiation usually results from local instabilities that could extend indefinitely in a infinite constant slope if the shear resistance is low (Varnes, 1978). In complex topography, however, the passive earth pressure at the bottom of the triggering zone reacts with a resisting force, contributing thereby to landslide stabilisation (Schwarz et al., 2015; Cislaghi et al., 2018). It is important to note here that the passive earth pressure is activated in a later phase of the triggering of a shallow landslide and should not be added to active earth pressure or tensile forces acting along the upper half of the shallow landslide (Cohen and Schwarz, 2017).

Besides hydrology, slope and soil characteristics, vegetation plays a key role in landslide triggering (Salvatici et al., 2018; Corominas et al., 2014; Greenway, 1987; González-Ollauri and Mickovski, 2014). The role of vegetation can be subdivided in hydrological and mechanical effects. Vegetation influences the effective soil moisture by interception, increased evapotranspiration and increased infiltration (Greenway, 1987; Masi et al., 2021). Over the short timescale with intense rainfall these hydrological effects are negligible, but do play an important role in pre-event disposition of slope instability (Feng et al., 2020). Among the mechanical effects, root reinforcement, mobilized during soil movement, is an essential component (Greenway, 1987; Schwarz et al., 2010). It is a leading factor in the failure criterion for many vegetated slopes (Dazio et al., 2018). In modelling studies, the influence of root reinforcement on slope stability is often quantified as an apparent added cohesion (Wu et al., 1978; Borga et al., 2002). This apparent cohesion in turn can be added in the limit equilibrium computation of a Safety Factor (SF). Using a Monte Carlo approach of this method (Zhu et al., 2017), it was found that the SF can gain up to 37% stability when including vegetation root reinforcement. In another study in New Zealand, trees showed an effect on soil stability up to 11 meter away from their position and had the ability to prevent 70% of instability events (Hawley and Dymond, 1988). Computational research furthermore shows that root reinforcement by the larger roots is dominant over the smaller roots, even though they are far less numerous (Vergani et al., 2014). The planting pattern and management of the vegetation can have a profound effect on root reinforcement and thus on slope stability (Sidle, 1992). Therefore a detailed approach to calculate the spatial distribution of root reinforcement is important for slope stability calculations. Root reinforcement can be subdivided into two major components: Basal root reinforcement and lateral root reinforcement. Basal root reinforcement is the anchoring of tree roots through the sliding plane into the deeper soil. Lateral root reinforcement is the reinforcement from roots on the edges of the potential slide that stick into the soil outside of the potential slide (Schwarz et al., 2010). In contrast, the mechanical influence of vegetation weight on slope stability is often considered negligible (Reinhold et al., 2009). In current shallow landslide probability modelling, whether deterministic or probabilistic, root reinforcement is generally modelled in a simplified way, for example by including homogeneous root reinforcement (Montgomery et al., 2000). These methods limit the evaluation of the effects of different forest spatial properties such as forest structure, and the contribution of different root reinforcement mechanisms to slope stabilisation (Schwarz et al., 2012). In order to overcome this limitation, we develop a shallow landslide probability model, named SlideforMAP. To ensure a wide applicability, SlideforMAP is designed for a regional scale. In concrete terms this means SlideforMAP should be applied to study areas of 1 - 1000 km$^2$. The main objectives of this work are to:

- Present the SlideforMAP model as a tool for shallow landslide probability assessment.

- Show a calibration of SlideforMAP through a performance indicator over three study areas with 78 field recorded shallow landslide events in Switzerland

- Analyze the expected improvement in the performance of SlideforMAP with a detailed inclusion of vegetation

125 - Provide a qualitative sensitivity analysis and identify the parameters that are of greatest influence on the slope stability

Strong emphasis within the SlideforMAP framework and this paper is put on the quantification of root reinforcement on a regional scale. We will show the effect of accurate, quantitative, representation of root reinforcement has on slope stability over three study areas. Simplifications, lack of a temporal component and calibration constraints make it impossible to use SlideforMAP as an exact forecast tool. The main application for SlideforMAP is as a tool to quantify the effects of vegetation
130 planting, growth and/or management for land managers in relation to shallow landslides.

## 2 Methods: SlideforMAP

### 2.1 Probabilistic modelling concept

SlideforMAP is a probabilistic model that generates a 2D raster of shallow landslide probability ($P_{\mathrm{ls}}$). It is an extension of the approach of Schwarz et al. (2010) and Schwarz et al. (2015). It generates a large number of hypothetical landslides (HLs,
135 singular: HL) within the limits of a pre-defined region of interest. These HLs are assumed to have an elliptic shape and are characterized by a mix of deterministic and probabilistic parameters, from which the landslide stability is computed following the limit equilibrium approach (section 2.2). The probabilistic parameters are the HL location, its surface area and its soil cohesion, internal friction angle and soil thickness parameters (drawn from appropriate random distributions). The location and surface area are approached in a probabilistic way to compute a spatial probability distribution. The soil parameters
140 are probabilistic because we assume their variation is high and important in mountainous environments. The deterministic parameters include several vegetation parameters and hydrological soil parameters. A key originality of the approach stems from the fact that the vegetation parameters can be derived from single-tree scale information (section 2.5). The number of generated landslides is high enough such that each point in a region of interest is overlain by multiple HLs from which a relative $P_{\mathrm{ls}}$ can be estimated by considering the ratio of unstable HLs. A general flow chart of SlideforMAP is given in Fig. 1. More
145 details on the modules follow in the subsequent sections.

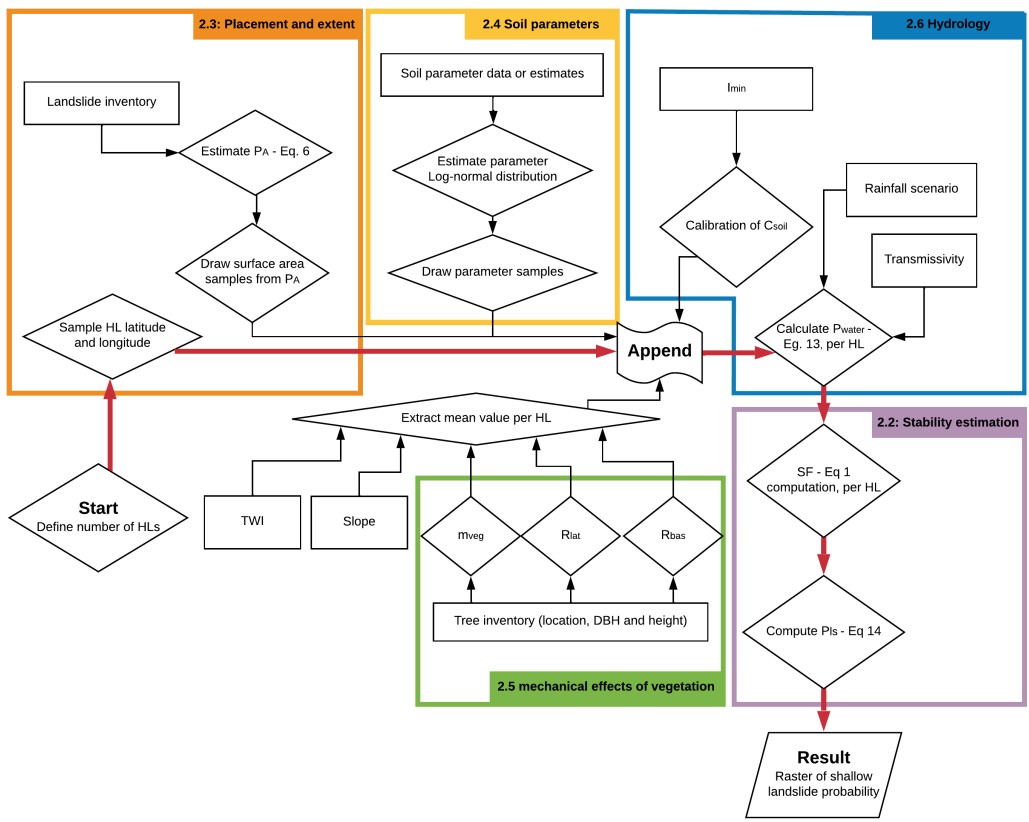

**Figure 1.** Flowchart of the computational steps in SlideforMAP. Separate sections are outlined in colors. The central workflow is highlighted in red.

## 2.2 Stability estimation

The estimate of the stability of each HL is calculated following the limit equilibrium approach (described well in the work of Day (1997)). In this method, a landslide is assumed to be stable if its safety factor (SF) is greater than 1.0. The SF is computed as the ratio of the parallel to slope stabilizing forces and the destabilizing ones:

$$SF = \frac{F_{\text{res}}}{F_{\text{par}}}, \tag{1}$$

150

where $F_{\text{par}}$ [N] is the force parallel to the slope, $F_{\text{res}}$ [N] is the maximum mobilized resistance force. The assumed forces that act upon a hypothetical landslide are schematically shown in Fig. 2.

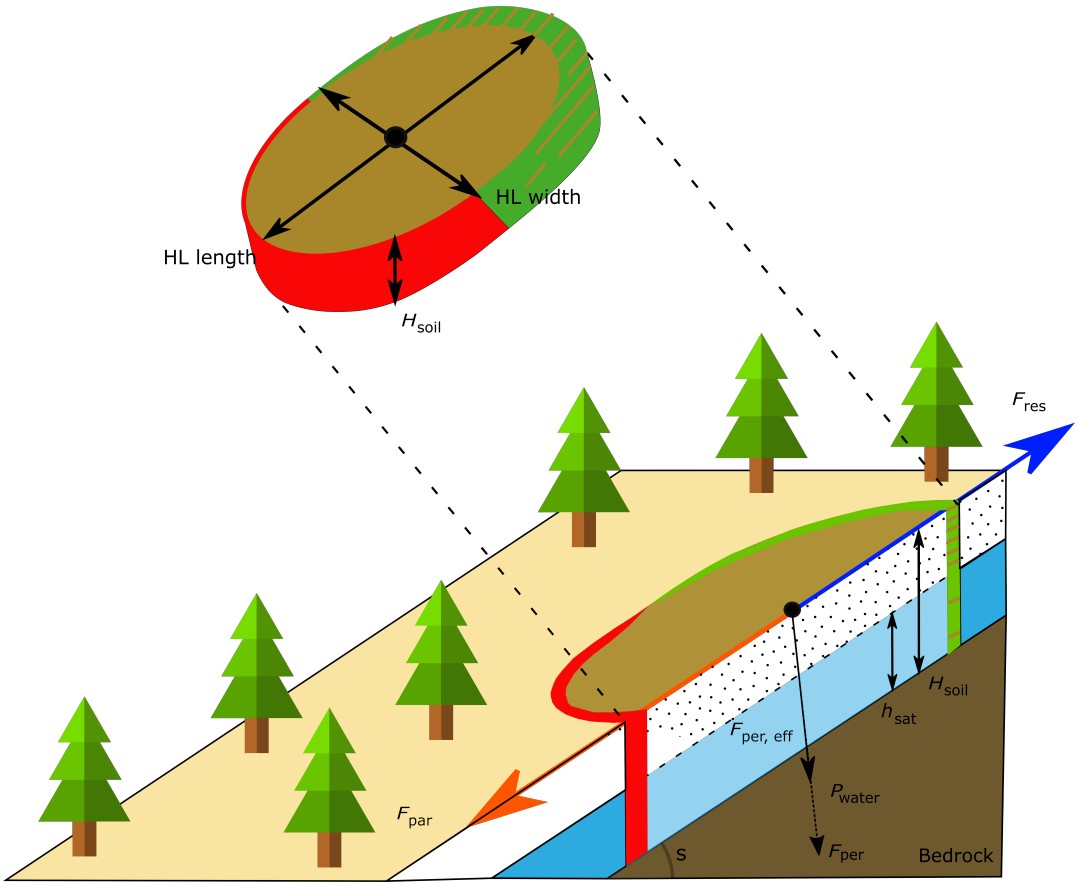

**Figure 2.** Schematic overview of the forces acting upon a hypothetical landslide, as assumed in SlideforMAP. Blue indicates the stabilizing forces and orange indicates the destabilizing forces. Lateral root reinforcement only acts upon the green part of the hypothetical landslide, where tension takes place. In the red is the compression zone in the shallow landslide. Basal root reinforcement and soil shear strength act on the whole potential failure surface.

As seen in Fig. 2, all landslides are assumed to be elliptical (Rickli and Graf, 2009) with a ratio between length and width, $l_{wr}$ = 2. The forces assumed in SlideforMAP are typical for the second stage of the activation phase: the displacement at which
155 lateral root reinforcement is maximized under tension along the tension crack and at which passive earth pressure, lateral root compression and lateral soil cohesion are assumed to not be fully mobilized (Cohen and Schwarz, 2017). The magnitude of the stabilisation's effects of the above mentioned mechanisms considerably change depending on the stiffness of the landslide material and the dimension of the landslide. The quantification of those effects are still a challenge for slope stability calculation at large scales. In order to develop a conservative approach, we neglect those effects in the stability calculations of SlideforMAP.
160 The tension crack is assumed to span the entire upper half of the circumference of the HL and has an assumed length in the range of 0.01 - 0.1 m (Schwarz et al., 2015) depending on the root distribution. This behaviour of progressive shallow landslide failure with a tension crack opening up in the upper half of a shallow landslides is described in detail in Cohen et al. (2009) and

Askarinejad et al. (2012). This is different from the assumptions taken in most landslide models involving root reinforcement (e.g. Montgomery et al., 2000; Schmidt et al., 2001), that assume lateral root reinforcement to be activated at the same time along the entire landslide perimeter. Quantification of the forces in the safety factor calculation follows the limit equilibrium assumptions. This method is outlined in equations 2 to 5 below:

$$F_{\text{par}} = g(m_{\text{soil}} + m_{\text{w}} + m_{\text{veg}}) \cdot \sin(s), \tag{2}$$

$$F_{\text{res}} = \frac{c_{\text{ls}}}{2} \cdot R_{\text{lat}} + F_{\text{res,bas}}, \tag{3}$$

$$F_{\text{res,bas}} = A_{\text{ls}} \cdot C_{\text{soil}} + A_{\text{ls}} \cdot R_{\text{bas}} + F_{\text{per,eff}} \cdot \tan(\phi), \tag{4}$$

$$F_{\text{per,eff}} = g \cdot (m_{\text{soil}} + m_{\text{w}} + m_{\text{veg}}) \cdot \cos(s) - P_{\text{water}}, \tag{5}$$

In these equations, $m_{\text{soil}}$ is the soil mass [kg], $m_{\text{w}}$ is the mass of the water [kg], $m_{\text{veg}}$ is the vegetation mass [kg], $g$ is the gravitational acceleration assumed at 9.81 [m/s$^2$], s is the slope [°], $c_{\text{ls}}$ is the circumference of the landslide [m], $R_{\text{lat}}$ is the lateral root reinforcement [N/m], $F_{\text{res,bas}}$ is the basal resisting force, $A_{\text{ls}}$ [m$^2$] is the area of the landslide, $C_{\text{soil}}$ [Pa] is the soil cohesion [Pa], $R_{\text{bas}}$ is the basal root reinforcement [Pa], $F_{\text{per,eff}}$ is the effective perpendicular resisting forces [N], $\phi$ is the angle of internal friction [°] and $P_{\text{water}}$ is the water pressure [Pa].

## 2.3 Placement and extent

The location of the center of mass of the HLs is generated from two uniform distributions covering the latitudinal and longitudinal extent of the study area. HLs on the edge of the study area are taken into account as well, though cut to the extent of the study area in the later spatial processes of SlideforMAP. The total number of HLs is determined by multiplying the landslide density parameter ($\rho_{\text{ls}}$) with the total surface area of the study area. This number is then uniformly sampled with replacements from the latitudinal and longitudinal distribution. The value of $\rho_{\text{ls}}$ should be high enough such that each raster cell of the study domain is covered by several HLs. The HL surface area is sampled from an inverse gamma distribution following the work of Malamud et al. (2004), which showed that the probability distribution of shallow landslide surface areas follows an inverse gamma distribution (Johnson and Kotz, 1970). The parameterization of a three parameter inverse gamma distribution is shown in equation 6 below.

$$P_{\text{A}} = \frac{1}{a \cdot \Gamma(b)} \left( \frac{a}{A_{\text{ls}} - c} \right)^{(b+1)} e^{\left( \frac{-a}{A_{\text{ls}} - c} \right)}, \tag{6}$$

where $A_{ls}$ is the area of the landslide, $P_A$ is the probability of $A$, $\Gamma$ is the gamma function, $a$, $b$ and $c$ are the scale, shape and location parameters. These distributional parameters are estimated using the landslide surface area data of the inventory (section 3). The estimation is based on minimizing the Root Mean Square Error (RMSE) between the histogram counts (size of histogram bins = 10) of the surface areas from the inventory and the distribution of equation 6. Users can follow this approach with an inventory or use a custom parametrization. The maximum HL surface area is set for all case studies based on the maximum surface area observed in the landslide inventory. This maximum is set to 3000 m$^2$, based on the rounded up maximum value of a well-distributed landslide inventory in Switzerland (section 3.3), but users can vary this parameter.

## 2.4 Soil parameters

Steep-sloped mountainous areas are prone to extreme and unpredictable heterogeneity in soil parameters (Cohen et al., 2009). This makes a heterogeneous deterministic parameterization inaccurate, even if based on observations. To overcome this limitation, a probabilistic approach in the parameterization of soil parameters of the model is applied. Values of soil cohesion and internal friction angle of each HL are randomly generated from independent probability distributions. This is an approach similar to the one taken in Griffiths et al. (2009), who use the log-normal distribution for soil cohesion only and Pack et al. (1998) who use a uniform distribution for soil cohesion and friction angle. We choose the log-normal distributions in our parametrization because it has shown to give a good fit (Fig. A1 with a comparison to a normal distribution in the Appendix; Corresponding code in the supplementary material), it ensures generating positive values only and its accuracy has been shown in Griffiths et al. (2009). The distribution is parametrized by the mean and the standard deviation of observed samples. The mean and the standard deviation are based on different information such as field soil classification or a geotechnical analysis. The soil cohesion in our computations is assumed to be representative for saturated, drained and unconsolidated conditions. Soil thickness is parametrized following a different approach to account for the shallow soils found on steep slopes. An initial soil thickness ($h_{soil}$) is derived from a log-normal distribution. This is then multiplied by a correction factor which is a function of slope inclination as shown in equation 7. Soil thickness is defined here perpendicular to the slope as opposed to soil depth, that is measured in the vertical direction.

$$H_{soil} = h_{soil}\big(1 - P_{\mathcal{N}}(S \leq s|\mu_1, \sigma_1)\big), \tag{7}$$

where $H_{soil}$ [m] is the soil thickness and $s$ is the observed slope, extracted for the HL. $P_{\mathcal{N}}(S \leq s|\mu_1, \sigma_1)]$ is the cumulative normal distribution of the slope $S$ with $\mu_1 = a \cdot m_h$ and $\sigma_1 = b \cdot \sigma_h$. $m_h$ and $\sigma_h$ are the mean and standard deviation of the slope angle of shallow landslides from an inventory or a best guess. a and b are estimated by fitting data from a landslide inventory containing slope angle and soil thickness. Other relations than used by SlideforMAP to correct the soil thickness to the slope (e.g. Prancevic et al., 2020) are possible as well.

## 2.5 Mechanical effects of vegetation

Three properties of vegetation are included in the model. These are vegetation weight, lateral root reinforcement and basal root reinforcement. SlideforMAP only incorporates trees and ignores possible effects by shrubs, grasses and other vegetation. This choice is due to the fact that trees are predominant in influencing slope stability (Greenway, 1987). Single tree detection (Korpela et al., 2007; Menk et al., 2017) serves as a basis to estimate these properties. Single tree position and dimensions are derived from a Canopy Height Model (CHM), which is the difference between the Digital Surface Model (DSM) and the Digital Elevation Model (DEM), using a local maxima detection method (LMD) described in the work of Eysn et al. (2015) and Menk et al. (2017). First, the trees are rasterized. The resolution of this raster has to exceed the effective radial dimension of the trees, in order to calculate representative vegetation parameter values at stand scale. The weight of the tree is calculated by using the tree height and the Diameter at Breast Height (DBH), assuming that the trees are cone shaped. The tree mass, $m_{\text{veg}}$, used in equation 2 and 5, is calculated assuming a mean tree density ($\rho_{\text{tree}}$) of 850 kg/m$^3$. Root reinforcement is added in the model using the method proposed by Schwarz et al. (2012), which relates the root reinforcement to the distance to a tree, the size of the tree and the tree species. Two rasters are computed. A raster with the nearest distance to a tree ($D_{\text{trees}}$) and a raster with the average DBH of all trees within an assumed maximum distance of root influence ($D_{\text{trees,max}}$), set at 15 m. We compute actual lateral root reinforcement for a given grid cell as a function of maximum lateral root reinforcement and soil thickness, which reduces maximum lateral root reinforcement. The maximum lateral root reinforcement, $RR_{\text{max}}$ [N/m], is computed as a function of $D_{\text{trees}}$ and DBH (Moos et al., 2016; Gehring et al., 2019) according to equation 8 below:

$$RR_{\text{max}} = (c \cdot DBH) \cdot \Gamma_{PDF}\left(\frac{D_{\text{trees}}}{DBH \cdot 18.5}\middle|\alpha_1, \beta_1\right), \tag{8}$$

In equation 8, $c$ is a fitting parameter in N/m$^2$ based on the work of Schwarz et al. (2010). DBH is in [m]. The $\Gamma_{\text{PDF}}(x|\alpha_1, \beta_1)$ is the gamma probability density function ($\Gamma_{\text{PDF}}$) evaluated as function of $x$ with shape parameter $\alpha_1$ and scale parameter $\beta_1$. Both $\alpha_1$ and $\beta_1$ are dimensionless. The parameters should ideally reflect any knowledge about how root reinforcement decreases with distance for specific tree species. The $\Gamma_{\text{PDF}}$ is written as:

$$\Gamma_{PDF}(x|\alpha, \beta) = \frac{\left(\frac{x-\mu}{\beta}\right)^{\alpha-1} \cdot e^{-\left(\frac{x-\mu}{\beta}\right)}}{\beta\Gamma(\alpha)}, x > \mu; \gamma, \beta > 0, \tag{9}$$

The location parameter $\mu$ is defined as zero in our application. Soil thickness reduces the effects of lateral root reinforcement that contributes to stabilize a shallow landslide. This decrease of lateral root reinforcement with soil thickness is obtained as follows:

$$R_{\text{lat}} = RR_{\text{max}} \cdot \int_0^{H_{\text{soil}}} \Gamma_{\text{PDF}}\left(H\middle|\alpha_2, \beta_2\right) dH, \tag{10}$$

In this equation $\Gamma_{\text{PDF}}(H|\alpha_2, \beta_2)$ is the $\Gamma_{\text{PDF}}$ for the normalized root distribution over the soil thickness with shape parameter $\alpha_2$ and scale parameter $\beta_2$. In this equation $\beta_2$ has the unit [m] in order to make the integral of the $\Gamma_{\text{PDF}}$ dimensionless. Slide-

forMAP computes this integral by numerical approximation. This method computes the root reinforcement where only one tree can influence a cell. A spatially representative minimum root reinforcement value is calculated in a stand assuming a triangular lattice (Giadrossich et al., 2020). Under this assumption, three root systems interact additively. Basal root reinforcement, $R_{\text{bas}}$ is assumed to be proportional to lateral root reinforcement and dependent on soil thickness according to the relation shown in equation 11:

$$R_{\text{bas}} = RR_{\text{max}} \cdot \Gamma_{\text{PDF}}(H_{\text{soil}}|\alpha_2, \beta_2), \tag{11}$$

where $\Gamma_{\text{PDF}}(H_{\text{soil}}|\alpha_2, \beta_2)$ is the normalized root distribution in the vertical direction. The $\Gamma_{\text{PDF}}$ in this application the unit [m$^{-1}$] which leads to a unit of [Pa] for the term $R_{\text{bas}}$, under the assumption of isotropic conditions.

## 2.6   Hydrology

The hydrological module in SlideforMAP is based on the TOPOG model (O'Loughlin, 1986), which includes a specific to-

pographic index as inspired by Kirkby (1975). In this framework we specifically assume macropore flow dominates hillslope hydrology. The identical model is used in the SHALSTAB stability model (Montgomery and Dietrich, 1994) and SINMAP (Pack et al., 1998). It is assumed that the saturated soil fraction of each cell holds a relation to its specific catchment area, its slope angle, a constant precipitation intensity and the soil transmissivity (equation 12). This is in close correspondence to the parameterization used in the widely used TOPMODEL (Beven and Kirkby, 1979). Limitations of this approach is the

assumption of uniform soil transmissivity, no inclusion of initial conditions, steady state flow and lateral flow governing of soil moisture pattern. These limitations and generalizations make the model insufficient in capturing detailed hydrological pattern, especially in mountainous regions modelled by SlideforMAP. Despite this, we assume the approach to be suitable for a general pattern of saturated fraction and subsequent pore pressure. In addition to this shortcoming we ignore the apparent hydrological cohesion (Chae et al., 2017) prominent in unsaturated fine and clayey soils, but of little prominence in other conditions

(Montrasio and Valentino, 2008). The saturated soil fraction, $h_{\text{sat}}^*$ [-], of a soil column is defined in equation 12 below:

$$h_{\text{sat}}^* = \frac{I \cdot a}{T \cdot b \cdot \sin(s)}, \tag{12}$$

$I$ [m/s] is the constant precipitation intensity, $T$ [m$^2$/s] is the transmissivity, $a$ is the contributing catchment area [m$^2$], $s$ is the slope inclination [°], and $b$ is the contour length [m] that in our model corresponds to the cell size (see Section 3.2 for details on its computation). We assume dominant macropore flow, which has the ability to quickly drain a catchment and potentially

reach a state of stationary flow. Using this estimated $h_{\text{sat}}^*$, pore water pressure is computed as:

$$P_{\text{water}} = H_{\text{soil}} \cdot \cos(s) \cdot h_{\text{sat}}^* \cdot g \cdot \rho_{water}, \tag{13}$$

where $P_{water}$ [Pa] is the pore water pressure (used in equation 5), $H_{soil}$ [m] is the soil thickness, $s$ is the slope angle, $g$=9.81 m/s$^2$ is the gravitational acceleration, $\rho_{water}$ is the density of water assumed equal to 998 kg/m$^3$. The same value for water density is used in the computation of the water mass in the HL.

## 2.7 Model initialisation

The model has a total of 3 probabilistic parameters and 15 deterministic parameters (Table 1). The deterministic parameters as well as the distributional parameters for the probabilistic parameters are determined from in-situ data or from literature (Section 3). In a first step of the workflow for the application of SlideforMAP, after assigning the deterministic parameter values and sampling a value for each probabilistic parameter, a minimum value of soil cohesion is computed for each HL to obtain stable conditions (safety factor, SF >= 1.0) under uniform a precipitation intensity of 28.3 mm/day or 1.2 mm/hr. This threshold of precipitation intensity is chosen according to Leonarduzzi et al. (2017), who statistically analyzed over 2000 landslides in Switzerland over the period 1972–2012 and found this as a triggering threshold. The minimum value of soil cohesion is obtained by equating $F_{par}$ (equation 2) and $F_{res}$ (equation 3). If the minimum value of soil cohesion is larger than the sampled soil cohesion, the soil cohesion is updated to the minimum value. This procedure can be altered by users when another threshold or no threshold at all applies.

**Table 1.** An overview of all variable model parameters of SlideforMAP. The second to last column indicates the source of the default value. The last column indicates whether the default is global or specific for this research in Switzerland (CH).

| Parameter | Description | Default value | Unit | Source | Extent |
|---|---|---|---|---|---|
| $m_\text{d}$ | Soil thickness mean | 1 | m | Estimate | Global |
| $\sigma_\text{d}$ | Soil thickness standard deviation | 0.25 | m | Estimate | Global |
| $m_\text{C}$ | Soil cohesion mean | 2 | kPa | Estimate | Global |
| $\sigma_\text{C}$ | Soil cohesion standard deviation | 0.5 | kPa | Estimate | Global |
| $m_\phi$ | Angle of internal friction mean | 30 | ° | Estimate | Global |
| $\sigma_\phi$ | Angle of internal friction standard deviation | 4 | ° | Estimate | Global |
| $\rho_\text{ls}$ | Density of the random generated landslides | 0.1 | HL/m$^2$ | Estimate | Global |
| $\rho_\text{soil}$ | Dry soil density | 1500 | kg/m$^3$ | Estimate | Global |
| $T$ | Soil transmissivity | 0.1 | m$^2$/s | Estimate | Global |
| $I$ | The precipitation event that is tested | 10 | mm/hr | Estimate | Global |
| $I_\text{min}$ | Precipitation intensity threshold for instability | 1.2 | mm/hr | Leonarduzzi et al. (2017) | CH |
| $r_\text{xy}$ | Raster resolution of the SlideforMAP run | 2 | m | Estimate | Global |
| $l_\text{wr}$ | Ratio between length and width of the landslides | 2 | - | Estimate | Global |
| $c$ | Fitting parameter for the lateral root reinforcement | 25068.54 | - | Gehring et al. (2019) | CH |
| $\alpha_1$ | Shape of root distribution in horizontal direction | 0.862 | - | Gehring et al. (2019) | CH |
| $\beta_1$ | Scale of root distribution in horizontal direction | 3.225 | - | Gehring et al. (2019) | CH |
| $\alpha_2$ | Shape of root distribution in vertical direction | 1.284 | - | Gehring et al. (2019) | CH |
| $\beta_2$ | Scale of root distribution in vertical direction | 3.688 | m | Gehring et al. (2019) | CH |
| $D_\text{trees,max}$ | maximum distance for influence of tree roots | 15 | m | Estimate | Global |
| $\rho_\text{tree}$ | Density of a tree | 850 | kg/m$^3$ | Estimate | CH |
| $\rho_\text{water}$ | Density of water | 998 | kg/m$^3$ | Estimate | Global |

## 2.8 Landslide probability computation

After model initialisation, SF (equation 1) is computed for each of the generated HLs. Based on the SF for all generated HLs, landslide probability per raster cell (with the resolution of the original DEM), $P_\text{ls}$, is computed as:

$$P_\text{ls} = \frac{n_\text{us}}{n_\text{HL}}, \tag{14}$$

where $n_\text{us}$ is the number of unstable HLs, i.e. of HLs with SF<1.0 and $n_\text{HL}$ is the total number of generated HLs (the HLs are overlapping). Both per raster cell. Finally, this results in a raster of shallow landslide probability on a resolution of the input DEM.

# 3 Data

## 3.1 Study areas

Three study areas were chosen to test SlideforMAP based on the availability of elevation data and detailed records of historical shallow landslide events (Fig. 3), each varying in size and location to test the robustness and the general applicability of the model.

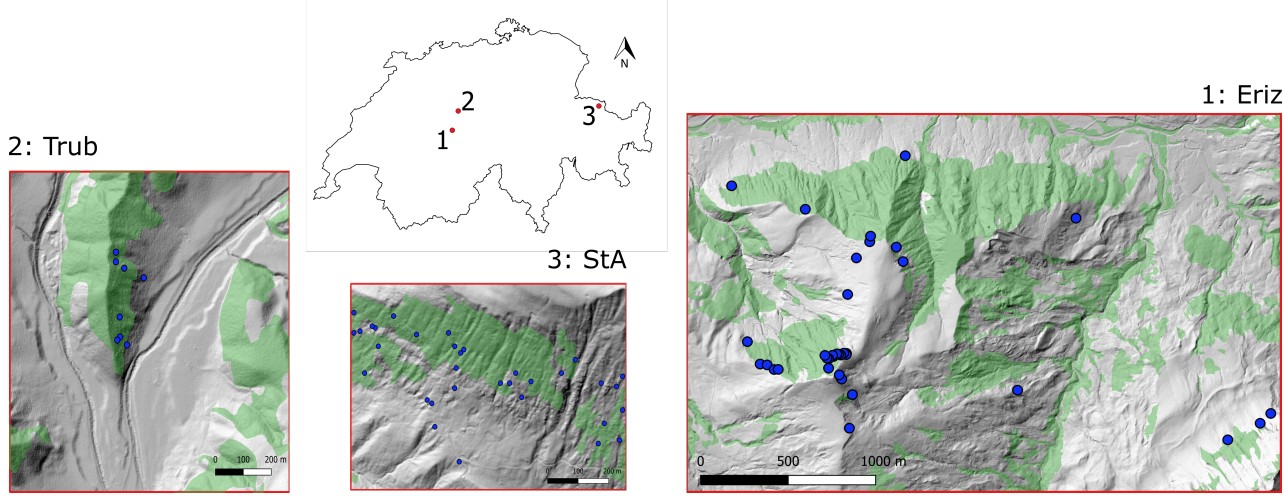

**Figure 3.** Locations of the study areas in Switzerland with observed Shallow landslide occurrence over the period 1997 - 2012 (blue dots); the case study names are given according to nearby villages: Trub, St. Antönien and Eriz. Forest covered area is presented in green (source: Swisstopo, 2020).

The geological formations in the Eriz study area vary from Oligocene freshwater Molasse in the lower northern part, morainic material in the central part and Cretaceous Limestone in the highest parts. Forests are dominated by spruce (*Picea abies*),

except for the lower regions where broad-leaved trees are dominant. In the Trub study area, the dominant geological formation is Miocene Marine Molasse and forests are dominated by spruce. In the St. Antönien (from here forward abbreviated to 'StA') study area, the dominant geological formation is Flysch (Prättigauer Flysch), partially covered by till (Moos et al., 2016). The forest in this study area is also dominated by spruce (Moos et al., 2016). Further characteristics of the study areas are given in Table 2.

**Table 2.** Study area characteristics. Meteorological data is from the HADES yearly average precipitation for the time period 1981 - 2010 (Frei et al., 2020). Shallow landslide number and density from the inventory in section 3.3.

| Name | Centre coordinate | Surface area | Mean prec. | Elevation | Number of slides | Slide density | Mean slope |
|------|-------------------|--------------|------------|-----------|------------------|---------------|------------|
|      | lat;lon (WGS84)   | km$^2$       | mm/year    | m.a.s.l.  |                  | Slides/km$^2$ | °          |
| Eriz | 7.81; 46.78       | 7.54         | 1700       | 960 - 1750 | 37              | 4.9           | 20.4       |
| Trub | 7.90; 46.96       | 1.00         | 1620       | 820 - 1020 | 8               | 8.0           | 18.3       |
| StA  | 9.80; 46.98       | 0.56         | 1310       | 1540 - 2010 | 33             | 58.9          | 27.5       |

## 3.2 Input data

To accurately measure $P_{ls}$ for each study area, the following data are required.

- Digital Surface Model (DSM) and Digital Elevation Model (DEM)

- Average and standard deviation values for soil cohesion, thickness and friction angle

- A representative landslide inventory containing at least:

  - Average landslide soil thickness
  - Landslide surface area

In addition to the DEM, the DSM is applied in the vegetation module of SlideforMAP. The DEM and the DSM are both acquired from the SwissAlti3D database (Swisstopo, 2018), which makes use of aerial laserscanning (ALS). Both the DSM and DEM are available at a resolution of 0.5 m. As an alternative to the use of a landslide inventory and the DSM for single tree identification, users can also use synthesized values for the parameters derived from this data. After pit filling, the DEM is used to compute a slope map following the method of Zevenbergen and Thorne (1987). The topographic wetness index $\theta$ for Fig. 4 is computed on a raster cell basis based on the 2 m DEM using equation 15.

$$\theta = \frac{a}{b \cdot \sin(s)}, \tag{15}$$

where $a$ is the specific upslope catchment area, $b$ is the contour length and $s$ is the slope angle. To avoid numerical problems for elongated catchments, $\theta$ is computed using a 2 km buffer around the catchment. The large buffer size is chosen arbitrarily, but can be reduced by other users. The standard D8 method is applied for the computation of the upslope catchment area from the DEM (O'Callaghan and Mark, 1984). For single tree detection, the FINT algorithm (Menk et al., 2017) is used. Since the results of such detection methods are strongly influenced by the resolution and smoothness of the input data (Eysn et al., 2015), we applied the LMD method to the canopy height model (CHM). This canopy height model is computed by subtracting the DEM from the DSM and is resampled to a resolution of 1, 1.5 and 2 m. In addition, three different Gaussian filters were applied

on the 1 m resolution CHM. These three filters have a radius of 3, 5 and 7 cells and a standard deviation of 2 m. To identify the input data that leads to LMD results with the highest accuracy, we evaluated the identified trees in three randomly selected forest inventory plots with an area of 20 m x 20 m for each study site. In these plots, we visually identified all recognisable tree crowns, on the basis of aerial photos (Swisstopo, 2017) and the CHM. The identified trees were then compared to the

LMD result, using the difference in the number of detected trees. The input data leading to the most accurate results in all three study sites was the 1 m resolution CHM with a Gaussian filter of a 3 cells radius and with the fixed standard deviation of 2 m. This combination has been applied to the entire area of the three study sites. To estimate the DBH from the tree heights of all detected trees, the following empirical equation (Dorren, 2017) was used:

$$DBH_{\text{tree}} = \frac{(H_{\text{tree}})^{1.25}}{100}, \tag{16}$$

where $DBH_{\text{tree}}$ [m] is the diameter at breast height of a given tree and $H_{\text{tree}}$ [m] its height. Details resulting from the LMD method for the three study areas are shown in Table 3.

**Table 3.** Vegetation parameters in the study areas. The forest cover is derived from Swisstopo (2020).

| Study area | Trees identified | Forest cover | Mean stem density | Mean DBH | Std. deviation DBH |
|---|---|---|---|---|---|
| | | % | Stems/ha | m | m |
| Eriz | 38923 | 32 | 165 | 0.51 | 0.27 |
| Trub | 7267 | 26 | 270 | 0.55 | 0.30 |
| StA | 1796 | 27 | 120 | 0.31 | 0.18 |

The lateral and the basal root reinforcement (equations 10 and 11) are parameterized using the values from Gehring et al. (2019) ($\alpha_1 = 0.862$, $\beta_1 = 3.225$, c = 25068.54, $\alpha_2 = 1.284$, $\beta_2 = 3.688$). In their work, the calibration was performed on beech (Fagus Sylvatica) stands over varying elevations. Our study areas, however, are predominantly vegetated by spruce trees. Therefore

a discrepancy in the estimated root reinforcement will likely arise. Unfortunately, this is the only published set of calibrated values.

### 3.3   Landslide inventory

A landslide inventory is required to quantify a distribution for slope, surface area and soil thickness for the HLs. This inventory does not necessarily have to be well distributed in the study area, or even be present in the area. However, it should be

representative of the conditions in the area of interest as much as possible. A dataset of 668 shallow landslides that occurred between 1997 and 2012 in Switzerland has been created by the Swiss Federal Office for the Environment (Rickli et al., 2019). Statistical information on the landslides can be seen in Fig. 4. We assume the properties in this inventory to be representative for shallow landslides in Switzerland. All landslides are triggered by rainfall and the majority of the landslides are shallower

than 1.5 m (Fig. 4). The landslides in the StA and Trub area took place in 2005 during or shortly after heavy rainfall in August. The landslides in the Eriz area from 2012 are related to heavy rainfall in July. Exact precipitation amounts and intensities are unknown. The data is formatted with centre points and surface area of the shallow landslide initiation area. In our analysis we assume they have an elliptical shape.

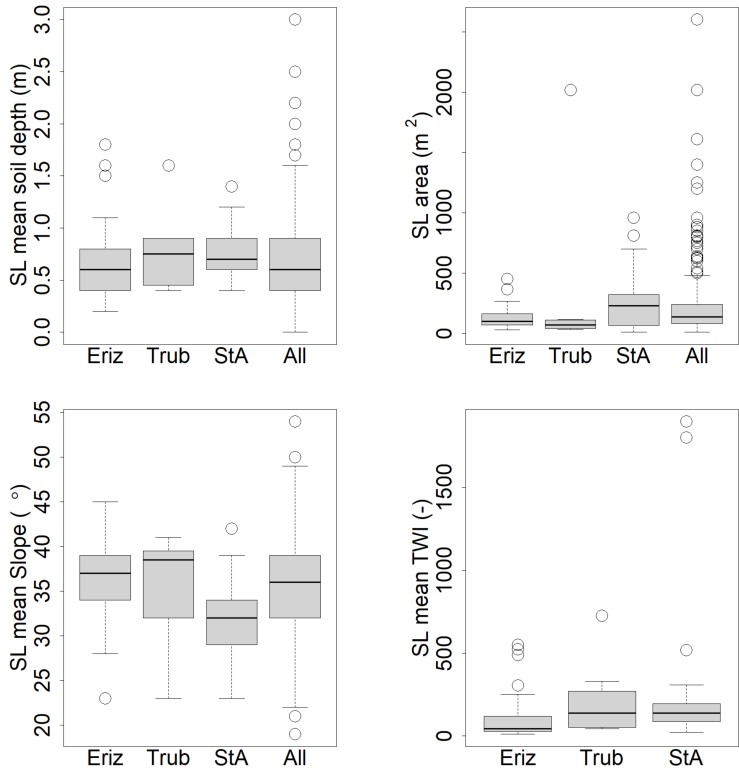

**Figure 4.** Overview of landslide properties for the studied regions. Top row: mean soil thickness (left) and the surface area (right) of the shallow landslide (SL) for the test areas and the total inventory; bottom row: mean slope (left) and mean TWI (right). The box plots show the 25, 50 and 75 percentiles, the whiskers extend to 1.5 times the length between the 25 and 75 percentile. Outliers are marked as circles. The TWI was extracted from the TWI raster cells that lie inside the landslide inventory polygons.

The inventory is used to estimate the parameters for the surface area distribution used in SlideforMAP (equation 6), via minimization of the RMSE between observed frequencies and theoretical frequencies. The estimated values of the slope, scale and location parameters are: a = 1.40, b = $1.5^{-4}$ m$^2$, c = $4.28^{-8}$ m$^2$. In addition, the inventory is used to calibrate the a and b parameters for the soil thickness correction factor as used in equation 7. For the fitting (Appendix, Fig. A2) of the correction factor we use classes of inclination of 2.5 degrees and the soil thickness values corresponding to 95[th] percentile. This best fit for equation 7 was obtained with the values of a = 1.47 and b = 0.50.

### 3.4 Model calibration and sensitivity analysis

The model has a total of 21 parameters that are derived from observed data, from literature or that are set to default values; their values, given in Table 1, are not further varied in the model behavior analysis due to their assumed low variance. The remaining parameters can potentially influence the landslide probability, mostly given their variation as observed in nature. These parameters are: $I_{\text{min}}$, $l_{\text{wr}}$, $c$, $\alpha_1$, $\beta_1$, $\alpha_2$, $\beta_2$, $D_{\text{trees,max}}$, $\rho_{\text{tree}}$, $\rho_{\text{water}}$. The remaining 12 parameters are then calibrated by Monte Carlo simulation, drawing a high number of parameter samples for all calibration parameters and evaluating the corresponding model performance based on the Area Under the Curve (AUC) method (Metz, 1978; Fawcett, 2006). We hereafter first present the used performance evaluation method, followed by the parameter sampling method used for the calibration as well as for the sensitivity analysis. In addition, we present four vegetation parameter scenarios that are developed to test the potential influence of vegetation. Due to the limited size of the landslide inventory, we do not include an independent validation of SlideforMAP.

### 3.4.1 Model performance evaluation

The basis of the application of the AUC method is a spatial representation of the landslide inventory in a boolean raster (0 = no past landslide present, 1 = past landslide present). For each randomly generated parameter set, the simulated $P_{\text{ls}}$ (section 2.8) is also converted to a boolean raster, by selecting a threshold to assign 0 or 1. Overlaying the inventory raster on the modelled raster, results in a confusion matrix with four possible combinations, as shown in Table 4.

**Table 4.** The confusion matrix, resulting from the comparison of a reference boolean raster and a raster corresponding to a simulation.

|  |  | Model | |
|---|---|---|---|
|  |  | **True** | **False** |
| Inventory | **True** | True positive (TP) | False negative (FN) |
|  | **False** | False positive (FP) | True negative (TN) |

A so-called Receiver Operator Curve (ROC) can be obtained by computing the values of the confusion matrix for all unique values in the simulated raster as threshold values and for each plotting the sensivitiy, TP/(TP+FN), against the specificity , TN/(TN+FP). The area under the ROC curve is the AUC and defines the accuracy of the model on a scale of 0.5 - 1.0, where 0.5 is being no better than a random guess and 1.0 is a perfect prediction.

### 3.4.2 Parameter sampling and qualitative sensitivity

The parameter samples for the Monte Carlo-based model calibration and the subsequent sensitivity analysis are generated using the Latin Hypercube Sampling (LHS) technique (McKay et al., 1979). This makes use of semi-random samples of variables over pre-defined ranges. The outcome of a Monte Carlo-based calibration is highly influenced by the ranges chosen for the parameters. For this reason, parameter ranges were chosen as realistically as possible. To estimate the parameter ranges for

soil properties, soil types in USCS classes are taken from the shallow landslide inventory (a total of 377 had their soil type listed). Soil types present more than ten times are taken into account and aggregated into a hybrid table of soil cohesion and angle of internal friction values per soil type based on the values given in the work of Dysli and Rybisar (1992) and VSS-Kommission (1998) (see Appendix, Table A1). In order to obtain a realistic range for the soil cohesion, first the mean soil cohesion (weighted on USCS soil type occurrence) is computed and then the weighted standard deviation is subtracted and added twice to the weighted mean. This is to account for 95% of the variation in the observed soil cohesion (assuming a normal distribution). The same procedure is performed for the angle of internal friction. The range of transmissivity values is obtained by taking the saturated hydraulic conductivity from the work of Freeze and Cherry (1979) for the respective soil classes and by multiplying these saturated hydraulic conductivities with the minimum and maximum soil thickness of the soil class. From the resulting list of possible transmissivity values per soil class, the minimum and maximum are taken for the LHS range. For the precipitation intensity, four depth duration values are defined. These correspond to a duration of 1 hour and 24 hours with subsequent return periods of 10 and 100 years. The duration of 1 to 24 hours is in line with the SlideforMAP assumption of quick macropore-flow dominated lateral groundwater flow. The return periods of 10 and 100 years were chosen arbitrarily in line with forest management timescales. Precipitation intensities are computed using data from the work of Jensen et al. (1997) and the methodology as described in the work of HADES (2020). An overview of the intensity - return period rainfall values is given in Table 5.

**Table 5.** Rainfall intensity [mm/h] for specific duration and return periods, used to define the boundaries in the sensitivity analysis D = duration, T = return period.

|  | D = 1 h<br>T = 10 y | D = 1 h<br>T = 100 y | D = 24 h<br>T = 10 y | D = 24 h<br>T = 100 y |
| --- | --- | --- | --- | --- |
| Eriz | 32 | 48 | 4 | 5 |
| Trub | 30 | 42 | 4 | 5 |
| StA | 30 | 43 | 4 | 4 |

The R-script implementing the sampling methodology and a description is included in the supplementary material. The minimum and maximum value from Table 5 are used as the range in the sensitivity analysis (Table 6). The maximum value for vegetation weight is taken from a biomass study in Switzerland by Price et al. (2017). For the other parameters, realistic ranges have been assumed. In Table 6 an overview is given of the tested parameters and the ranges used to generate the parameter samples. The precipitation intensity and transmissivity together determine the saturation degree of the soil (equation 12) and are therefore prone to equifinality. We grouped them as an additional parameter, the $I/T$ ratio.

**Table 6.** Parameters used in the SlideforMAP qualitative sensitivity analysis and corresponding ranges for parameter sampling via LHS. $RR_{max}$ and $W_{veg}$ are given as spatially uniform parameters and not computed by the methodology in section 2.5. This is to create scenarios that are comparable with and without single-tree detection.

| Parameter | Unit | Description | LHS Range | |
|---|---|---|---|---|
| $\rho_{ls}$ | m$^{-2}$ | Density of the randomly generated landslides | 0.02 - | 0.10 |
| $\rho_{soil}$ | kg/m$^3$ | Dry soil density | 1.00 - | 1.50 |
| $m_d$ | m | Mean soil thickness | 0.20 - | 1.80 |
| $\sigma_d$ | m | Standard deviation of the soil thickness, as a fraction of $m_d$ | 0.00 - | 0.50 |
| $m_C$ | kPa | Mean saturated soil cohesion | 0.00 - | 12.5 |
| $\sigma_C$ | kPa | Standard deviation of the soil cohesion, as a fraction of $m_C$ | 0.00 - | 0.50 |
| $m_\phi$ | ° | Mean angle of internal friction | 24.00 - | 41.50 |
| $\sigma_\phi$ | ° | Standard deviation of the angle of internal friction | 0.00 - | 5.00 |
| $T$ | m$^2$/s | Soil transmissivity | $10^{-8}$ - | $10^{-3}$ |
| $I$ | mm/h | The precipitation event that is tested | 4.0 - | 48.0 |
| $I/T$ | m$^{-1}$ | Ratio between precipitation and transmissivity | $8.9^{-3}$ - | 1390 |
| $RR_{max}$ | N/m | Maximum lateral root reinforcement | 0.00 - | 15.0 |
| $W_{veg}$ | tonne/m$^2$ | The weight of the vegetation | 0.00 - | 0.10 |

For the model calibration and qualitative sensitivity analysis, 1000 LHS parameter sets were generated per study area by drawing samples from the ranges in Table 6. The number 1000 was chosen arbitrarily for computational constraints. The vegetation is set to a global uniform vegetation, which results in constant root reinforcement and vegetation weight in space. This is necessary because the same runs are used for model calibration and for model sensitivity analysis, where we need such uniform vegetation to ensure that the sensitivity of the (hypothetical) vegetation has an effect on all raster cells of the whole study area (and not only on the actually vegetated cells). The parameter set with the highest AUC value is retained for model calibration. In addition, all 1000 parameter sets are used for a qualitative sensitivity analysis. The response variables are the AUC as a measure for accuracy and the ratio of unstable landslides as a measure for instability. The AUC is chosen for the sensitivity analysis as the main response variable since it expresses the performance relative to the independent landslide inventory. We then consider AUC as a generalized measure of parameter likelihood (Beven and Binley, 1992) and assess how selected best parameter sets (e.g. the best 10 % out of the 1000 sampled sets) are distributed (parameter subsampling).

### 3.4.3   Vegetation parameter scenario analysis

SlideforMAP has potential in testing the effect of different vegetation scenarios on the landslide probability. For this research, besides the reference scenario for model calibration and sensitivity analysis (global uniform vegetation), three additional scenarios are tested: i) without vegetation, ii) with uniform vegetation in forested areas and iii) with a fully diverse vegetation

based on single-tree detection. The single-tree version uses the input data as mentioned in section 3.2. The forested areas are defined as areas where the single tree detection method leads to a lateral root reinforcement (Fig. 9) which is not equal to zero.

## 4 Results

### 4.1 Sensitivity analysis

We use the 1000 model simulations corresponding to the 1000 generated parameter sets per study area for a sensitivity analysis of the model. The objective of this analysis is to quantify how the distribution of AUC values and of the landslide probability vary as a function of the parameters. Applying the parameter subsampling technique (see section 3.4.2), we see that for some parameters, the histogram shape (i.e. their marginal distribution) does not significantly deviate from the initial uniform distribution (from which we sampled), even if we retain only the best 10% (in terms of AUC) of all parameter sets (Fig. 5). This

apparent lack of sensitivity does not necessarily mean that the model is not sensitive to this parameter; in fact, the sensitivity could be hidden by strong parameter correlation (see Bárdossy, 2007, for a discussion of how uniform marginal distributions can result from strong parameter correlation). Our addition of the $I/T$ ratio gives a hint at such behaviour. From Fig. 5 it appears that the sensitivity to AUC of the $I/T$ ratio is slightly stronger than either the precipitation or transmissivity independently. Some parameters, in exchange, show very strong sensitivity of their marginal distributions if only the best (in terms of AUC)

parameter sets are retained. For the Trub case study (Fig. 5), we see that the mean thickness $m_{\mathrm{d}}$, the mean cohesion $m_{\mathrm{C}}$, the $I/T$ ratio and the transmissivity show a well defined maximum around the parameter values retained for calibration (the best performing ones). This suggests a good sensitivity of the model to these parameters in terms of model performance. Two of these three parameters also show a clear sensitivity if we retain subsamples that lead to successively higher unstable landslide ratio (Fig. 6): high unstable ratios are obtained for high $m_{\mathrm{d}}$ values or for low $m_{\mathrm{C}}$. Also for $RR_{\mathrm{max}}$, highest ratios are clearly

obtained for low lateral root reinforcement values (for all three case studies, Fig. 6, Supplementary Material S-2, S-4). For transmissivity, while it shows a clear effect on model performance, the relation between its marginal distribution and the ratio of unstable landslides is less visible.

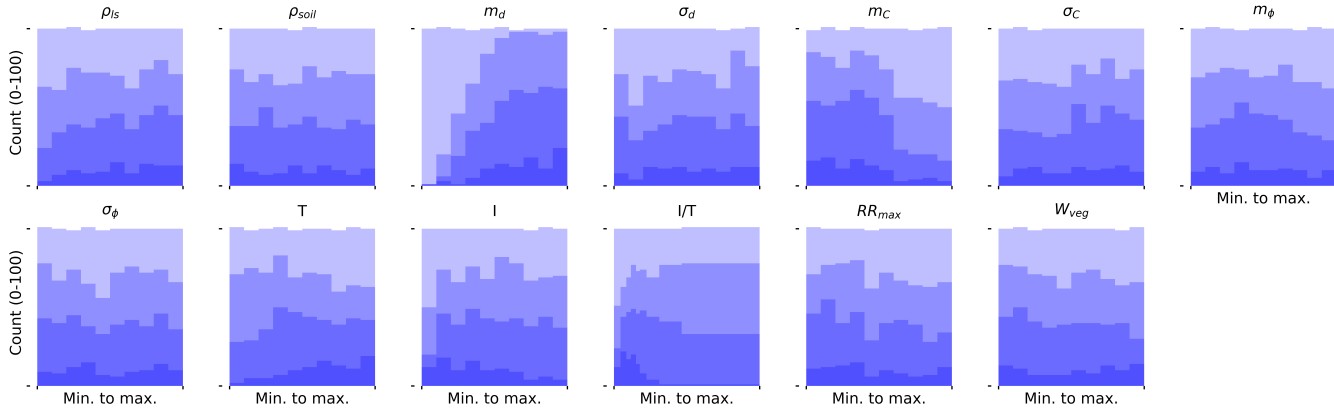

**Figure 5.** Histograms of different subsamples of the LHS parameter sets for the Trub study area. The shading (from light to dark) corresponds to subsamples retaining only the $x\%$ highest parameter sets in terms of AUC; the shown fractions are: 1, 0.7, 0.4, 0.1.

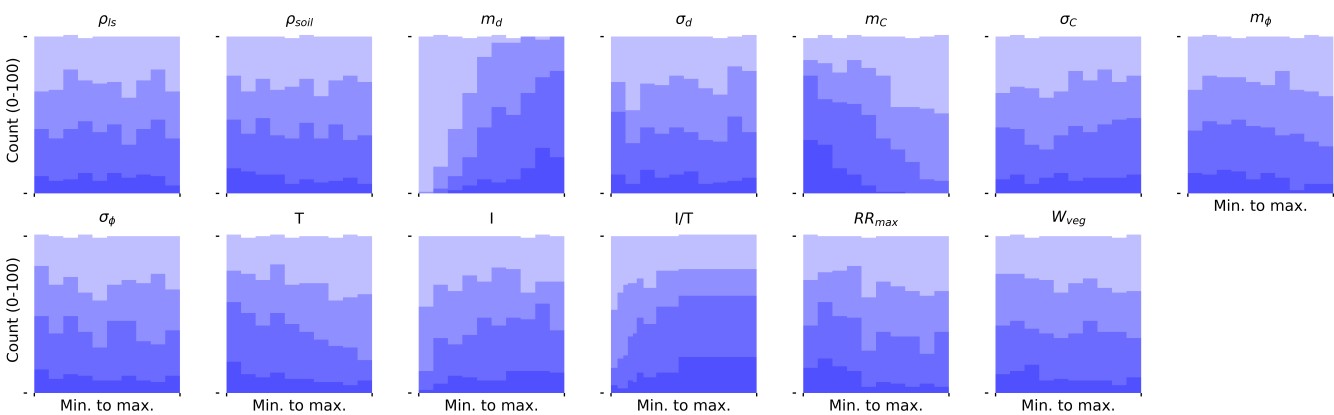

**Figure 6.** Histograms of different subsamples of the LHS parameter sets for the Trub study area. The shading (from light to dark) corresponds to subsamples retaining only the $x\%$ highest parameter sets in terms of Unstable ratio; the shown fractions are: 1, 0.7, 0.4, 0.1.

## 4.2 Model calibration

Based on the generated 1000 parameter sets, we identified the parameter set that resulted in the highest AUC value and assumed
this to be an optimal calibration of the model. These calibrated parameter sets for each study area and their AUC values are shown in Table 7 together with the ratio of generated HLs that are unstable.

**Table 7.** Outcome of the Monte Carlo-based calibration: the parameter sets per study area resulting in the highest AUC value. The last row shows the ratio of unstable HL resulting from these parameter sets.

| Parameter | Eriz | Trub | StA | Parameter | Eriz | Trub | StA |
|---|---|---|---|---|---|---|---|
| $\rho_{\mathrm{ls}}$ | 0.095 | 0.041 | 0.093 | $T$ | 0.000148 | 0.000473 | 0.000582 |
| $\rho_{\mathrm{soil}}$ | 1.40 | 1.20 | 1.49 | $I$ | 40.3 | 24.2 | 14.0 |
| $m_{\mathrm{d}}$ | 1.62 | 1.02 | 1.78 | $I/T$ | 0.077 | 0.014 | 0.007 |
| $\sigma_{\mathrm{d}}$ | 0.32 | 0.13 | 0.31 | $RR_{\mathrm{max}}$ | 12.3 | 4.7 | 10.3 |
| $m_{\mathrm{C}}$ | 4.29 | 1.75 | 2.51 | $W_{\mathrm{veg}}$ | 0.05 | 0.02 | 0.03 |
| $\sigma_{\mathrm{C}}$ | 0.43 | 0.32 | 0.30 | AUC | 0.924 | 0.940 | 0.693 |
| $m_{\phi}$ | 34.0 | 29.3 | 26.0 | Unstable ratio | 0.197 | 0.308 | 0.387 |
| $\sigma_{\phi}$ | 0.37 | 1.39 | 0.92 | | | | |

Parameter consistency between the study areas appears to be visible in $\rho_{\mathrm{soil}}, m_{\mathrm{d}}, m_{\mathrm{C}}, \sigma_{\mathrm{C}}, m_{\phi}, \sigma_{\phi}, T$ and $W_{\mathrm{veg}}$. Other parameters show stronger variation, relative to their LHS range, between case studies. A realization of the shallow landslide probability computed with SlideforMAP for the three areas with their calibrated parameter set is given in Fig. 7.

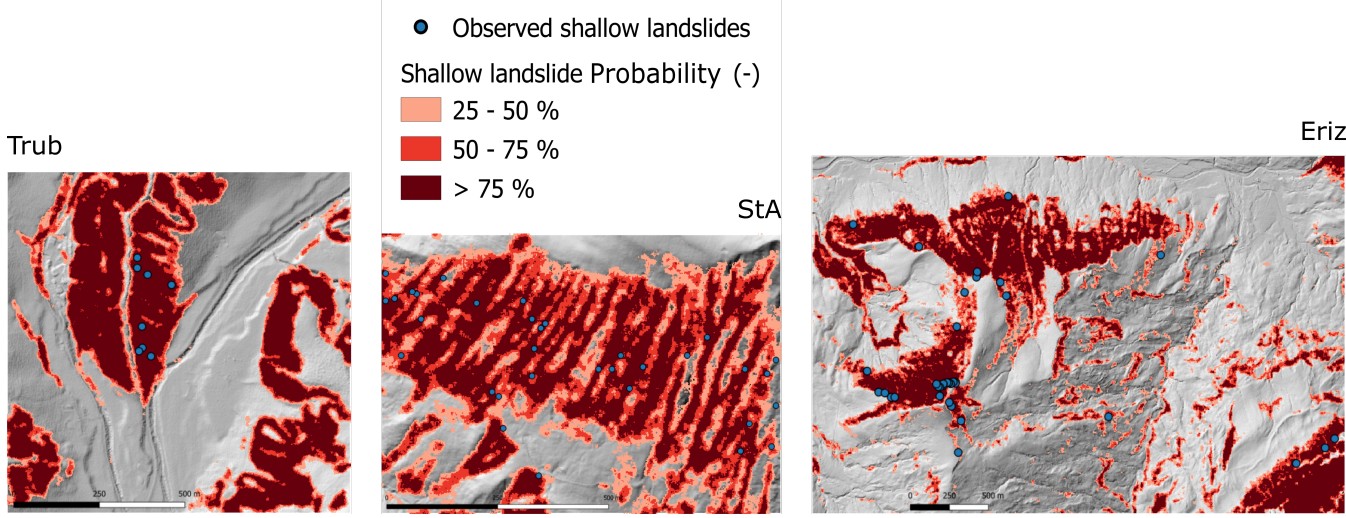

**Figure 7.** Overview of the landslide probability of the study areas simulated with the calibrated parameter sets of Table 7. Added as blue points are the observed landslides from the inventory.

In general, the model represents well the spatial distribution of the shallow landslides from the inventory. A cumulative plot of the shallow landslide probability for the study areas based on Fig. 7 is given in Fig. 8

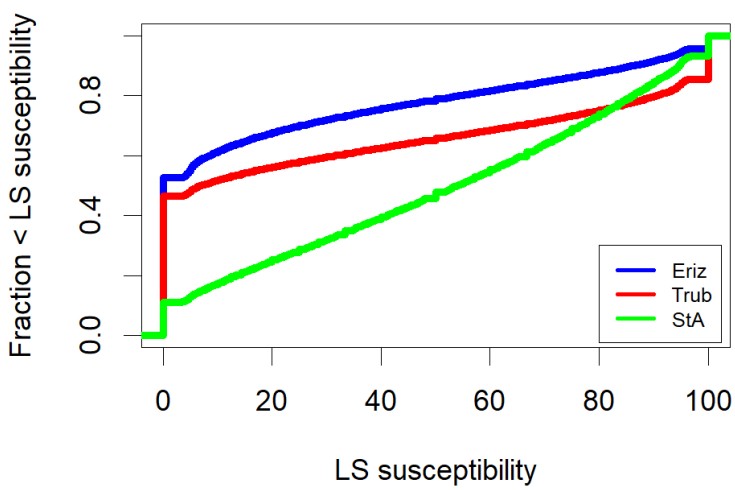

**Figure 8.** Cumulative plots for shallow landslide probability in the study areas, derived from the results in Fig. 7.

### 4.3    Mechanical effects of vegetation

To test the impact of vegetation on the model behavior, we compare the different vegetation scenarios. The spatial distribution of lateral root reinforcement, resulting from single tree detection and SlideforMAP, is given in Fig. 9.

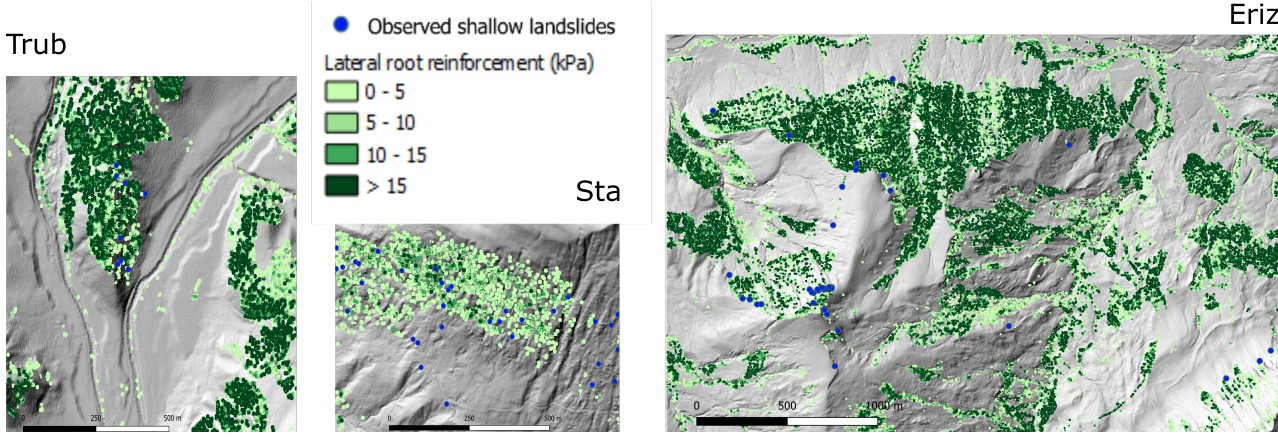

**Figure 9.** The spatial distribution of maximum root reinforcement (equation 8) in the study areas as used in SlideforMAP

The selected vegetation scenarios (no vegetation, global uniform vegetation, forest area uniform vegetation, single tree detection) affect the computation of the vegetation weight, the lateral root reinforcement and the basal root reinforcement. The latter

is due to its dependence on lateral root reinforcement (equation 11). Accordingly, the vegetation scenario has a direct impact on SF (equation 1, 3, 4) and on $P_{ls}$ (equation 14). For the analysis, we use the optimal parameter set from Table 7, obtained for a global uniform vegetation cover. The model runs are repeated 10 times to produce an average result and to show the variation

from the probabilistic approach. Due to sampling from distributions, every realization produces a (slightly) different result. The resulting influence of the selected vegetation scenarios on AUC and on the ratio of unstable landslides is given in Table 8. The results from Table 8 and Table 9 display that the model is sensitive to the vegetation scenarios and that it predicts lower ratios of unstable ratios for vegetated scenarios as compared to the unvegetated scenario. This underlines the value of the model for future scenario analyses.

**Table 8.** AUC and unstable ratio under different vegetation scenarios with the optimal parameter sets of Table 7 and averaged over 10 runs. The "Overall" is composed of the mean value of all three study areas. In the global uniform vegetation scenario, the reference scenario is used during parameter optimisation.

| | | AUC | | | | Unstable ratio | | | |
|---|---|---|---|---|---|---|---|---|---|
| | | **Overall** | **Eriz** | **Trub** | **StA** | **Overall** | **Eriz** | **Trub** | **StA** |
| mean | Global uniform vegetation | 0.808 | 0.910 | 0.844 | 0.669 | 0.299 | 0.197 | 0.311 | 0.388 |
| | Forest area uniform vegetation | 0.801 | 0.901 | 0.861 | 0.641 | 0.400 | 0.250 | 0.371 | 0.580 |
| | Single tree detection | 0.831 | 0.925 | 0.925 | 0.644 | 0.336 | 0.199 | 0.217 | 0.593 |
| | No vegetation | 0.785 | 0.880 | 0.854 | 0.622 | 0.475 | 0.309 | 0.413 | 0.704 |
| Std. dev. | Global uniform vegetation | 0.017 | 0.007 | 0.029 | 0.016 | 0.001 | 0.000 | 0.001 | 0.002 |
| | Forest area uniform vegetation | 0.021 | 0.008 | 0.039 | 0.016 | 0.001 | 0.000 | 0.001 | 0.002 |
| | Single tree detection | 0.012 | 0.005 | 0.011 | 0.021 | 0.001 | 0.001 | 0.001 | 0.001 |
| | No vegetation | 0.025 | 0.013 | 0.044 | 0.019 | 0.002 | 0.001 | 0.002 | 0.002 |

ROC curves corresponding to the scenarios with repetitions as presented in Table 8 are given in Fig. 10. Significance of the differences between vegetation scenarios from Table 8 as given in Table 9.

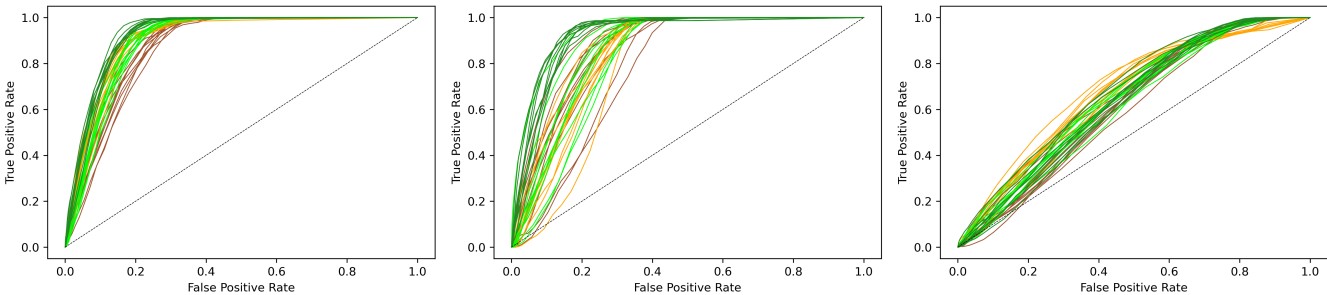

**Figure 10.** ROC curves of the 10 runs per vegetation scenarios from Table 8. Orange: Global uniform vegetation, light green: Forest area uniform vegetation, Dark green: Single tree detection, Brown: no vegetation. Corresponding study areas from left to right are: Eriz, Trub and StA.

**Table 9.** Significance of the difference in distribution between results of vegetation scenarios at a 90 and 99 % confidence level. Scenario names are shortened. Significance measured by Welch's t test (Welch, 1947). T (True) indicates a significant difference, F (False) indicates no significant difference. Three indicator per cell are related to the three study area, ordered as: Eriz, Trub, StA.

|  |  | 99% | | | |
|---|---|---|---|---|---|
|  |  | Global | Forest | Single | No |
|  | Global | - | T,F,T | T,T,T | T,F,T |
| 90% | Forest | T,F,T | - | F,T,F | T,F,F |
|  | Single | T,T,T | T,T,F | - | T,T,F |
|  | No | T,F,T | T,F,T | T,T,T | - |

## 5   Discussion

It is important to point out that the inventory to which the model performance is calibrated plays a key role in all the results discussed below. The inventory was obtained after triggering rainfall events, for which the precipitation intensity, duration
and the spatial distribution are not known precisely. Despite this shortcoming, the inventory represents a unique source of information and the spatial localisation of the landslides can be assumed to be of high quality. Below, we discuss the model behavior as a function of the different model parameter groups and the performance of the model and give directions for future research.

### 5.1   Soil parameters

The best performing parameter sets show high values for the soil thickness for all study areas (by comparing the values of Table 7 and Table 6). The qualitative sensitivity analysis (Fig. 6) also shows that the highest unstable ratios are obtained for

highest soil thicknesses; this indicates that a certain minimum soil thickness is required for landslide triggering, which is in line with previous findings by D'Odorico and Fagherazzi (2003) and by Iida (1999). In these studies, soil thickness is noted as the conditional factor for landslide triggering along with precipitation intensity and duration. The best performing parameter sets display cohesion values with a clear tendency to low values for all three study areas (Fig. 6, Supplementary Material S-4, S-2), which suggests that the observed landslides can only be reproduced with low soil cohesion for all case studies. The mean angle of internal friction appears to show consistency for a low value (Table 7). The sensitivity of the AUC and unstable ratio on the angle of internal friction, however, appears to be small (Fig. 6 and Fig. 5).

## 5.2 Hydrological parameters

Soil transmissivity showed considerable sensitivity to the AUC (Fig. 5) and the values are consistently high for all three case studies for the parameter range (Fig. 7), which is a hint that a correct estimation of soil transmissivity is paramount for a reliable estimate of shallow landslide occurrence. Regarding precipitation intensity, we see variability between the best values for the three case studies and minor univariate sensitivity of the model performance or the model output (ratio of unstable landslides). The application of the TOPOG approach has the major shortcoming that it assumes a groundwater gradient parallel to the surface gradient. It has been shown in the past that this assumption decreases the accuracy of water content simulations as compared to distributed dynamic hydrological models (Grabs et al., 2009). However, as discussed earlier, it has also been shown in the past that macropore flow is omnipresent in landslide triggering and SlideforMAP has been parameterized assuming an important role of macropore flow. In macropore-driven systems, steady state groundwater flow can be reached (see Introduction), which implies that the TOPOG assumption holds well in this case. Due to the lack of detailed meteorological data, the precipitation intensity and duration is unknown. This makes computation on the exact pore pressure during the landslide event impossible. The precipitation intensity / transmissivity ratio ($I/T$) is assumed to include both precipitation intensity and transmissivity sensitivity. This is reflected in Fig. 5 and Fig. 6. The calibrated values for $I/T$ ratio and subsequent pore pressure computation should be regarded as a measure for landslide propensity. In the landslide inventory underlying the study here, the dominant soil types are GM (silty gravel), GC (clayey gravel) and CL (low plasticity clayey silt); accordingly. Due to large pore size, we can assume that the TOPOG assumptions are valid for a wide range of the domain (for GM and GC soil type), even if it probably holds less well for the CL soil types.

## 5.3 Vegetation

A key aspect of the model is the use of single tree detection to parameterize vegetation, a method that was previously found by Menk et al. (2017) to be reliable to detect single trees and derive their DBH's from the detected tree heights for sloped forests. As mentioned in Section 3.2, we found for the selected case studies that single tree detection provides the best results in terms of correct number of trees counted if applied on a 1 m resolution DSM with a 3 cell kernel Gaussian filter. This is in line with the results of Menk et al. (2017) who found in a similar scenario-testing approach that a 1 m resolution DSM with no Gaussian correction provided the most accurate results, noting, however, that the difference in performance between these two methods (with and without Gaussian filter) is small. In SlideforMAP, we do not only consider basal but also lateral root reinforcement.

This is unique for shallow landslide probability models. As shown in the sensitivity analysis (Fig. 6), $RR_{\mathrm{max}}$ has a clear effect on the ratio of unstable landslides, with low values leading to high ratios. In the SlideforMAP workflow and calibration, a fixed relationship between the lateral and the basal root reinforcement is assumed, accordingly, the model sensitivity cannot be attributed to $R_{\mathrm{lat}}$ or $R_{\mathrm{bas}}$. Mobilization of the lateral root reinforcement in the SlideforMAP workflow is independent of time and not countered by passive earth pressure. A shortcoming in this parameterization of the effect of vegetation is the assumption

of uniform forest structure and a uniform tree species (beech) within a landslide area. The field recordings in the StA area of Moos et al. (2016) show that the forest consists mainly of Norway spruce. For the Trub and Eriz area, visual interpretation of aerial photos allowed us to identify mixed forests with Norway spruce and beech. The latter are known for having a high root reinforcement and therefore the beech assumption will overestimate both the lateral and the basal root reinforcement (Gehring et al., 2019). Vegetation weight shows no clear relation to both the AUC and the unstable ratio (Fig. 5, Fig. 6). However,

this does not mean that vegetation weight does not influence the response variables. The relationship could depend on other parameters and therefore obscured (Bárdossy, 2007). In contrast to the soil and hydrological parameters, vegetation configures both the magnitude and the spatial pattern of the probability. Vegetation can be modified by land management practices with relative ease (Amishev et al., 2014) and is therefore of ultimate importance in shallow landslide mitigation.

## 5.4    Implementation of the mechanical effects of vegetation

In Table 8 it can be seen that the vegetation scenario has a considerable impact on the modelled unstable ratio for all study areas. Unstable ratio is lowest in the single tree detection scenario for the Trub study area. In the StA and Eriz study area, it is the lowest for the uniform vegetation. We assume this is caused by the low calibrated uniform root reinforcement in Trub and a higher value in the other study areas (Table 7). Both single-tree detection and uniform vegetation are determined to have the ability to decrease instability. From a practical perspective vegetating parts of a study area is more realistic than uniformly

vegetating the whole area. Influence of the vegetation scenario on the AUC is present, with an absolute mean increase of 0.023 AUC points between single tree detection and uniform vegetation and to forest uniform vegetation and unvegetated of 0.030 and 0.046 AUC points respectively (Table 8). Additionally the performance improvement can be described relatively in terms of percentage of extra AUC gained (AUC range from 0.5 - 1.0) between two vegetation scenarios. For the overall single tree detection compared to uniform vegetation, forest uniform vegetation and no vegetation this is 8%, 10% and 16% respectively.

Results in Table 9 show that the differences are relevant for the uniform scenario in all study areas at both a 90% and 99% confidence level. The difference between single tree detection and no vegetation is relevant for all confidence levels and study areas except for the StA study area at 99% confidence. The difference between single tree detection and forest uniform is more ambiguous, with notably a significant difference at a 90% confidence level in the Trub and Eriz study area. This is likely related to the forest uniform scenario being most close to single tree detection in the distribution of root reinforcement of all scenarios.

In both Eriz and Trub, the single tree detection is the best performing scenario. Our overall finding that the model output is sensitive to the vegetation scenario and gives second lowest values in unstable ratio and highest values in AUC for single-tree detection. We argue that even though the model is calibrated on a global uniform vegetation scenario (Table 7) and the single-tree detection gives a significantly better overall performance, single tree detection is more accurate in assessing

shallow landslide susceptibility (Table 8 and Table 9). Adding to this explanation is that in these study areas, where slope angle is a highly predictive factor, even marginal gains in AUC due to vegetation are important and the result of extensive parameterization. Our analysis is in line with the findings of Roering et al. (2003), who state that single tree based modelling, including the tree dimensions, has the highest accuracy in the prediction of shallow landslides. Moreover, Vergani et al. (2014) state that a site specific estimation of vegetation and root extent is essential in the correct estimation of root reinforcement.

## 5.5 Model performance

As pointed out by Corominas et al. (2014), the absolute values of AUC are dependent on the characteristics of the study area. In larger areas, with low overall landslide activity, the AUC will overestimate the predictive performance. This most likely explains why the StA study area has a low overall AUC compared to Eriz and Trub (Table 8). In particular, StA study area shows a higher prevalence of steep slopes. The Trub and the Eriz study area show both relatively high AUC values, indicating high model performance, with very similar AUC values; this is in agreement with a similar occurrence of steep and gradual slopes in these areas. Another explanation for the discrepancy in model performance between the study areas could be the assumption that all trees are beech trees. This does not hold equally well for all three study areas. Based on visual inspection and on elevation, the mismatch between actual vegetation and this assumption is probably most pronounced in the StA area, where the dominant tree species appears to be Spruce. Though no published data is available, it can be estimated from the work of Moos et al. (2016) that the root reinforcement of a spruce forest is lower than that of a beech forest, but this cannot confirmed by our parameter analysis at this stage.

A comparison between the shallow landslide density (Table 2) and the calibrated unstable ratio (Table 7) shows moderate consistency. The Eriz and Trub study areas have a low unstable area corresponding to a low shallow landslide density. StA both has a higher landslide density and higher unstable ratio. From the consistency in Table 7 and the sensitivity analysis results of Fig. 5, it can be concluded that the main configuration of the model lies in the parametrization of the mean soil thickness, the mean cohesion and the $I/T$ ratio. In addition, the vegetation scenario strongly influences the model performance and is of high influence on calculated shallow landslide probability (Table 8). Equifinality between the parameters in the qualitative sensitivity analysis is likely as it is very common in similar multi-parameter modelling (Beven and Binley, 1992). However, we believe, the sensitivity as observed in Fig. 5 is valid and a qualitative indicator for important parameters in SlideforMAP. The calibrated optimal parameter set (Table 7) is still within realistic bounds as is the ranges for the sensitivity analysis. In addition, the calibrated combination of mean friction angle (26 - 34 °) and mean soil cohesion (1.75 - 4.29 kPa) are possible, according to Supplementary material Table A1. Finally, we would like to add here that the case study dependence of the used model performance measure is a limitation that typically occurs for all model performance measures that compare the model behavior to some reference model (Schaefli and Gupta, 2007) (the reference model for AUC is a random process). Accordingly, we cannot compare the performance of SlideforMAP to other published AUC values despite of the fact that values above 0.8 are considered as indicating good performance (e.g. Xu et al., 2012).

## 5.6 Comparison to other slope stability models

The main advantage of SlideforMAP to other models is the more realistic approach to implement root reinforcement. It includes a spatial distribution in both the basal and lateral root reinforcement and the focus on second stage of the activation phase in accordance with the Root Bundle Model as described in Gehring et al. (2019). Compared to previous slope stability models that include the effect of root reinforcement, SlideforMAP uses a more realistic implementation of root reinforcement based on recent knowledge of shallow landslides triggering mechanisms and root reinforcement activation (Schwarz et al., 2012, 2013; Cohen and Schwarz, 2017). In particular, only part of the lateral root reinforcement under tension is considered for the force balance calculation. Moreover, the spatial distribution of root reinforcement as function of forest structure is included. The assumptions made in SlideforMAP allow a probabilistic calculation at regional scale that are not possible with more complex models such as SOSlope (Cohen and Schwarz, 2017). In comparison to more simple models based on infinite slope calculations (Pack et al., 1998; Montgomery and Dietrich, 1994, SINMAP,SHALTAB), SlideforMAP considers the effect of lateral root reinforcement on landslide of different sizes. SINMAP with a homogeneous root reinforcement is comparable to our global uniform vegetation scenario (Table 8). A version of SINMAP with no root strength is comparable to our no vegetation scenario. When no vegetation data is available or complexity is not desired, these are valid option to assess shallow landslide susceptibility in a probabilistic way.

A hydrological and slope stability model identical to SlideforMAP is applied in Montgomery et al. (2000), which is used to estimate sediment yield resulting from forest clearing. This is comparable to our global uniform vegetation scenario as well. Their result of a high significance of root reinforcement is in line with our findings. Other differences in the model approach are the assumption of fixed landslide dimensions, including soil thickness. In addition, the root reinforcement is assumed to act around the full perimeter of the landslide. In its approach, SlideforMAP shares many similarities with PRIMULA, as developed by Cislaghi et al. (2018), which applies a probabilistic approach and a spatially distributed root reinforcement as well. the PRIMULA root reinforcement is based on a stand scale approach rather than single-tree detection though. The AUC values in this paper are higher, but that could be the result of different characteristics of the study areas and our parameter optimization by the qualitative sensitivity analysis. Other differences as compared to PRIMULA are their assumption of lateral root reinforcement along the entire landslide perimeter, the inclusion of lateral soil cohesion simultaneously with lateral root cohesion, the assumption of rectangular shaped landslides rather than elliptical ones and a different landslide surface area distribution. 3DTLE (Hess et al., 2017) is a deterministic landslide susceptibility model with a similar detailed spatially heterogeneous inclusion of root reinforcement. Differences are their deterministic approach and the assumption of a simultaneous maximum of tension and compression forces.

## 5.7 Future research

SlideforMAP uses a relatively simple hydrological module to estimate soil saturation. The used TOPOG approach could be improved and multiple papers have presented simple to more advanced rewriting of formulas (e.g. Beven and Freer, 2001; Blazkova et al., 2002). Common denominator is the inclusion of time dependency, since the stationary flow assumption rarely,

if ever, holds in nature. This time dependency is a solution to simulate a different response to a precipitation event at different locations within a study area. Future work could also focus on improving the vegetation module by including different tree species (those that are often used in protection forest) in the parametrization of lateral root reinforcement (equation 10). For practical application of SlideforMAP we have not found a specific lower boundary in landslide density, to still generate reliable results. More specific testing on this would be useful for future application of SlideforMAP. A comparison between SlideforMAP and SHALSTAB and/or SINMAP would be interesting. It can validate whether the uniform vegetation scenario in SlideforMAP produces similar results to these models in terms of shallow landslide probability. Finally doing a validation over study areas with a larger shallow landslide inventory would be a vital procedure to further analyze the SlideforMAP model.

## 6  Conclusions

In this paper, we present a probabilistic model to assess shallow landslide (landslides with a scar thickness < 2 m) probability. The main motivation to develop yet another model is to provide a detailed inclusion of the influence of root reinforcement. Its application is illustrated based on three mid-elevation case studies from Switzerland, for which a detail landslide inventory is available. The model has a total of 21 parameters, of which 12 are calibrated using the AUC of the Receiver Operator Curve as performance measure to identify the best parameter set among a large set generated using Latin hyper cube sampling. The AUC maximum values for the three study areas vary between 0.64 and 0.93 under a single tree detection vegetation scenario, which reflects an overall good model performance. Our model parameter analysis has shown that soil thickness, precipitation intensity to transmissivity ratio and soil cohesion, are the key parameters to predict slope stability in the studied mountainous regions. A major focus of the presented work was the assessment of the model's ability to study scenarios of vegetation distribution. Comparison of different scenarios ranging from uniform to single-tree detection-based vegetation clearly showed that the model output, in terms of shallow landslide probability, is sensitive to the spatial distribution of vegetation. Additionally, in two of our three study areas, the single-tree detection scenario provides significantly (Welch's t test confidence > 99 %) higher AUC values. Accordingly, the model is fit for future scenario analysis, including e.g. different protection forest management scenarios. In fact, a single-tree scale model parameterization provides the opportunity to run hypothetical vegetation scenarios reflecting on small scale managements strategies or disturbances. Future improvements in the hydrological approach, concerning a more catchment based approach to compute saturation degree, could likely further improve the performance of SlideforMAP.

 **Appendix A: Appendix**

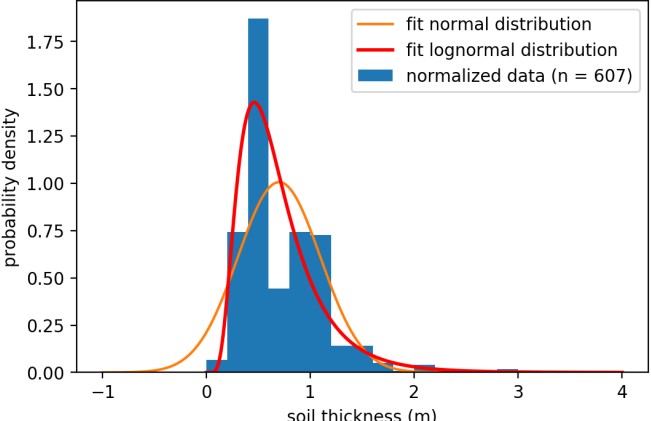

**Figure A1.** Plot of the probability density of the soil thickness data from the BAFU dataset as used in this paper. The best fit is given of a normal and a log-normal distribution. The mean square errors are 0.096 and 0.053 for the normal and log-normal fit respectively.

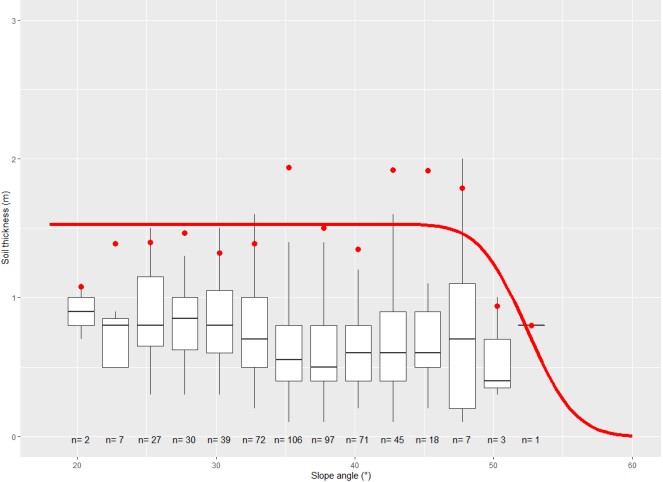

**Figure A2.** Shallow landslide Slope soil thickness relationship as used in this research. Boxplots are classes with a width of 2.5 Slope units. The red dots are the 95[th] percentile per class. The red line is the fit of equation 7 to the 95[th] percentiles.

**Table A1.** The hybrid table for the soil cohesion and angle of internal friction for the relevant set of USCS soil classes. Derived from laboratory experiments (VSS-Kommission, 1998; Dysli and Rybisar, 1992) and combined in this research to exclude values that seemed unrealistic.

| USCS soil class | Mean soil cohesion | Std. dev. soil cohesion | Mean friction angle | Std. dev. friction angle |
|---|---|---|---|---|
| SM | 0 | 0 | 34.5 | 5.0 |
| CL-ML | 0.4 | 1.3 | 32.7 | 4.8 |
| GM | 0.0 | 0.0 | 35.0 | 5.0 |
| GC-GM | 5.0 | 5.0 | 33.0 | 3.0 |
| CL | 6.2 | 11.3 | 27.1 | 5.2 |
| OL | 2.5 | 5.0 | 32.8 | 2.2 |
| GC | 20.0 | 52.9 | 31.4 | 3.6 |

*Data availability.* All data used in this research is open data. The topographical data and the landslide inventory as used in this research are published on Zenodo

*Author contributions.* A.A. collected the landslide inventory and made it ready for use. D.C. and M.S. developed the basic concept of SlideforMAP. L.D. contributed in the further development. F.Z. executed further development, the sensitivity analysis and testing. F.Z. is the main writer. B.S. L.D. C.P. A.A. and M.S. revised the text. C.P. and M.S. organized funds.


*Competing interests.* The authors declare no conflict of interest.

*Disclaimer.* The shallow landslide probability maps generated by SlideforMAP are a guideline and should be interpreted by an expert before application.

*Acknowledgements.* We thank the STEC (Smarter Targeting of Erosion Control) project by the Ministry of Business, Innovation and Employment of New Zealand for the financial support. In addition we would like to thank the two anonymous reviewers and a community review by David Milledge. Their contribution was of great improvement to the quality of this paper.

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
