# Peer review of "Introducing SlideforMAP; a probabilistic finite slope approach for modelling shallow landslide probability in forested situations"

_Natural Hazards and Earth System Sciences, 2021_

## Author Response (AR1)

**nhess-2021-140 -Introducing SlideforMap; a probabilistic finite slope approach for modelling shallow landslide probability in forested situations - van Zadelhoff et al., 2021**

**Response to the comments of the reviewers**

In these responses, we provide the original comments (in italics) and our related responses. The corresponding changes in the manuscript are indicated in the track-change version submitted along with the revised paper at the end of this response.

**Response to Reviewer #1**

*I reviewed with interest this manuscript for possible publication in NEHSS journal. The work describes a comprehensive modeling tool to assess shallow landslides initiated by rainfall, in a probabilistic framework. The manuscript provides an interesting contribution in this field, although some aspects are strongly simplified, in contrast with others. The scientific quality is good, the reading is agile although the manuscript is overall a bit long and often dispersive. The literature review can be improved with additional appropriate references of strictly related works. The description of the climate forcing that initiates (or not) the landslide events requires significant improvement. To my opinion, the work can be published after some important clarifications and revisions.*

Thanks for the general positive assessment. We gave the manuscript the best we could to shortening without skipping vital points.

*1. Literature review (introduction/discussion).*

*The discussion on the impacts and costs of the natural hazards, from the point of view of insurance institutes, is interesting. However, in general, I found the introduction a bit dispersive, lacking in some aspects. The work of Dietrich and Montgomery, 1994, (SHALSTAB) represents the pioneering work within this approach, and it has been followed by many other deterministic work that gave different contributions in improving the hydrological modeling at support for the shallow landslide, such as the cited Iverson 2000, and, additionally, Rosso et al., 2006; Claessens et al., 2007, Arnone et al., 2011; Lepore et al., 2013; Simoni et al., 2008, Baum et al., 2002 (TRIGRS), Montrasio et al., (2011) (SLIP) (among the others). With regard to the effect of vegetation, the aspects related to the hydrological effects should be at least discussed, which can sometime be even more significant than the mechanical ones (Feng et al., 2020). An interesting review are by Chae et al., 2017, Gasser et al., 2019 and the just published by Masi et al., 2021.*

We extended the introduction and included more references to the pioneering and subsequent research in order to embed our research better. We added a discussion on the hydrological effects of vegetation with reference to Feng et al., in the introduction L96-97.

*2. Definition of the stability problem.*

*I found the definition of the problem of stability estimation (section 2.2, Figure 2) a bit misleading. It is not clear the definition of the volume of soil to which forces are applied. In the method of the limit equilibrium, under the hypothesis that the width of the landslide is sufficiently large so that the deformations are in the plane parallel to the soil thickness Hsoil (i.e. perpendicular to the elliptic*

*landslide in figure 2), forces are assessed by considering a 'slice' of soil with unit width (in the direction parallel to the elliptic landslide plane). Figure 2 is confusing and the planes of forces are not well drawn. The limit equilibrium method (and infinite slope model) is based on the hypothesis of large and elongated element with respect to the soil thickness, so that a unit in width element can be considered. Also, Pwater is not indicated in the Figure 2. According to the definition in the manuscript, Rlat and Fres apply on different planes. I suggest to modify in a 3D perspective the Figure 2 and specify the hypothesis/assumptions.*

We adjusted figure 2 to a 3D perspective in order to enhance clarity on the dimensions, volume and force application planes of the assumed shallow landslide. In addition, we added the water pressure (Pwater) as a subtraction of perpendicular force and emphasize the points or fields on which the forces apply.

*3. Hydrology and precipitation.*

*Here is my main comment. The proposed modeling framework addresses shallow landslides that are initiated by rainfall, which is the triggering factor. The approach used (based on TOPMODEL) is extremely simplified because based on steady state conditions, which do not take into account the transient of the hydrological processes (Chae et al., 2017). The authors declare the limitation of the approach used in the discussion section, but this should be clearly stated soon in the methodology. As correctly written by the author, the stationarity is supposed to be reached within the hour of timestep. Clearly, this cannot be largely verified. That said I arise two more critical issues that are not mentioned by the authors: Under unsaturated conditions, soil (especially fine and clayey soils) exerts a strong water uptake effect due to suction, which leads to an apparent 'hydrological' cohesion. This represent a further limitation of the Montgomery and Dietrich approach that the authors should mention (see, works mentioned in Chae et al., 2017, e.g. Lepore et al., 2013).*

We added limitations of the used hydrological approach more explicitly in the methodology L244-249.

*In the description of the model application (section 3.4.2) it is not clear how rainfall initiating events are selected. If I understood well, only events of 1hour duration are selected, whose intensity is identified from the Depth-Duration-Frequency (DDF) curve at different return periods (i.e. from 10 to 100 years). Therefore, I guess 10 events of 1 hours are simulated. Is that correct? If so, it should be explained and justified the reason of analyzing events of only 1 hours, which cannot be 'critical' for landslide initiation. Authors should deeply clarify this part in the manuscript, explain the methodology used to define the events, and report the parameters of the DDF curves.*

We emphasized the choice for 1-hour events (assumed macro-pore activation time period) in the methodology L256 and the parametrization of the DDF curves in the supplementary material.

*4. Data inventory*

*The proposed methodology used to characterize the hypothetical landslides (extent) is strictly dependent on the data inventory (section 2.3), as also stated somewhere by the authors. However, it is important that the observed landslides used to characterize the model are of the same type, according to the hypothesis of the stability model used and all triggered by rainfall. Is it so? Please specify.*

We specified the triggering mechanism and assumed representativeness for Switzerland in the data section L334-335.

*With regard to the best set of parameters, my question is: are the found parameters consistent and realistic?*

Consistency in the found parameters is arguable. We suspect equifinality is at play. However, with the data available to us, we believe we made the most realistic assumptions on the parameter ranges. We added a comment on consistency and realism to sensitivity analysis discussion section L540-545, 550.

*For example, I argue the choice of including the precipitation intensity as calibration parameter. As discussed in the previous comment, rainfall represents the triggering forcing and it is a dynamic variable. Ideally, we should know the precipitation intensity associated to each observed landslide. Otherwise, if used as parameter, it seems that the model is tuned ad hoc just to reproduce the past events. If so, which could be its utility?*

We agree that in an ideal case this should be known, however no detailed information is available. Therefore, we have to rely on the more simplified steady state approach. We added the lack of detailed information in the data section to justify our approach L335-336.

*Additionally, it would be interesting to see the AUC curves for the calibrated and the best model combinations. The shape of the curve also tells about the model performance. Then, to my opinion, sensitivity analysis should go before the model calibration. Normally, calibration is done on parameters that are more sensitive. I understand figure 7 and 8, but not sure this is the most efficient way to verify the sensitivity of the parameters. I am curious to see how, for example, the landslide probability varies with the chance of parameters values. This test could be shown with the least and most sensitive parameters.*

In our opinion the sensitivity of all these parameters is interesting and can help in future development of SlideforMap or other models employing a similar method. We specified this choice in the description of the sensitivity analysis. As suggested by the reviewer, we added the corresponding AUC curves to the results.

*The result of high m_f and low m_c is quite obvious; as the author clearly say in the discussion, and as found by other past works, in the end only few parameters really affect the process: the geometry of the slope (i.e. the soil thickness), the mechanical properties (i.e. friction angle) and the characteristic of the trigger (i.e. precipitation) whose effects are controlled by the soil transmissivity. With regard to the vegetation: different vegetation scenarios are analyzed (and this is fine). which is the real configuration? Which is the ultimate target of the simulations?*

The ultimate goal is to assess forest (management) scenarios on slope stability. The real configuration is the single tree detection method. This is emphasized in the introduction L124-128 and conclusion L594-596 of this new manuscript.

*I suggest to clearly state which is the ultimate main target of the model. Can we use it as forecast tool in an early warning system? If so, in which way? My impression is that it is too constrained to the calibration parameters, which, in some cases, may lose their physical meaning.*

The main application the authors intended to model for is as a tool to quantify the effects of different vegetation scenarios for land managers. We state this more clearly in the introduction L124-128 and conclusion L594-596 of this new manuscript.

*TECHNICAL CORRECTIONS*

Abstract: I strongly recommend to reduce the abstract to make it more concise.

We reduced the abstract to the best of our abilities.

L3: I do not completely agree with this sentence given that there are of works that take into account the effect of vegetation, although from different perspective such as the hydrological one, together with the mechanical one. Please remove this sentence from the abstract, where you do not have room to discuss.

The sentence has been removed.

L72-L80- I suggest to synthesize.

This part is vital for our assumption of macropore flow dominance and the 1-hour rainfall event. We tried to synthesize to the best of our abilities though.

Figure 1: it is useful and appropriate. However, consider to improve it to make it clearer. Not clear from where to start. "extract mean value for each landslide": do you mean hypothetical landslide? Emphasize the 'append' box where everything converges. Avoid text outside from the box. Also, I suggest to use the symbol used in the section (instead of the description). For example: definition of rho_ls; it would improve the correspondence with, for example, section 2.3.

Figure 1 has been improved along the suggestions of the reviewer.

Line 417: Not clear which single rainfall event you refer to. I understand that the database include landslide triggered by different storms across the years.

You are correct. We gave the wrong impression. We mean a single rainfall event per landslide. We corrected the text at line L457

Lines 378-379: it is not really clear the procedure. Please try to write more clearly.

We tried to write this more clearly L405-406

Lines 385-386: I understand the reference, but please give an explanation also here, based on your results.

This has been improved (L415-417)

Section 2.7: how did you define the threshold from daily to hourly??

By dividing by 24. This has been updated in L266.

**Response to Reviewer #2**

*The authors describe a probabilistic model called SlideforMap (SfM) which generates a map of shallow landslide probability across an area of interest. (..) The authors have organized their manuscript well, and they have described a complex workflow in a straightforward way. They also build a convincing case for the utility and need for a model of this type, and the described case studies illustrate the applications well. In my opinion, this manuscript should be published in NHESS after some clarifications and revisions. Most of my criticisms are focused on areas where the authors need to provide additional clarifications, either to adequately explain their approach or to explain how this model could be used by others.*

Thanks for the summary and the positive assessment. The revised paper contains more details on how the model could be used by others.

*SPECIFIC COMMENTS*

*1) The authors say that their model demonstrates the importance of root reinforcement on shallow landslides, but the authors need to define what "shallow" means so that it is clear where their conclusions apply.*

We stated the definition of shallow landslides to which SlideforMap applies in the introduction and conclusion.

*The authors are persuasive about the importance of root reinforcement in modeling landslide hazards, but they do not provide much discussion of how this model compares to other previously published models, including both related models (such as SOSlope or SlideForNET) or other models that compute landslide susceptibility on a regional scale. Some additional discussion of where this model fits within the context of other landslide susceptibility models generally would be helpful for prospective users.*

We added a paragraph explicitly comparing SlideforMap to other landslide susceptibility models in the discussion Section 5.6.

*In describing the methodology, the authors are not always clear about which values are assumed for their own case study, and which values are fixed in the model. For example, at a number of places in the methodology section, the authors assign values and limits on parameters (e.g., maximum HL surface area, mean tree density, precipitation intensity threshold, etc.) based on data from Switzerland (where the case study is located), but it is not clear whether a given user would have the freedom to change these values.*

Future users have the opportunity to change these values and are encouraged to do so if they apply SlideforMap in other areas. We made clear in the revised version which parameters are specifically selected for Switzerland (Table 1)

*The structure of the model requires that soil depth, soil cohesion, and the angle of internal friction be modeled as random variables with normal distributions, but the other 16 parameters are assumed to be deterministic. The authors need to explain why these three parameters specifically were chosen to be random variables. For instance, variables can be randomized when the uncertainty in their values is either shown or assumed to have the most significant effects on the results. This is suggested somewhat by the sensitivity analysis for the case of soil cohesion and soil depth, but this choice is not explained explicitly.*

To summarize, this choice was made because the soil depth, soil cohesion and friction angle vary in mountainous areas and are suspected to be sensitive parameters. We added this choice explicitly in the methodology (L186-189) on the soil parameters.

*The authors make use of two datasets, a tree inventory and a landslide inventory, in their analysis. However, they do not spend much time explaining how a prospective user would apply this model if they were lacking these datasets. It seems that users could still apply this model without these datasets, either by creating synthetic datasets or assuming specific values for the parameters that would be derived from these datasets. Providing some more guidance on applying the model without these datasets this would make the model more accessible to users.*

Synthetic parametrization is possible for users lacking certain data/datasets. We made this clearer in the methodology L181-184, 270, 300 of the paper.

*The sensitivity analysis is interesting but not entirely convincing. If strong parameter correlation is at play, as the authors suggest, then how would we know which parameters are truly important?*

What we intended to say in the original manuscript is that a *potential* correlation between parameters can lead to apparent absence of sensitivity. As an example, we used the paper by Bardossy (2007). We wrote this example more explicitly in the new manuscript L415-417.

*In a couple of places within the text (L49-52; L169-170) the authors conflate deterministic models with spatial homogeneity. This is misleading, as it is possible to have deterministic models that account for spatial heterogeneity, and probabilistic models that are spatially homogeneous. I would suggest that the explanation the authors are after is that the spatially heterogeneous values themselves are uncertain, and this is the motivation for using a probabilistic approach.*

We stated explicit definitions of both deterministic and probabilistic modelling in 40-50

*Is it valid to compare the globally uniform vegetation scenario to the other three scenarios if the globally uniform scenario was used to calibrate the parameters?*

We acknowledged this fact and use it to make our case for single tree detection in the discussion L520-521.

*It appears that the authors used the same landslide inventory to both calibrate the dataset and to validate the performance of the model against different vegetation scenarios. Did the authors consider using any portion of the landslide inventory as an independent validation dataset?*

We wanted to analyze the performance of the model, not do a validation. For a proper validation we think the size of the dataset is too limited. We made this clearer in the methodology L350-351.

*L44-45.  The authors need to give some additional definition of a deterministic approach and why SHALSTAB is an example of this approach.*

We gave a better description of models with similarities in the introduction L40-50 including a better definition of deterministic models.

*L128-130.  It seems that the unstable ratio is a very limited metric, particularly if the landslide density is already very low.  Shouldn't the landslide density be relevant in addition to the unstable ratio?  If there is an explicit requirement that the number of HLs be large enough to compute the unstable ratio with a large denominator, does this effectively put a lower bound on the landslide density for this model?*

We choose the AUC as main metric since it is a performance measure to the historical landslide inventory. We emphasized this choice in the methodology L393-394. We are not sure on the lower bound of landslide density. We stressed this in the discussion L576.

*L152-153.  I am surprised that the landslides are generated using a spatially uniform distribution, as this may result in landslides being simulated in areas that are not landslide prone.  What is the rationale behind this?  Shouldn't they follow a spatially distributed density, or at least be restricted to susceptible areas?*

In order for comparability of our results within the study area, between the study areas and with other model, we decided to keep the landslide density constant through a heterogeneous study area.

*L278.  A 2km buffer seems extremely large, especially if topographic wetness is computed over multiple small catchments. How was this value chosen, and is it adequate for other studies?*

The value is arbitrary. Model users are free to have a smaller buffer if they are confident it gives good results. We added this in the methodology L305.

*L407-408.  What does this mean if the unstable ratio decreases when single tree detection is used? Does this indicate that heterogeneity is important for slope stability, or does it simply mean that the uniform vegetation scenarios are not realistic?*

We added some possible explanations in the discussion section 5.4.

*Table 7.  Why are the AUC and Unstable ratio values different for the globally uniform vegetation scenario compared to the results with the optimal parameters (Table 6)?  Is this due to the difference in the landslide density?*

We emphasize in the results L444-445 that these are realizations from a probabilistic model.

*L471-473.  Does this high unstable ratio match with long term observations about landslide occurrence in StA?  In other words, is the unstable ratio realistic?*

We added a column in Table 2 of the landslide inventory derived shallow landslide density in the study area description and compared this in the discussion L540-542 with the unstable ratio.

*L480.  This suggests that AUC is a poor choice of performance metric for comparing the three study areas.  Are there other metrics which would be better?*

In order to compare the results in an easy manner to performance of other models, we decided to stick with the AUC.

TECHNICAL CORRECTIONS

L14. This should be "ratio" instead of "fraction."

Corrected.

L121. Does SfM generate a raster image of probability values?

Yes, we specified this in the methodology L131.

L134. Do the authors mean "greater than 1.0"?

Yes, corrected.

L163. What does "distance of 10" refer to?

Bin size of the histogram. This is specified in the new version L181.

L 271-273. What resolution is the unstable ratio computed at? This is not made explicit here in the paper.

The same resolution as the DEM input. This is specified in the methodology L277 of the new version.

L300-307. What is the spatial format of the landslide inventory? If they are polygons, how are they compared to the unstable ratio map so that the AUC can be computed? Does the landslide inventory need to be converted or rasterized at a specific resolution?

They are rasterized points. This is better specified in the methodology L336-337.

L309. The format for the numbers a,b, and c looks unusual. Please verify that the values and formats are correct.

We verified. They are corrected in our opinion, but we added units to the b and c parameters in accordance with the paper by Malamud et al., 2004.

L312-313. Why are these 11 parameters fixed while the others are varied?

We assume these parameters to be invariable and focus our sensitivity analysis on parameters that are variable and relevant in nature. We specify this in L343-344.

L332. What is n?

Total number of shallow landslides in the inventory with a known USCS soil class. We emphasized this in the revised version L366.

L336-337. Please explain why weighting is being used and how this weighting is calculated.

For representativeness. Weighted according to occurrence. We specified this more clearly L370.

L343. Please explain why the parameter range is using intensity values from different return periods.

We specified this more clearly in the preceding text L378-379.

Table 5. The value for vegetation weight, Wveg, uses a different name and different units than the rhotree in Table 1 (tonne per square meter vs. kg per cubic meter). Is there a reason for this difference?

The tree density (rho_tree in kg/m3) is used in conjunction with the single tree detection to compute a vegetation weight (in tonne per square meter). For the sensitivity analysis parameters in Table 5 we do not use the single tree detection and use the vegetation weight directly with a range of values from literature.

L348. Is 1000 an adequate size to represent the sample space over the 12 parameters used in the sensitivity analysis?

Unfortunately, due to computational constraints it is the best we could do. We specified this more clearly L387.

L360-361. Does this model assume that root reinforcement comes only from trees, and not from shrubs, grasses, or other vegetation types? Is the single-tree detection scenario using the same trees as the tree inventory cited in 3.2?

$1^{st}$ question: Yes, we added a reference to this assumption in the introduction L207-208.
$2^{nd}$ question: Yes, we specified this more clearly L400.

L363. Please verify that the exponent is correct in the expression for landslide density.

We verified and believe it is correct.

Fig. 8. How is "x% best" defined for the unstable ratio?

Best isn't the correct term, we correct this. Highest in terms of unstable ratio. We corrected this clearly in the paper figure 6.

L403. Do the model runs assume randomization of the three parameters (as in the original model setup)?

No parametrization of the 10 runs is identical. The drawn samples however can vary. We stated this more clearly L444-445.

L508. Are the 12 parameters all included in the 22 original parameters?

Yes, they are, we wrote it more clearly L586.

**Response to CC1**

*This is a really interesting paper that demonstrates the applicability and predictive capability of a new model for shallow landslides to provide a detailed inclusion of the influence of vegetation. The use of LiDAR data to deduce tree properties and thus root characteristics is a really exciting development.*

Thank you for the positive overall review.

*The model itself is similar to a number of existing models but also makes some important changes. It would be really useful to make these similarities and differences more explicit. The striking similarities to me were: 1) the hydrological model (Eqns 11-12) is exactly that of SHALSTAB (Montgomery and Dietrich, 1994) and SINMAP (Pack et al., 1998); 2) modelling discrete landslides of defined dimensions with lateral resistance due to roots only (Eqns 1-6) follows Montgomery et al. (2000), Schmidt et al. (2001) and Roering et al. (2003); 3) the probabilistic treatment of stability using distributions for parameters follows Pack et al. (1998) who represented c, phi and the R/T ratio as uniform distributions; 4) introducing a slope dependence to failure depth follows Prancevic et al. (2020), though with a different functional form. The similarities are strongest between SfM and Montgomery et al. (1998), they use very similar stability models (both infinite slope with root cohesion only on the margins), the same hydrological model, and both impose discrete landslide dimensions; so differentiating your work from theirs will be important.*

SlideforMap is similar in many aspects of the approaches. We added the specific similarities and distinctions to our method section in the revised paper L155, 190-194, 239-245 . A subsequent discussion was added in the discussion section 5.6.

*Having read the paper I have one primary outstanding question: What do you gain as a result of the additional data collection and modelling efforts involved in a detailed inclusion of the influence of vegetation?*

We expected an improvement in the performance of the model with the detailed inclusion of vegetation, which we analyzed in the paper. We emphasized better as a goal in the Introduction L124-128 and a discussion on the outcome in the discussion section 5.4.

*Your paper focuses on predictive skill (using ROC AUC) and predicted instability (using an unstable area ratio). That focus enables a straightforward assessment of improvement in predictive skill from this more complex model relative to a simpler models such as SHALSTAB or SINMAP. In fact, I think you already have an answer to this in Table 7. The 'no vegetation' case in SfM is very close to the SINMAP model: in this case, there is no lateral resistance (i.e. an infinite slope), probability of failure is calculated from pdfs of friction, cohesion and depth with pore pressure predicted using the SINMAP/SHALSTAB model. The uniform vegetation cases (Global and Forest area) are very close to the SHALSTAB implementation of Montgomery et al. (2000): in these cases landslides have predefined dimensions and lateral cohesion is spatially uniform. The difference is that landslide dimensions (area and depth), and material properties (c and phi) are sampled from distributions to generate a probability of failure rather than using the critical P/T as a metric for propensity to failure (as in SHALSTAB). In all these cases I would expect a direct comparison to SINMAP and the SHALSTAB of Montgomery et al. (2000) to yield almost exactly the same AUCs as those from SfM. The clear structural difference between SfM and previous models comes in the case of 'Single tree detection'.*

We added the similarities and distinctions between our model scenarios/metrics and the existing models to our methodology.

*Reading Table 7 in the context of these connections to simpler early models leads to three conclusions:*

- *Landslide predictions are surprisingly (and encouragingly) skilful even when models as simple as the 'No vegetation' SfM (equivalent to SINMAP) are used. Models like SINMAP are very attractive if they perform so well given their simple structure and parsimonious parameterisation.*
- *Representing landslides as discrete features (as in SfM or Montgomery et al. (2000)) rarely improves predictive skill unless detailed vegetation information is available. Best AUC for SfM with 'Global' or 'Forest area vegetation' are equal to the 'No vegetation' case for 2 of the 3 study sites and only 1% better for Sta.*
- *Detailed vegetation information from single tree detection does subtly improve predictive skill but only in 2 of the 3 sites (slightly worse for Eriz) and only by 3.8 and 3.2% in AUC for Trub and Sta respectively.*

*One interpretation of this would be that while SfM is much more satisfying from a process representation point of view it offers only very marginal gains in predictive skill and has considerable cost in that it is more highly parameterised and more complex. An alternative interpretation would be that small skill improvements on an already excellent model are worth the additional complexity (and cost). Reframing the percentage changes in AUC as percentage of the unrealised AUC that has been eroded by the new model (thus changing in denominator from $AUC_{pre}$ to $1-AUC_{pre}$) the same values are: 6% and 43% for Trub and Sta respectively. I think this interpretation, which recognises the diminishing returns in model improvement is reasonable and if so it suggests the improvement is non-trivial.*

*It is interesting that the unstable ratio metric is more sensitive to model structure than AUC, and perhaps encouraging that this ratio is reduced by improved process representation. However as you point out (L355), this ratio is a measure of instability rather than accuracy.*

The authors agree to a large degree with this interpretation and would like to thank CC for this interesting and concise discussion. We added this to our discussion section 5.4 on the vegetation scenarios and the model in general.

***SfM also makes predictions about the size of landslides** most likely to be triggered in each location (though these are not currently reported in the paper). This is an important difference from previous models. Few models have done this before and those that have are extremely computationally expensive. Therefore the most exciting aspect of SfM to me is its ability to predict landslide size. The authors are clear that the model requires a prior distribution of landslide sizes but this does not prevent SfM from producing useful information on landslide size both in global/lumped terms and spatially distributed terms. In lumped terms, you could compare the size distribution for triggered landslides with the prior distribution. In the current case the prior is the observed size distribution for the study area but you could equally impose a uniform prior and assess the extent to the posterior matches the observed approaches both would be informative. In spatially distributed terms, the pattern of landslide size and its relationship to local conditions would be interesting and you would also be able to assess model performance with respect to landslides size by comparing the areas of predicted landslides that overlap observed landslides and (do they correlate? What is the form of the relationship?). Perhaps this type of analysis is reserved for a later study but it would fit nicely in the current paper.*

We decided, as suggested by CC, to not include this (though interesting) analysis in this manuscript and save it for a later date.

*Beyond these three major points I have several other questions that are more specific but less important. I don't expect any of them to alter the primary messages of either the paper or the points I raise above but I hope they might be useful for the authors during revision. I do not understand the rationale behind some of the assumptions in SfM's boundary resistance representation*

- *Neglecting lateral earth pressure. It is true that active and passive earth pressure are maximised at some strain but neglecting them on this basis leaves two problems: a) you still need a treatment for the forces acting at the head and toe of the landslide; b) you need to apply the same criteria to root reinforcement since this is also maximised at some strain.*

In the revised version we specified the phase in the landslide we assume and better explained the resulting force balance in the methodology L152.

- *Neglecting soil cohesion on the sides. It seems inconsistent to apply root reinforcement but not soil cohesion on the lateral boundaries if you apply both on the base*

Like the answer above, we explained this better in the methodology.

- *Lateral root reinforcement acts only over the upslope half of the landslide's perimeter (Eqn 3). I don't see a justification for this and Schwarz et al., 2010 point out that it underestimates lateral reinforcement.*

Like the answer above, we explained this better in the methodology.

- *Lateral root reinforcement in Eqn. 9 is depth independent. This seems inconsistent with observed depth dependent rooting (density and size); and the depth dependence of basal reinforcement in SfM (Eqn 10).*

You are right. We integrated the RBM root probability density distribution and added this as a correction factor to the lateral root reinforcement. This is described in the methodology section 5, equation 8 and the results are recalculated.

- *Calculating root reinforcement using spatially averaged distance to trees within the Gamma function. Previous applications of the Gamma function (Eqn 9) appear to use it to predict root reinforcement at a known distance from the nearest tree (Moos et al., 2016). Given its nonlinearities, is it reasonable to use an average distance in Eqn 9 rather than evaluating Eqn 9 for the distribution of distances then averaging?*

For the current paper, we decided to keep this methodology as it is.

*Variability*

*The form amplitude and spatial pattern of variability in material properties are all likely important in defining landslide location and size (e.g. Bellugi et al., 2021). Representing this variability seems important. I would have liked to see more detail on your rationale for your choice of distribution form and spatial (de)correlation. I recognise that observations to inform this are sparse and these properties are not well known. The normal distribution has some specific problems that you grapple with but that others chose to avoid by using a log-normal (e.g. Griffiths et al., 2007). You deal with*

*unphysical negative values by truncating, and claim these are rare but this places strict constraints on the variability that you can impose (small coefficients of variation for soil depth and cohesion in Table 5). In the absence of evidence to the contrary, a distribution that is limited to positive values (e.g. lognormal) would seem a more appropriate choice.*

We agreed that the log-normal distribution is a more appropriate choice. We applied this in the paper, section 2.4, in a recomputation of the results. In addition, we added a comparative figure in the appendix.

*Soil depth variability is treated slightly differently (spatially de-correlated but slope dependent). I was unsure whether soil depths distribution was parameterised from observed landslide scar depths (L178) or using mean and standard deviation as parameters to optimise (Figure 7). The former seems problematic: landslides likely occur in deeper soils biasing the sample. Perhaps Eqn 7 was designed to account for this? However, I don't understand why the coefficients on mu (1.35) and sigma1 (0.75) in Eqn 7 have these particular values. The second approach, tuning mean depth rather than setting it from observations seems more appealing to me and would also enable a comparison between model results and observed landslide depths, which would be a nice addition.*

We adjusted, pointing out that this tuning is optional L203-204.

**Hydrology**

*Your approach is exactly the same as that of SHALSTAB and SINMAP but is considerably different from Topmodel (Beven and Kirkby, 1979). All three use a topographic index to define hydrologically similar units. Topmodel uses these (with simple treatments for evaporation and infiltration) to simulate a time-varying catchment averaged response to a rainfall timeseries that can be mapped back onto the HSUs; the others simply solve for a single steady recharge rate (neglecting these processes). Even the topographic index (i.e. $A/sin(B)$) differs from that of Topmodel (which uses $ln(A/tan(B))$). This reflects differences in reference frame (the sin vs tan) and assumed conductivity profile (uniform vs exponential). I don't disagree with the approach but I think it follows Montgomery and Dietrich (1994) and Pack et al. (1998) so it would be simpler to say that. If you wanted to give credit to earlier work then the TOPOG model of O'loughlin (1986) was behind the original derivation of SHALSTAB and the first introduction of a topographic index was by Kirkby (1975).*

We let go of the formulation of using TOPmodel or TOPmodel assumptions and gave explicit credit to O'loughlin (1986) and Kirkby (1975) in the methodology L239-240.

*Previous papers that apply this hydrological model do not claim that it is particularly well suited to slopes with macropore flow. Montgomery et al. (2002) highlight the importance of macropores and fractures (and a steep soil water characteristic curve) for hillslope hydrologic response but also recognise that "that rapid pore pressure response that controls slope instability […] is driven by vertical flow, not lateral flow" (Montgomery et al., 2004). **There is general agreement that lateral flow (modelled here) strongly influences the pore pressure field antecedent to a burst of rain that could initiate a landslide** (Iverson, 2000; Montgomery et al., 2002; 2004). This has important implications for the approach though because it implies that Q/T is an index for the 'propensity for landsliding' rather than a parameter to be calibrated within a complete hydrological treatment. This*

*explains the apparent problem of predicted pore pressures independent of rainfall duration but observations that landslide triggering depends on both intensity and duration. Broad spatial patterns of pore pressure and instability should be well captured but triggering rainfall properties may not be. In fact, discussion of the influence of macropores on pore pressure tends to focus on the unpredictable localised pressure peaks associated with constrictions or terminations to macropores (e.g. Pierson, 1983; Montgomery et al. 2002). Even given these limitations I don't think this is a bad model relative to the alternatives because it captures broad phreatic surface patterns and I'm convinced that the finer detail of these patterns is set by (unknown and perhaps unknowable) heterogeneity in material properties (e.g. macropores). If so, a more refined and expensive hydrological model may improve predictions of spatial pore pressure patterns very little.*

We updated the introduction L84-86 and methodology section 2.6 stating our assumptions, limitations and similarities to our model more explicitly.

**Sensitivity Analysis**

*As you point out parameter interaction makes it very difficult to infer parameter sensitivity from Figure 7 I think that may make it difficult to support some of your assertions in L388-395 because you cannot guarantee that interactions are not masking other stronger sensitivities. For me the clearest example is the interaction between P and T (Table 6). Both are listed as uncertain parameters within the sensitivity analysis but only feature in pore pressure definition and only in that equation as the P/T ratio. As a result their inclusion as two separate variables in this analysis is likely to lead to severe equifinality (with high or low values will result in the same outcome as long as P/T is constant). Why not include the ratio of the two in your sensitivity analysis?*

We added P/T ratio to our sensitivity analysis results and discussed the equifinality L546-547 that appears to be at play.

**Queries on equations:**

*1) I think there is a dimensional problem in either the first term of Eqn 3 or the second term of Eqn 4. Eqn 10 expresses $R_{bas}$ as a function of $R_{lat}$ so I think both should be either a force per unit length or a stress. If $R_{lat}$ (in Eqn 9) is a stress then Eqn3 is dimensionally incorrect because the first term is a force per unit length and the second a force. The first term needs integrating over landslide depth. This could take the form cos(s) H if you assume reinforcement is depth invariant. However, this would then be inconsistent with Eqn10, which assumes that root reinforcement declines with depth. On the other hand, if $R_{lat}$ is a force per unit length (which might be more consistent with Moos et al (2016), Fig 3) then the problem may be more difficult to solve because the lateral depth integrated stress (N/m) is being applied across a basal area ($m^2$).*

Indeed, it uses a dimension correction factor, we added this to Equation 9.

*2) Are h and H measured in a vertical reference frame as indicated in Figure 2? If so then I think there is a cos(s) missing from Eqn 12. The first cos(s) converts vertical depth to slope normal thickness, the second converts phreatic surface thickness to pressure head (under assumptions of: uniform steady slope parallel seepage).*

You are right, we corrected this and recalculated.

*3) Eqn 15 is incorrect because the  original equation calculates DBH in cm from tree height in metres (Dorren, 2017) but you use DBH in metres (L292). I think Eqn 15 should  be adjusted to $0.01H^{1.25}$.*

You are right, we corrected this.

---

## Referee Report (RR1)

**Review of van Zadelhoff et al. 2021 for NHESS by David Milledge**

**I made a community comment on the first version of this manuscript and was then asked by the Associate Editor to review the revised paper. The revisions have improved an already very interesting paper, which demonstrates the applicability and predictive capability of a new model for shallow landslides to provide a detailed inclusion of the influence of vegetation. I have responded in bold to the authors' responses below. I think my additional comments or queries are only fairly minor and should require only fairly minor alterations to the paper. The only exception to this the updated Equations 8 and 9, where I think there is an inconsistency in the way basal and lateral root cohesion are being calculated.**

*The model itself is similar to a number of existing models but also makes some important changes. It would be really useful to make these similarities and differences more explicit. The striking similarities to me were: 1) the hydrological model (Eqns 11-12) is exactly that of SHALSTAB (Montgomery and Dietrich, 1994) and SINMAP (Pack et al., 1998); 2) modelling discrete landslides of defined dimensions with lateral resistance due to roots only (Eqns 1-6) follows Montgomery et al. (2000), Schmidt et al. (2001) and Roering et al. (2003); 3) the probabilistic treatment of stability using distributions for parameters follows Pack et al. (1998) who represented c, phi and the R/T ratio as uniform distributions; 4) introducing a slope dependence to failure depth follows Prancevic et al. (2020), though with a different functional form. The similarities are strongest between SfM and Montgomery et al. (1998), they use very similar stability models (both infinite slope with root cohesion only on the margins), the same hydrological model, and both impose discrete landslide dimensions; so differentiating your work from theirs will be important.*
SlideforMap is similar in many aspects of the approaches. We added the specific similarities and distinctions to our method section in the revised paper L155, 190-194, 239-245 . A subsequent discussion was added in the discussion section 5.6.

**The comparison to other models in Section 5.6 is very useful. However, I think there are a few other models that capable of catchment scale application and that resolve lateral root reinforcement which are worth discussing. Montgomery et al., (2000) is particularly important to discuss in Section 5.6 because they use your hydrological model and a very similar geotechnical model (infinite slope with roots the only lateral reinforcement). Other important models to consider are: Hess et al. (2017), who also assume fixed dimensions for the landslides but include a treatment for boundary friction and passive resistance (which you neglect); Cislaghi et al (2017, 2018) sample landslide sizes from a distribution (similar to your approach); von Ruette et al. (2013) and Bellugi et al. (2015) predict size in the model rather than imposing a single size or distribution of sizes. I suspect that both of the latter are fairly slow to run relative to your approach and you certainly improve on the approaches that define a single landslide size. You should perhaps comment on the ways in which your approach differs from that of Cislaghi et al. (2017, 2018). It would be useful to say a little more about the respects in which SfM "uses a more realistic implementation of root reinforcement" and how this differs from these other models. If the main difference is that you neglect compressive resistance this may need stronger justification either here or in the methods section. Do any other studies predict landslide locations accounting for the spatial distribution of root reinforcement as a function of forest structure? We can argue about the specifics of the stability model but this for me is probably the most novel aspect of the work.**

**I also had a couple of other minor comments on the new text:**
**L113-4: "displacement independent": I think your approach also models displacement-independent reinforcement. If you make the point that this is a limitation of previous approaches up here, you should probably add in your comparison to other stability models that SfM uses displacement independent reinforcement whereas Cohen and Schwarz (2017) and to some degree von Ruette et al. (2013) account for displacement in their treatments.**

**L155: "second stage of the activation phase" this isn't explicitly referred to in the introduction so it is not clear what you are pointing back to here. The sentence that follows ("This coincides with…") is very helpful and it might be easier to understand this paragraph if you simply said something like: "…typical for the second stage of the activation phase: the displacement at which lateral root reinforcement is maximised under tension along the tension crack and at which passive earth pressure, lateral root compression and lateral soil cohesion no longer act."**

*Having read the paper I have one primary outstanding question: What do you gain as a result of the additional data collection and modelling efforts involved in a detailed inclusion of the influence of vegetation?*

We expected an improvement in the performance of the model with the detailed inclusion of vegetation, which we analyzed in the paper.

**This sentence is great! It captures your hypothesis for the paper really nicely and would be worth including around L124 because I don't see such a clear statement of your expectation in the current paper.**

We emphasized better as a goal in the Introduction L124-128 and a discussion on the outcome in the discussion section 5.4.

**This additional text is useful.**

*Your paper focuses on predictive skill (using ROC AUC) and predicted instability (using an unstable area ratio). That focus enables a straightforward assessment of improvement in predictive skill from this more complex model relative to a simpler models such as SHALSTAB or SINMAP. In fact, I think you already have an answer to this in Table 7. The 'no vegetation' case in SfM is very close to the SINMAP model: in this case, there is no lateral resistance (i.e. an infinite slope), probability of failure is calculated from pdfs of friction, cohesion and depth with pore pressure predicted using the SINMAP/SHALSTAB model. The uniform vegetation cases (Global and Forest area) are very close to the SHALSTAB implementation of Montgomery et al. (2000): in these cases landslides have predefined dimensions and lateral cohesion is spatially uniform. The difference is that landslide dimensions (area and depth), and material properties (c and phi) are sampled from distributions to generate a probability of failure rather than using the critical P/T as a metric for propensity to failure (as in SHALSTAB). In all these cases I would expect a direct comparison to SINMAP and the SHALSTAB of Montgomery et al. (2000) to yield almost exactly the same AUCs as those from SfM. The clear structural difference between SfM and previous models comes in the case of 'Single tree detection'.*

We added the similarities and distinctions between our model scenarios/metrics and the existing models to our methodology.

**I think it would be useful in the discussion of similarities to other models (Section 5.4) to explain that your 'no vegetation' case differs from SINMAP only in: 1) the form of the distributions that you use to sample c and phi (log-normal for you, uniform for SINMAP); and 2) the treatment of soil depth, which is spatially uniform in SINMAP and spatially variable for you. This means that the AUC values for 'no vegetation' are indicative of those that you would expect from SINMAP. I would have really liked to see a comparison to SHALSTAB in terms of AUC because it is so simple and so widely used. But it certainly isn't essential that the authors choose to do that.**

*Reading Table 7 in the context of these connections to simpler early models leads to three conclusions:*
* *Landslide predictions are surprisingly (and encouragingly) skilful even when models as simple as the 'No vegetation' SfM (equivalent to SINMAP) are used. Models like SINMAP are very attractive if they perform so well given their simple structure and parsimonious parameterisation.*
* *Representing landslides as discrete features (as in SfM or Montgomery et al. (2000)) rarely improves predictive skill unless detailed vegetation information is available. Best AUC for SfM with 'Global' or 'Forest area vegetation' are equal to the 'No vegetation' case for 2 of the 3 study sites and only 1% better for Sta.*
* *Detailed vegetation information from single tree detection does subtly improve predictive skill but only in 2 of the 3 sites (slightly worse for Eriz) and only by 3.8 and 3.2% in AUC for Trub and Sta respectively.*

*One interpretation of this would be that while SfM is much more satisfying from a process representation point of view it offers only very marginal gains in predictive skill and has considerable cost in that it is more highly parameterised and more complex. An alternative interpretation would be that small skill improvements on an already excellent model are worth the additional complexity (and cost). Reframing the percentage changes in AUC as percentage of the unrealised AUC that has been eroded by the new model (thus changing in denominator from $AUC_{pre}$ to $1-AUC_{pre}$) the same values are: 6% and 43% for Trub and Sta respectively. I think this interpretation, which recognises the diminishing returns in model improvement is reasonable and if so it suggests the improvement is non-trivial.*

*It is interesting that the unstable ratio metric is more sensitive to model structure than AUC, and perhaps encouraging that this ratio is reduced by improved process representation. However as you point out (L355), this ratio is a measure of instability rather than accuracy.*

The authors agree to a large degree with this interpretation and would like to thank CC for this interesting and concise discussion. We added this to our discussion section 5.4 on the vegetation scenarios and the model in general.

**I don't see where the first two bullets above have been incorporated into Section 5.4, though I do see where the third bullet has been included and where some of the interpretation might have influenced your revised text. I still think those first two points are worth making (even if the 'no veg' case is not directly equivalent to SINMAP) because they capture the move from infinite to finite slope stability modelling, which has been a subject of debate**

in the literature. I have one final suggestion in relation to Table 7, since you know mean, standard deviation and sample size you should be able to estimate the significance of differences in AUCs between different approaches. A quick attempt at this for the overall performances using a t-test (which may not be the best approach) suggests that single tree detection is significantly better than no veg and Forest uniform veg at 99% and 90% confidence respectively; but is not significantly different from Global uniform veg. The Global uniform veg is significantly better than the no veg case but the Forest uniform veg is not, nor is there any significant difference between Global and Forest uniform veg cases.

*SfM also makes predictions about the size of landslides.* We decided, as suggested by CC, to not include this (though interesting) analysis in this manuscript and save it for a later date.
**This is fine by me.**

*Beyond these three major points I have several other questions that are more specific but less important. I don't expect any of them to alter the primary messages of either the paper or the points I raise above but I hope they might be useful for the authors during revision. I do not understand the rationale behind some of the assumptions in SfM's boundary resistance representation*
• *Neglecting lateral earth pressure. It is true that active and passive earth pressure are maximised at some strain but neglecting them on this basis leaves two problems: a) you still need a treatment for the forces acting at the head and toe of the landslide; b) you need to apply the same criteria to root reinforcement since this is also maximised at some strain.*
In the revised version we specified the phase in the landslide we assume and better explained the resulting force balance in the methodology L152.
**This new text is much clearer. It would be helpful to put a range of values to the 'minor movement' that you have in mind (I guess this is on the order of 10-100 mm?). If I understand L153-54 correctly you argue that beyond this displacement there is no passive earth pressure, lateral root compression or lateral soil cohesion. I think each of these claims needs a citation. How do you reconcile that with your earlier findings that passive resistance remains important even for displacements > 300 mm (Schwarz et al., 2015; Cohen and Schwarz, 2017)? Alternatively, if your reason for neglecting some of these components is that we don't yet understand them well enough to represent them satisfactorily then I think it is fine to simply say that here.**

• *Neglecting soil cohesion on the sides. It seems inconsistent to apply root reinforcement but not soil cohesion on the lateral boundaries if you apply both on the base*
Like the answer above, we explained this better in the methodology.
**This is much clearer in the new text. I still don't understand why lateral soil cohesion would go to zero at some displacement but basal cohesion would be unaffected, unless you are assuming that a tension crack opens up along the entire upper half of the landslide. In that case, it might be worth adding a comment that explains that you neglect all resistance in the compression zone and assume that a tension crack opens along the entire length of the tension zone such that lateral resistance in this zone is only due to root reinforcement.**

• *Lateral root reinforcement acts only over the upslope half of the landslide's perimeter (Eqn 3). I don't see a justification for this and Schwarz et al., 2010 point out that it underestimates lateral reinforcement.*
Like the answer above, we explained this better in the methodology.
**This is a little clearer in that it explicitly explains what you do but you still don't explain why you do it. As above, I think adding a sentence that explains that: you neglect all resistance in the compression zone and assume that a tension crack opens along the entire length of the tension zone such that lateral resistance in this zone is only due to root reinforcement. I don't necessarily agree that this is the best approach, I would do it differently but this isn't my model.**

• *Lateral root reinforcement in Eqn. 9 is depth independent. This seems inconsistent with observed depth dependent rooting (density and size); and the depth dependence of basal reinforcement in SfM (Eqn 10).*
You are right. We integrated the RBM root probability density distribution and added this as a correction factor to the lateral root reinforcement. This is described in the methodology section 5, equation 8 and the results are recalculated.
**This sounds great. However, I'm still a little confused. Perhaps out of ignorance around Gamma distributions. I have two questions: 1) is this an analytical integration? If so can you write it out in Equation 8? 2) if the parameters should be selected to ensure that the function decreases with increasing Hsoil does that require alpha=1, in which case the function becomes an exponential and everything gets simpler? 3) is the second gamma**

density function in equation 8 the same as the gamma density function in equation 9? The notation in the equations and the definition in the text suggests not but I haven't understood the distinction between the two.

Looking at equation 8 again, I think there may still be an error in the way that you relate basal and lateral reinforcement. The best way I can illustrate that is to set the alpha parameter to unity in the gamma distribution so that it reduces to an exponential distribution. In that case I can rewrite equation 8 as:

$$R_l = c\,\Gamma\left(\frac{D_{trees}}{DBH\,D_{treesmax}}\middle|\alpha_1,\beta_1\right)\int_0^H \Gamma(h|\alpha_2,\beta_2)\,dh$$

Applying alpha=1 and substituting a constant C0 for the first term (since this is depth invariant) gives:

$$R_l = C_0 \int_0^H \Gamma(h|1,\beta_2)\,dh$$

In this case the gamma density function simplifies to an exponential and the equation can be re-written as:

$$R_l = C_0\,\beta_2 \int_0^H e^{-h\beta_2}\,dh = C_0\,\beta_2\left(\frac{1-e^{-H\beta_2}}{\beta_2}\right) = C_0\left(1-e^{-H\beta_2}\right)$$

If root density follows this (exponential distribution) then it is clear that basal cohesion (assuming isotropic rooting and strength) is:

$$R_b = C_0\,\beta_2\,e^{-H\beta_2}$$

In this case the dimensions also work out, without the need for the dimensional correction coefficient k (though k takes a value of unity and can be ignored in calculations). However, applying equation 10 in this case results in:

$$R_b = k\,R_l\,\Gamma(H|1,\beta_2) = R_l\,\beta_2\,e^{-H\beta_2} = C_0\,\beta_2\left(1-e^{-H\beta_2}\right)e^{-H\beta_2}$$

This extra term has a very large influence at small depths, pulling Rb down to zero at the surface, which is not compatible with a gamma root density distribution with alpha =1 (i.e an exponential root density distribution). The differences between the two formulations differ by <20% when depth exceeds 0.5 m and by <5% when depth exceeds 1 m (using beta=3.2 as you do in the paper). Therefore this error (if you agree that it is an error) is unlikely to result in large changes to the stability in the areas of deeper soil where landslides typically occur but should make other parts of the catchment where soils are shallow considerably more stable.

*• Calculating root reinforcement using spatially averaged distance to trees within the Gamma function. Previous applications of the Gamma function (Eqn 9) appear to use it to predict root reinforcement at a known distance from the nearest tree (Moos et al., 2016). Given its nonlinearities, is it reasonable to use an average distance in Eqn 9 rather than evaluating Eqn 9 for the distribution of distances then averaging?*
For the current paper, we decided to keep this methodology as it is.
**I think that is reasonable, but I think you should say that explicitly in the paper and explain why you make that choice (e.g. because it is prohibitively expensive to evaluate 9 for the distribution of distances).**

*Variability*
*The form amplitude and spatial pattern of variability in material properties are all likely important in defining landslide location and size (e.g. Bellugi et al., 2021). Representing this variability seems important. I would have liked to see more detail on your rationale for your choice of distribution form and spatial (de)correlation. I recognise that observations to inform this are sparse and these properties are not well known. The normal distribution has some specific problems that you grapple with but that others chose to avoid by using a log-normal (e.g. Griffiths et al., 2007).*
We agreed that the log-normal distribution is a more appropriate choice. We applied this in the paper, section 2.4, in a recomputation of the results. In addition, we added a comparative figure in the appendix.
**Addressed. The only remaining point here is that I don't see where you define the symbols you use to represent these parameters in Table 6. It would also be worth explaining that the parameters of the log-normal are mean and standard deviation of log transformed data. I guess you then transform them back for table 6?**

*Soil depth variability is treated slightly differently (spatially de-correlated but slope dependent). I was unsure whether soil depths distribution was parameterised from observed landslide scar depths (L178) or using mean and standard deviation as parameters to optimise (Figure 7). The former seems problematic: landslides likely occur in deeper soils biasing the sample. Perhaps Eqn 7 was designed to account for this? However, I don't understand why the coefficients on mu (1.35) and sigma1 (0.75) in Eqn 7 have these particular values. The second approach, tuning mean depth rather than setting it from observations seems more appealing to me and would also enable a comparison between model results and observed landslide depths, which would be a nice addition.*
We adjusted, pointing out that this tuning is optional L203-204.

**This is now much clearer. I have just a few remaining minor suggestions / queries:**

**L197: "definitive values for soil thickness": I'm not sure what you mean by definitive in this context.**

**L198: It would help to clarify what sets the "initial thickness" if you added "which is sampled from a log-normal distribution".**

**L202: Why do the coefficients used to find mu1 and sigma1 from mu_h and sigma_h take these particular values (1.35 and 0.75 respectively)? Did you establish them by trial and error based on your perceptual model for how soil depth varies with slope in your study areas?**

*Hydrology*

*Your approach is exactly the same as that of SHALSTAB and SINMAP but is considerably different from Topmodel (Beven and Kirkby, 1979). All three use a topographic index to define hydrologically similar units. Topmodel uses these (with simple treatments for evaporation and infiltration) to simulate a time-varying catchment averaged response to a rainfall timeseries that can be mapped back onto the HSUs; the others simply solve for a single steady recharge rate (neglecting these processes). Even the topographic index (i.e. A/sin(B)) differs from that of Topmodel (which uses ln(A/tan(B))). This reflects differences in reference frame (the sin vs tan) and assumed conductivity profile (uniform vs exponential). I don't disagree with the approach but I think it follows Montgomery and Dietrich (1994) and Pack et al. (1998) so it would be simpler to say that. If you wanted to give credit to earlier work then the TOPOG model of O'loughlin (1986) was behind the original derivation of SHALSTAB and the first introduction of a topographic index was by Kirkby (1975).*

We let go of the formulation of using TOPmodel or TOPmodel assumptions and gave explicit credit to O'loughlin (1986) and Kirkby (1975) in the methodology L239-240.

**Addressed, my only minor comments would be:**

1) **L239: TOPOG should not be cited as Montgomery and Dietrich (1994), because they didn't develop it, but should instead cite O'loughlin (1986).**

2) **L241: Pack et al. (1998) is the incorrect citation for SHALSTAB, this should be Montgomery and Dietrich (1994).**

3) **I agree though that Pack et al. (1998) should be cited here because the identical model is used in SHALSTAB and SINMAP and the latter is Pack et al.'s model.**

4) **L478: 'application of the TOPOG/TOPMODEL approach'. I suggest you cut reference to TOPMODEL here because you aren't using a TOPMODEL approach.**

*Previous papers that apply this hydrological model do not claim that it is particularly well suited to slopes with macropore flow. Montgomery et al. (2002) highlight the importance of macropores and fractures (and a steep soil water characteristic curve) for hillslope hydrologic response but also recognise that "that rapid pore pressure response that controls slope instability [...] is driven by vertical flow, not lateral flow" (Montgomery et al., 2004). **There is general agreement that lateral flow (modelled here) strongly influences the pore pressure field antecedent to a burst of rain that could initiate a landslide** (Iverson, 2000; Montgomery et al., 2002; 2004). This has important implications for the approach though because it implies that Q/T is an index for the 'propensity for landsliding' rather than a parameter to be calibrated within a complete hydrological treatment. This explains the apparent problem of predicted pore pressures independent of rainfall duration but observations that landslide triggering depends on both intensity and duration. Broad spatial patterns of pore pressure and instability should be well captured but triggering rainfall properties may not be. In fact, discussion of the influence of macropores on pore pressure tends to focus on the unpredictable localised pressure peaks associated with constrictions or terminations to macropores (e.g. Pierson, 1983; Montgomery et al. 2002). Even given these limitations I don't think this is a bad model relative to the alternatives because it captures broad phreatic surface patterns and I'm convinced that the finer detail of these patterns is set by (unknown and perhaps unknowable) heterogeneity in material properties (e.g. macropores). If so, a more refined and expensive hydrological model may improve predictions of spatial pore pressure patterns very little.*

We updated the introduction L84-86 and methodology section 2.6 stating our assumptions, limitations and similarities to our model more explicitly.

**This is partly addressed in that the assumptions and limitations are now clearer and the connection to macro-pore flow has been clarified. I have two outstanding comments though:**

**L255: "Based on the literature data discussed in the introduction": I think the literature you refer to needs citing again here, ideally with a sentence that explains the basis for your claim that the time to equilibrium is 1 hour. Li et al. (2013) aren't much help here because they are talking about the time to equilibrium for vertical infiltration only, there is no lateral flow in their model as I understand it. Montgomery et al. (2002, 2004) and Iverson (2000) are counter examples because while they disagree on many things, they agree that the equilibrium time even in**

the conductive Coos Bay soil should be much longer than 1 hour. As I said above, I don't think that makes the model inappropriate because (lateral flow sets the antecedent conditions for the triggering burst). But it does mean that R/T should be treated as an index for the propensity of landsliding rather than a quantity that can be directly compared to observed rainfall.

If you disagree, and instead think hillslope lateral response times in your study areas are fast enough to allow direct connection to observed rainfall then a sentence or two recognising the debate around these response times would probably be sufficient to make readers aware that you differ with others on this point.

*Sensitivity Analysis*
*As you point out parameter interaction makes it very difficult to infer parameter sensitivity from Figure 7 I think that may make it difficult to support some of your assertions in L388-395 because you cannot guarantee that interactions are not masking other stronger sensitivities. For me the clearest example is the interaction between P and T (Table 6). Both are listed as uncertain parameters within the sensitivity analysis but only feature in pore pressure definition and only in that equation as the P/T ratio. As a result their inclusion as two separate variables in this analysis is likely to lead to severe equifinality (with high or low values will result in the same outcome as long as P/T is constant). Why not include the ratio of the two in your sensitivity analysis?*

We added P/T ratio to our sensitivity analysis results and discussed the equifinality L546-547 that appears to be at play.

**Addressed**

*Queries on equations:*
*1) I think there is a dimensional problem in either the first term of Eqn 3 or the second term of Eqn 4. Eqn 10 expresses $R_{bas}$ as a function of $R_{lat}$ so I think both should be either a force per unit length or a stress. If $R_{lat}$ (in Eqn 9) is a stress then Eqn3 is dimensionally incorrect because the first term is a force per unit length and the second a force. The first term needs integrating over landslide depth. This could take the form cos(s) H if you assume reinforcement is depth invariant. However, this would then be inconsistent with Eqn10, which assumes that root reinforcement declines with depth. On the other hand, if $R_{lat}$ is a force per unit length (which might be more consistent with Moos et al (2016), Fig 3) then the problem may be more difficult to solve because the lateral depth integrated stress (N/m) is being applied across a basal area ($m_2$).*

Indeed, it uses a dimension correction factor, we added this to Equation 9.

**Addressed, though see my earlier comment about a potential problem with Equation 9. If you took my alternative approach then the dimensional mismatch would disappear because Eqn 8 would be integrated over depth while Eqn 9 would be calculated directly from the depth decay function (whether gamma or exponential).**

*2) Are h and H measured in a vertical reference frame as indicated in Figure 2? If so then I think there is a cos(s) missing from Eqn 12. The first cos(s) converts vertical depth to slope normal thickness, the second converts phreatic surface thickness to pressure head (under assumptions of: uniform steady slope parallel seepage).*

You are right, we corrected this and recalculated. Page 15 of 15

**Addressed**

*3) Eqn 15 is incorrect because the original equation calculates DBH in cm from tree height in metres (Dorren, 2017) but you use DBH in metres (L292). I think Eqn 15 should be adjusted to $0.01H_{1.25}$.*

You are right, we corrected this.

**Adjusting the units in Eqn 14 fixes the problem there but leads to ambiguity earlier in the paper where you define is one outstanding problem here in that you define DBH earlier in the paper (L214) then use it as a term in equation 8 is DBH also being measured in cm here or is a conversion required? I think it would be better to adjust Eqn 14 so that DBH is expressed in metres there in order to avoid confusion elsewhere.**

**References**
Bellugi, D., Milledge, D.G., Dietrich, W.E., Perron, J.T. and McKean, J., 2015. Predicting shallow landslide size and location across a natural landscape: Application of a spectral clustering search algorithm. *Journal of Geophysical Research: Earth Surface*, *120*(12), pp.2552-2585.
Cislaghi, A., Chiaradia, E.A. and Bischetti, G.B., 2017. Including root reinforcement variability in a probabilistic 3D stability model. *Earth Surface Processes and Landforms*, *42*(12), pp.1789-1806.
Cislaghi, A., Rigon, E., Lenzi, M.A. and Bischetti, G.B., 2018. A probabilistic multidimensional approach to quantify large wood recruitment from hillslopes in mountainous-forested catchments. *Geomorphology*, *306*, pp.108-127.

Hess, D.M., Leshchinsky, B.A., Bunn, M., Mason, H.B. and Olsen, M.J., 2017. A simplified three-dimensional shallow landslide susceptibility framework considering topography and seismicity. *Landslides*, *14*(5), pp.1677-1697.

Montgomery, D.R., Schmidt, K.M., Greenberg, H.M. and Dietrich, W.E., 2000. Forest clearing and regional landsliding. *Geology*, *28*(4), pp.311-314.

von Ruette, J., Lehmann, P. and Or, D., 2013. Rainfall-triggered shallow landslides at catchment scale: Threshold mechanics-based modeling for abruptness and localization. *Water Resources Research*, *49*(10), pp.6266-6285.

---

## Referee Report (RR2)

*Review of van Zadelhoff et al. for NHESS by David Milledge*

Thank you to the authors for taking the time to respond to my comments, almost all of which have been addressed. However, I have one major outstanding concern. I think that the equations for basal and lateral reinforcement are still incorrect. It is essential that the authors address this prior to publication. I also have three, fairly minor concerns that can be very easily addressed:

1) The need for a clearer justification for the choice to neglect passive resistance;
2) Some statements in the paper have not been updated to reflect new results and are now incorrect;
3) Discussion of similarities between your 'no vegetation' parameterisation of your model and SINMAP.

My other concerns from the previous review have been addressed. Each of the remaining concerns has some history in previous reviews I have tried to summarise my concern and the history from previous reviews below.

**Major Comment: Representing basal root reinforcement as a function of lateral reinforcement is incorrect**

It is now clear that you use a Gamma pdf and the additional equation is useful, you clarified in your response that you used numerical integration but should update the text to reflect this. However, my main original concern has not been addressed. Eqn10 calculates basal reinforcement as a function of lateral reinforcement. That is incorrect conceptually because if lateral reinforcement is a depth integration of reinforcement per unit area then basal reinforcement should simply be reinforcement per unit area at the basal depth. Your response to my previous comment suggests that you agree (you said *"The equation in essence is the same in both equation 8 and 9. Equation 8 however, applies the cumulative value (soil depth in range [0,x]), equation 9 applies the point value soil depth at [x]"*). However, your implementation is at odds with this. Eqn 4 in Gehring et al. (2019) is much more consistent with both your response and my expectation because they use of $R_{max}$ rather than $R_{lat}$. Is it possible there is an error in Eqn 10 in the paper and it should be using $R_{max}$?

In addition to the conceptual problem, your approach introduces two errors:

1) The dimensional error that you have to correct with the unit coefficient k.
2) A relationship between basal reinforcement and depth that is at odds with observations of root density and reinforcement. Your group's previous observations (e.g. Schwarz et al., 2012) and those of other groups all suggest that reinforcement declines with depth, or at least does not increase. Your equations ensure that basal root reinforcement increases with depth. The graph below was generated using your gammaPDF parameters with an $R_{max}$ ($R_{max} = c \, \Gamma \left( \frac{D_{trees}}{DBH \, D_{treesmax}} | \alpha_1, \beta_1 \right)$) of 1 kN/m2 (to keep things simple).

I'm convinced that this representation is incorrect and that it is potentially important because in shallow soils, where basal reinforcement is particularly important, it will result in a very large underestimation of reinforcement.

[Figure]

My second concern having plotted the graph above is that the parameters of your gammaPDF implemented with Gehring et al.'s (2019) parameters implies that root density initially increases rapidly with depth while your own observations of the vertical distribution of roots in Switzerland and many observations from other groups all suggest that root biomass (and thus reinforcement) decays with depth, usually exponentially (see Dazio et al., 2018; Schwarz et al., 2012 and references therein: Abe and Iwamoto, 1990, Abe and Ziemer, 1991, Abernethy and Rutherfurd, 2001, Schmidt et al., 2001, Schenk and Jackson, 2002, Bischetti et al., 2005, Laio et al., 2006, Docker and Hubble, 2009). This suggests that your alpha2 and beta2 parameters are not physically reasonable.

Do you agree that gammaPDF values should be lower at 2 m depth than they are at 0.2 m depth?

A minor point, Table 1 would be improved by replacing 'literature' entries with the relevant references.

**Minor Comment 1: The need for a clearer justification for your choice to neglect passive resistance.**
I asked that you clarify the term: *"second stage of the activation phase"*; and provide citations to support your argument that passive resistance can be neglected in this phase.

In response you added the following text:
*"The forces assumed in SlideforMap are typical for the second stage of the activation phase: the displacement at which lateral root reinforcement is maximised under tension along the tension crack and at which passive earth pressure, lateral root compression and lateral soil cohesion are assumed to not be fully mobilized (Cohen and Schwarz, 2017) and we choose to neglect these."*

It is problematic to neglect the resistance on the basis that it is "not fully mobilised" because that gives no indication of its relative or absolute magnitude. It could mean (in relative terms) that the resistance in this phase is 90% of the maximum (fully mobilised) resistance; alternatively, even if the resistance were only 10% of the maximum this could still be larger than the resistance provided by roots. In fact, Cohen and Schwarz (2017) say: *"The largest force that contributes to slope stability is soil compression in the area above the landslide toe (p472)"*; and *"soil compression, due to its magnitude, dominates and controls the slope stability and its time to failure. (p472)"*; their Fig 3 suggests passive earth pressure >5 kPa for most of your second phase displacement range (0.01-0.1 m). I don't think you can use this argument to justify your choice. You might instead neglect passive resistance because it is too difficult or complex or uncertain to model but if that is the case you should say so.

It would also be helpful to explain your choice to model the second activation phase rather than first or third. Why is the second phase the most important phase to model?

**Minor Comment 2: Some statements in the paper have not been updated to reflect new results and are now incorrect.**
For example you say *"The mean angle of internal friction shows a high variation, from a very low value for StA to close to the maximum tested value for Trub" (*L498-9*)*; but with the new results, variation is much reduced, Trub is far from maximum and StA is now the second largest value. On L527 you say *"This is unique for shallow landslide probability models"*; but this is contradicted in your discussion of PRIMULA as a probability model with a treatment of basal and lateral root reinforcement (L607-13).

**Minor Comment 3: Recognising similarities between your 'no vegetation' model and SINMAP.**
I suggested that *in the discussion of similarities to other models (Section 5.4) you explain that your 'no vegetation' case differs from SINMAP only in: 1) the form of the distributions that you use to sample c and phi (log-normal for you, uniform for SINMAP); and 2) the treatment of soil depth, which is spatially uniform in SINMAP and spatially variable for you. This means that the AUC values for 'no vegetation' are indicative of those that you would expect from SINMAP.* This was not addressed in your response but I think it is important that you do add this discussion because it has implications for the trade-off between model complexity and model skill. The no veg case represents a simple (low parameter) infinite slope model very similar to SINMAP that could be widely applied. If it works almost as well as the more complex model then that is an important finding worth reporting.

A summary of each of your vegetation scenario models, and what they mean for model structure would slot nicely into Section 3.4.3 'Vegetation parameter scenarios' (L425-30). Section 5.4 would be the place to add any interpretation of the performance of different parameter scenarios in terms of their implications for the value of different model structures i.e. what is gained by representing lateral reinforcement in general then what further gains are available when you use single tree detection.

In Section 5.6 (L601) you say: *"In comparison to more simple models based on infinite slope calculations (Pack et al., 1998; Montgomery and Dietrich, 1994, SINMAP,SHALTAB), SlideforMAP considers the effect of lateral root reinforcement".* This is the perfect place to point out that SfM in the 'no veg' case becomes an infinite slope calculation that does not consider the effect of lateral root reinforcement. You could then report performance

of this version (which neglects lateral reinforcement) and comment on the performance improvement associated with representing lateral reinforcement.

My suggestions in relation to Minor Comment 3 are only suggestions though, not requirements, I'm happy for you to simply say no thankyou we don't want to do that.

**Line specific comments:**
L473: Why did "10 repetitions" of the global uniform veg produce such different mean AUC values (Table 8) to the AUC recovered during calibration (Table 7)? Global mean AUCs for each catchment (Table 8) are 4-7% lower than calibrated values (Table 7). The calibrated value is 1.8-3.9 standard deviations from the mean depending on catchment. Perhaps this is because the calibrated AUC takes the maximum from a large number of random realisations. However, in that case the calibrated AUCs (and associated ROC curves, Fig 8) are an inflated values, which are not representative and should not be reported elsewhere in the paper (e.g. L15, L636). The AUCs in Table 8 are much more representative and should be reported instead. You should consider entirely removing the AUC row from Table 7.

L475: "compared to the reference scenario used for calibration, the AUC remains almost unchanged or slightly increases for the vegetation based on single tree detection.": I have three points here: 1) this text appears to contradict later claims that single tree detection improves model performance (I think this problem will go away when you address my second point); 2) you shouldn't compare to the AUC for your reference because this is the maximum of a much larger number of trials (see previous comment) so the comparison is not a fair one; 3) Can you be more specific? This is a key result! it is worth adding a little more detail on the relative improvement for each site and on average. From a quick look, I think that in 2 of 3 cases AUC is increased by 1-4% in the other it is reduced by 4% and on average there is a 1% increase in AUC.

L549: "relative gain of 2%": It isn't clear what this comparison is referring to, which two vegetation parameterisations are being compared? How significant is the change? You can now comment on this using Table 9.

L551: "single tree detection is the best performing scenario": This is a key result. Can you give more detail here? What about StA and what about on average? How much better is it than the next best scenario? Is it significantly better than all the other scenarios or only a subset? Again this is a suggestion, not critical for publication.

**Additional reference (not cited in paper)**
Dazio, E., Conedera, M. and Schwarz, M., 2018. Impact of different chestnut coppice managements on root reinforcement and shallow landslide susceptibility. *Forest Ecology and Management*, *417*, pp.63-76.

---

## Referee Report (RR3)

**Third review of van Zadelhoff et al. for NHESS by David Milledge**

My major concern with the previous version was in the calculation of basal and lateral root reinforcement. The method of calculating each is now clear with lateral reinforcement a depth integration of reinforcement per unit area then basal reinforcement simply reinforcement per unit area at the basal depth.

There is only one minor outstanding issue here. You were unable to reproduce my depth dependent lateral and basal root reinforcement values though they had been generated using your equations and parameters. You suggested that "while creating the figure provided by the reviewer, the values of the coefficients for the calculation of the basal and lateral root reinforcement were mixed up." However, this is not the case.

Instead, I think the mismatch in our root reinforcement-depth functions is related to how we are parameterising the gamma function. I have previously implemented your equations in Excel and have now repeated the exercise using the Matlab function (gampdf(x,alpha,beta)) using exactly the parameter values in your R script. In both Excel and Matlab I get the same result but this differs from your result generated using the R function dgamma(x, shape, rate = 1, scale = 1/rate, log = FALSE). I'm not an R user so I can't check this in R, and I had to look the function up online. I found it here: https://stat.ethz.ch/R-manual/R-devel/library/stats/html/GammaDist.html

In all cases (excel, R, Matlab) there is agreement with your equation (9) both in relation to the functional form of the gammapdf and in how the parameters $\alpha$ (shape) and $\beta$ (scale) are defined. However, I strongly suspect that the syntax involved in calling the dgamma function in R is leading to the scale ($\beta$) parameter being miss-defined as the rate ($\lambda=1/\beta$) parameter in the R function. I suspect this because when I use your beta_2 value as a rate (i.e. $\lambda=3.688$) then convert to scale (i.e. $\beta =1/ \lambda = 1/3.688 = 0.2711$), I can reproduce the reinforcement curves in your response.

[Figure]

*Fig. 1: a) my excel implementation of your previous equations using $\alpha=1.284$, $\beta=3.688$; b) my Matlab implementation of your current equations using $\alpha=1.284$, $\beta=3.688$; c) your R implementation of your current equations using $\alpha=1.284$, $\beta=3.688$; d) my Matlab implementation of your current equations using $\alpha=1.284$, $\beta= 0.2711$ (i.e. 1/3.688). Note the similarity between GammaPDF in a and basal in b (which is expected given the RRmax coefficient =1) and the similarity between Rlat in a and Lateral in b (which is expected since the functional form is unchanged in the new version). Note also the similarity in both Lateral and Basal curves in c and d when the second parameter is adjusted in Matlab to account for the rate-scale mismatch.*

I'm not a frequent user of gammaPDF functions so I was concerned that I could be getting something wrong but the results I can find online seem to support my implementation and suggest that the error is in the R implementation.

[Figure]

*Fig. 2: Gamma distributions generated from an online calculator using: a) your parameters as shape and scale parameters; b) your parameters as shape and rate parameters. The online calculator is available here:*
https://homepage.divms.uiowa.edu/~mbognar/applets/gamma.html

Minor Comment 1: The need for a clearer justification for your choice to neglect passive resistance. I asked that you clarify the term: "second stage of the activation phase"; and provide citations to support your argument that passive resistance can be neglected in this phase. In response you added the following text: "The forces assumed in SlideforMap are typical for the second stage of the activation phase: the displacement at which lateral root reinforcement is maximised under tension along the tension crack and at which passive earth pressure, lateral root compression and lateral soil cohesion are assumed to not be fully mobilized (Cohen and Schwarz, 2017) and we choose to neglect these." It is problematic to neglect the resistance on the basis that it is "not fully mobilised" because that gives no indication of its relative or absolute magnitude. It could mean (in relative terms) that the resistance in this phase is 90% of the maximum (fully mobilised) resistance; alternatively, even if the resistance were only 10% of the maximum this could still be larger than the resistance provided by roots. In fact, Cohen and Schwarz (2017) say: "The largest force that contributes to slope stability is soil compression in the area above the landslide toe (p472)"; and "soil compression, due to its magnitude, dominates and controls the slope stability and its time to failure. (p472)"; their Fig 3 suggests passive earth pressure >5 kPa for most of your second phase displacement range (0.01-0.1 m). I don't think you can use this argument to justify your choice. You might instead neglect passive resistance because it is too difficult or complex or uncertain to model but if that is the case you should say so.  It would also be helpful to explain your choice to model the second activation phase rather than first or third. Why is the second phase the most important phase to model?

We indeed choose to neglect passive resistance. We argue that adding all lateral forces (assuming both activation phases simultaneously) as done in previous models is too optimistic for slope stability. Therefore, we add just one. We argue not all resisting forces are activated at once as shown by the results in Cohen and Schwarz (2017). We focus the modeling on the phase where lateral root reinforcement reaches the maximum values. Our focus on the detailed inclusion of vegetation contributes to this choice. In this phase, we argue that it is more conservative (safer) to neglect passive earth pressure forces, because they are not fully activated and the magnitude considerably changes depending on the stiffness of the landslide material and the dimension of the landslide. Further research and work on this topic is definitely required. We improved this argumentation in the paper (L154-159).

**This is a good response why not include it more completely in the paper? You make several points here very clearly that you don't really make in the paper itself. You recognise here that you are choosing to represent one particular phase of the failure process and choosing to do so partly because that is the part that interests you most as opposed to based on some argument about which stage is the limiting stage in generation of landslides that run out rapidly over long distances causing considerable damage. I would like to see you move this text into the paper in full, I think it will help readers understand your motivation and approach, but that is only a suggestion.**

Minor Comment 2: Some statements in the paper have not been updated to reflect new results and are now incorrect. For example you say "The mean angle of internal friction shows a high variation, from a very low value for StA to close to the maximum tested value for Trub" (L498-9); but with the new results, variation is much reduced, Trub is far from maximum and StA is now the second largest value. On L527 you say "This is unique for shallow landslide probability models"; but this is contradicted in your discussion of PRIMULA as a probability model with a treatment of basal and lateral root reinforcement (L607-13).

Thank you for pointing this out: we read the paper thoroughly and believe that all results now correspond to the current submission.

**This has now been addressed.**

Minor Comment 3: Recognising similarities between your 'no vegetation' model and SINMAP.  I suggested that in the discussion of similarities to other models (Section 5.4) you explain that your 'no vegetation' case differs from SINMAP only in: 1) the form of the distributions that you use to sample c and phi (log-normal for you, uniform for SINMAP); and 2) the treatment of soil depth, which is spatially uniform in SINMAP and spatially variable for you. This means that the AUC values for 'no vegetation' are indicative of those that you would expect from SINMAP. This was not addressed in your response but I think it is important that you do add this discussion because it has implications for the trade-off between model complexity and model skill. The no veg case represents a simple (low parameter)

infinite slope model very similar to SINMAP that could be widely applied. If it works almost as well as the more complex model then that is an important finding worth reporting.

We thank the reviewer for the suggestion of putting the SlideforMAP "No vegetation" scenario in direct comparison to the well-established SINMAP model. We added an overview of the complexity/model skill trade-off and put this in connection with SINMAP and SlideforMAP (L590- 593) as we also believe this would be helpful to users. We did not test SINMAP, therefore we think AUC similarities are speculative. The differences between the model results due to parametrizations from different distribution and the uniform vs. adapted soil depth can be significant. It is not the aim of this paper to compare in detail the results of other models.

**Thank you, I think you could perhaps have pushed this further, but I think it is suitable and useful as it is.**

A summary of each of your vegetation scenario models, and what they mean for model structure would slot nicely into Section 3.4.3 'Vegetation parameter scenarios' (L425-30). Section 5.4 would be the place to add any interpretation of the performance of different parameter scenarios in terms of their implications for the value of different model structures i.e. what is gained by representing lateral reinforcement in general then what further gains are available when you use single tree detection. In Section 5.6 (L601) you say: "In comparison to more simple models based on infinite slope calculations (Pack et al., 1998; Montgomery and Dietrich, 1994, SINMAP,SHALTAB), SlideforMAP considers the effect of lateral root reinforcement". This is the perfect place to point out that SfM in the 'no veg' case becomes an infinite slope calculation that does not consider the effect of lateral root reinforcement. You could then report performance of this version (which neglects lateral reinforcement) and comment on the performance improvement associated with representing lateral reinforcement.  My suggestions in relation to Minor Comment 3 are only suggestions though, not requirements, I'm happy for you to simply say no thankyou we don't want to do that. We thank the reviewer for pointing this out.

Our idea was to indicate the benefits of the inclusion of lateral root reinforcement over infinite slope calculation, but we agree that we can state this more explicitly, both in the vegetation scenarios and the discussion. We revised the lines accordingly (L531 - 542).

**Thank you, this is now clearer.**

**Line specific comments: these have all been addressed so there is no need to reproduce them here.**

---

## Author Response (AR2)

**nhess-2021-140 -Introducing SlideforMap; a probabilistic finite slope approach for modelling shallow landslide probability in forested situations - van Zadelhoff et al., 2021**

**Response to the comments of the reviewers**

In these responses, we provide the original comments (in italics) and our related responses. The corresponding changes in the manuscript are indicated in the track-change version submitted along with the revised paper at the end of this response. for readability, we removed certain old section in the review-rebuttal threads and marked these as:  (...)

*Dear Editor,  This is my second review of the manuscript proposed by Van zadelhoff et al. I went through the point by point replies and the revised paper. I can confirm that the authors have responded adequately to most of the comments/suggestions and made significant improvements to the manuscript. The two criticisms that I had raised actually remained, i.e. the ones related to the rainfall intensity as calibration parameter and the selection of only rainfall events of 1-hour duration.  Please, read below a few additional comments.*

*1. Literature review (introduction/discussion).*
*The author added this sentence in the introduction, under my request of discussing the hydrological effects of vegetation. L95-98: "The hydrological effect influences effective soil moisture by interception, increased evapotranspiration and increased infiltration (Greenway, 1987; Masi et al., 2021). Over the short timescale with intense rainfall these hydrological effects are negligible (Feng et al., 2020)."*
*Actually, I found it as an extrapolation of an opposite statement suggested in my first review!*

*MY PREVIOUS COMMENT: "With regard to the effect of vegetation, the aspects related to the hydrological effects should be at least discussed, which can sometime be even more significant than the mechanical ones (Feng et al., 2020)".*

*MY NEW COMMENT: I absolutely agree that during the single event the hydrological effect of vegetation are negligible, but in a long-term analysis, which is required to assess the initial condition prior the event, this is not true. Please smooth it.*

The authors agree and we smoothed the text (L97) emphasizing the importance of knowledge on initial conditions. In the methodology we stated the shortcoming of SlideforMap that is does not include prior knowledge on the initial conditions (L255).

*2. Hydrology and precipitation.*
*MY PREVIOUS COMMENT: "In the description of the model application (section 3.4.2) it is not clear how rainfall initiating events are selected. If I understood well, only events of 1hour duration are selected, whose intensity is identified from the Depth-Duration-Frequency (DDF) curve at different return periods (i.e. from 10 to 100 years). Therefore, I guess 10 events of 1 hours are simulated. Is that correct? If so, it should be explained and justified the reason of analyzing events of only 1 hours, which cannot be 'critical' for landslide initiation. Authors should deeply clarify this part in the manuscript, explain the methodology used to define the events, and report the parameters of the DDF curves".  –*

*AUTHOR REPLY: We emphasized the choice for 1-hour events (assumed macro-pore activation time period) in the methodology L256 and the parametrization of the DDF curves in the supplementary material.*

*MY NEW COMMENT: I think that this is still a strong limitation. I would suggest to report the values of the rainfall intensities to have an idea. Just report a summarizing table in the paper. I think that rather than the R.code (supplementary material, that I appreciated though) it would be more useful to have the resulting rainfall intensities.*

Thank you for pointing this out. we added a table with the rainfall intensity to corresponding return periods per study area in the methodology (Table 5). A 1 hour and 24-hour period were chosen to account for both a short and long duration event. The maximum and minimum rainfall depth of the Table is subsequently used in the sensitivity analysis.

*3. Calibration/sensitivity analysis*
*MY PREVIOUS COMMENT: "With regard to the best set of parameters, my question is: are the found parameters consistent and realistic? For example, I argue the choice of including the precipitation intensity as calibration parameter. As discussed in the previous comment, rainfall represents the triggering forcing and it is a dynamic variable. Ideally, we should know the precipitation intensity associated to each observed landslide. Otherwise, if used as parameter, it seems that the model is tuned ad hoc just to reproduce the past events. If so, which could be its utility?"*

*AUTHOR REPLY: We agree that in an ideal case this should be known, however no detailed information is available. Therefore, we have to rely on the more simplified steady state approach. We added the lack of detailed information in the data section to justify our approach L335-336.*

*MY NEW COMMENT: The criticism still remains. Then, it should be clear that the calibrated model (with these found parameter values) cannot be used in a forecast usage mode, i.e. with rainfall series as a forecasted input.*

We updated the introduction to strongly emphasized that SlideforMap should not be used as a forecast tool (127 to 129).

*Review of van Zadelhoff et al. 2021 for NHESS by David Milledge I made a community comment on the first version of this manuscript and was then asked by the Associate*
*Editor to review the revised paper. The revisions have improved an already very interesting paper, which demonstrates the applicability and predictive capability of a new model for shallow landslides to provide a detailed inclusion of the influence of vegetation. I have responded in bold to the authors' responses below. I think my additional comments or queries are only fairly minor and should require only fairly minor alterations to the paper. The only exception to this the updated Equations 8 and 9, where I think there is an inconsistency in the way basal and lateral root cohesion are being calculated.*

*(...) The comparison to other models in Section 5.6 is very useful. However, I think there are a few other models that capable of catchment scale application and that resolve lateral root reinforcement which are worth discussing. Montgomery et al., (2000) is particularly important to discuss in Section 5.6 because they use your hydrological model and a very similar geotechnical model (infinite slope with roots the only lateral reinforcement). Other important models to consider are: Hess et al. (2017), who also assume fixed dimensions for the landslides but include a treatment for boundary friction and passive resistance (which you neglect); Cislaghi et al (2017, 2018) sample landslide sizes from a distribution (similar to your approach); von Ruette et al. (2013) and Bellugi et al. (2015) predict size in the model rather than imposing a single size or distribution of sizes. I suspect that both of the latter are fairly slow to run relative to your approach and you certainly improve on the approaches that define a single landslide size. You should perhaps comment on the ways in which your approach differs from that of Cislaghi et al. (2017, 2018). It would be useful to say a little more about the respects in which SfM "uses a more realistic implementation of root reinforcement" and how this differs from these other models. If the main difference is that you neglect compressive resistance this may need stronger justification either here or in the methods section.*

*Do any other studies predict landslide locations accounting for the spatial distribution of root reinforcement as a function of forest structure? We can argue about the specifics of the stability model but this for me is probably the most novel aspect of the work.*

We added a comparison to the forest clearing model by Montgomery et al., (2000) in the discussion (L583-587). Differences to SlideforMap in this model are quite profound in their fixed values for soil depth and landslide dimensions, which are variables from distributions or in SlideforMap. Their fixed value for root reinforcement also is a different approach from our spatial distribution.

As suggested by the author we added a comparison to PRIMULA by Cislaghi et al. (2018) in the discussion (L588-594). We agree this is important as they use a similar application as SlideforMap, the main difference is our computation of root reinforcement on a shallow landslide scale

In comparing to Hess et al., (2017), the difference is that to our knowledge, adding all forces (both passive and active) at the same time overestimates the slope stability. We make our argument in the methodology (L153-155) and added a comparison in the discussion (L594-596)

We see a similarity in computation of slope stability by von Ruette et al. (2013) and Bellugi et al. (2015), but the approach is different from ours by focusing on a single cell, where we focus on the whole landslide surface area. We think this is not appropriate to compare here, as they are more similar to SOSlope (Cohen & Schwarz, 2017).

*I also had a couple of other minor comments on the new text:*

*L113-4: "displacement independent": I think your approach also models displacement-independent reinforcement. If you make the point that this is a limitation of previous approaches up here, you should probably add in your comparison to other stability models that SfM uses displacement independent reinforcement whereas Cohen and Schwarz (2017) and to some degree von Ruette et al. (2013) account for displacement in their treatments.*

We worded this poorly. Our model applies the maximum root reinforcement under tension, which is in itself a displacement independent value, but based on displacement-tension based pullout tests. We worded this more concise (L114) and do indeed think this experimental root reinforcement is an improvement upon a single assumed value. We asserted this in our objectives (L 123).

*L155: "second stage of the activation phase" this isn't explicitly referred to in the introduction so it is not clear what you are pointing back to here. The sentence that follows ("This coincides with…") is very helpful and it might be easier to understand this paragraph if you simply said something like: "…typical for the second stage of the activation phase: the displacement at which lateral root reinforcement is maximised under tension along the tension crack and at which passive earth pressure, lateral root compression and lateral soil cohesion no longer act."*

Thank you, we took the reviewer suggestion and defined the second activation phase (maximum tension related forces, neglected compression related forces) right after introducing the term (L 153-157).

*(…) We expected an improvement in the performance of the model with the detailed inclusion of vegetation, which we analyzed in the paper. This sentence is great! It captures your hypothesis for the paper really nicely and would be worth including around L124 because I don't see such a clear statement of your expectation in the current paper.  (…) This additional text is useful.*

Thank you for the suggestion. We included the hypothesis of performance improvement by including vegetation this in our objectives (L 123).

*(...) I think it would be useful in the discussion of similarities to other models (Section 5.4) to explain that your 'no vegetation' case differs from SINMAP only in: 1) the form of the distributions that you use to sample c and phi (log-normal for you, uniform for SINMAP); and 2) the treatment of soil depth, which is spatially uniform in SINMAP and spatially variable for you. This means that the AUC values for 'no vegetation' are indicative of those that you would expect from SINMAP. I would have really liked to see a comparison to SHALSTAB in terms of AUC because it is so simple and so widely used. But it certainly isn't essential that the authors choose to do that.*

We agree that comparison of AUC values in our study areas generated by both SHALSTAB and SINMAP to SlideforMap would be interesting but the presented material is already extensive".

*(...) In the literature. I have one final suggestion in relation to Table 7, since you know mean, standard deviation and sample size you should be able to estimate the significance of differences in AUCs between different approaches. A quick attempt at this for the overall performances using a t-test (which may not be the best approach) suggests that single tree detection is significantly better than no veg and Forest uniform veg at 99% and 90% confidence respectively; but is not significantly different from Global uniform veg. The Global uniform veg is significantly better than the no veg case but the Forest uniform veg is not, nor is there any significant difference between Global and Forest uniform veg cases.*

We agree and added a the generalized (Welch) t-test (Table 9) on the values as computed in Table 8 identifying the reliability of the difference in the mean, finding a significant difference in the improvement of the AUC by single tree detection in 2 of the 3 study areas. We added the significance of this result to the discussion as well.

*(...) It would be helpful to put a range of values to the 'minor movement' that you have in mind (I guess this is on the order of 10-100 mm?). If I understand L153-54 correctly you argue that beyond this displacement there is no passive earth pressure, lateral root compression or lateral soil cohesion. I think each of these claims needs a citation. How do you reconcile that with your earlier findings that passive resistance remains important even for displacements > 300 mm (Schwarz et al., 2015; Cohen and Schwarz, 2017)? Alternatively, if your reason for neglecting some of these components is that we don't yet understand them well enough to represent them satisfactorily then I think it is fine to simply say that here.*

The maximum passive earth pressure is activated in a different moment than the tensile forces in the tension crack, though we agree there still will be a certain compressive force under maximum tensile force. This depends strongly on the stiffness of the soil, which we do not yet fully understand (as formulated well by the reviewer). Our understanding in the initiation phase so far of shallow landslides Askarinejad (2012) and partly from Cohen and Schwarz (2017); Figure 7 with a numerical approach. Our negligence of the passive forces is a short

As suggested by the reviewer we added a specific range for minor movement from Schwarz et al., 2015, giving a 10-100 mm displacement (L 157).

*(...) I still don't understand why lateral soil cohesion would go to zero at some displacement but basal cohesion would be unaffected, unless you are assuming that a tension crack opens up along the entire upper half of the landslide. In that case, it might be worth adding a comment that explains that you neglect all resistance in the compression zone and assume that a tension crack opens along the entire length of the tension zone such that lateral resistance in this zone is only due to root reinforcement.*

We assume as the reviewer describes that a tension crack opens up along the entire upper half of the landslide circumference indeed leaving root reinforcement as the sole lateral resistance. The behavior of

initial movement and cracking in the upper part of the landslide is well described in Askarinejad (2012), Figure 14.  As with the previous comment, we described this more explicitly (L 153-157).

*(...) I think adding a sentence that explains that: you neglect all resistance in the compression zone and assume that a tension crack opens along the entire length of the tension zone such that lateral resistance in this zone is only due to root reinforcement. I don't necessarily agree that this is the best approach, I would do it differently but this isn't my model.*

As with previous comments. We read this in displacement measurements of shallow landslides Askarinejad (2012) which indicates that the upper part start moving first and independently from the bottom part.  A similar behavior is numerically modelled in Cohen and Schwarz (2017); Figure 7, which shows that the maximum compression force is after the tension crash has opened up. But of course, we can agree to disagree.

*(...) OUR PREVIOUS REPLY: We integrated the RBM root probability density distribution and added this as a correction factor to the lateral root reinforcement. This is described in the methodology section 5, equation 8 and the results are recalculated.*
*This sounds great. However, I'm still a little confused. Perhaps out of ignorance around Gamma distributions. I have two questions: 1) is this an analytical integration? If so can you write it out in Equation 8? 2) if the parameters should be selected to ensure that the function decreases with increasing Hsoil does that require alpha=1, in which case the function becomes an exponential and everything gets simpler? 3) is the second gamma density function in equation 8 the same as the gamma density function in equation 9? The notation in the equations and the definition in the text suggests not but I haven't understood the distinction between the two. Looking at equation 8 again, I think there may still be an error in the way that you relate basal and lateral reinforcement. The best way I can illustrate that is to set the alpha parameter to unity in the gamma distribution so that it reduces to an exponential distribution. In that case I can rewrite equation 8 as:*

$$R_l = c\, \Gamma\left(\frac{D_{trees}}{DBH\, D_{treesmax}} | \alpha_1, \beta_1\right) \int_0^H \Gamma(h | \alpha_2, \beta_2)\, dh$$

**Applying alpha=1 and substituting a constant C0 for the first term (since this is depth invariant) gives:**

$$R_l = C_0 \int_0^H \Gamma(h | 1, \beta_2)\, dh$$

**In this case the gamma density function simplifies to an exponential and the equation can be re-written as:**

$$R_l = C_0\, \beta_2 \int_0^H e^{-h\beta_2}\, dh = C_0\, \beta_2 \left(\frac{1 - e^{-H\beta_2}}{\beta_2}\right) = C_0 \left(1 - e^{-H\beta_2}\right)$$

**If root density follows this (exponential distribution) then it is clear that basal cohesion (assuming isotropic rooting and strength) is:**

$$R_b = C_0\, \beta_2\, e^{-H\beta_2}$$

**In this case the dimensions also work out, without the need for the dimensional correction coefficient k (though k takes a value of unity and can be ignored in calculations). However, applying equation 10 in this case results in:**

$$R_b = k\, R_l\, \Gamma(H | 1, \beta_2) = R_l\, \beta_2\, e^{-H\beta_2} = C_0\, \beta_2 \left(1 - e^{-H\beta_2}\right) e^{-H\beta_2}$$

*This extra term has a very large influence at small depths, pulling Rb down to zero at the surface, which is not compatible with a gamma root density distribution with alpha =1 (i.e an exponential root density distribution). The differences between the two formulations differ by <20% when depth exceeds 0.5 m and by <5% when depth exceeds 1 m (using beta=3.2 as you do in the paper). Therefore this error (if you agree that it is an error) is unlikely to result in large changes to the stability in the areas of deeper soil where landslides typically occur but should make other parts of the catchment where soils are shallow considerably more stable.*

We thank the reviewer for the detailed mathematical analysis. We believe we did not accurately state that we use the Gamma probability density function rather than the gamma function as in the rewritten equation by the reviewer. We adjusted the paper by writing the Gamma probability density distribution explicitly. Additionaly we have the following responses to the numbered questions posed by the reviewer:

1. In our code we apply numerical (approximation) integration instead of analytical integration. We assume a step size of 0.01 meters makes the approximation suitable for practical application.
2. Upon considering, this is wrong, our apologies. The function can take non-monotone decreasing shapes as well (for example: Moos et al., 2016; Figure 3)
3. The equation in essence is the same in both equation 8 and 9. Equation 8 however, applies the cumulative value (soil depth in range [0,x]), equation 9 applies the point value soil depth at [x].

*Calculating root reinforcement using spatially averaged distance to trees within the Gamma function. Previous applications of the Gamma function (Eqn 9) appear to use it to predict root reinforcement at a known distance from the nearest tree (Moos et al., 2016). Given its nonlinearities, is it reasonable to use an average distance in Eqn 9 rather than evaluating Eqn 9 for the distribution of distances then averaging? OUR PREVIOUS REPLY: For the current paper, we decided to keep this methodology as it is. I think that is reasonable, but I think you should say that explicitly in the paper and explain why you make that choice (e.g. because it is prohibitively expensive to evaluate 9 for the distribution of distances).*

Thank you for the suggestion. We explicitly stated this choice, attributing it to the tendency of root systems to not overlap (L225-226).

*Variability*
*(...) The only remaining point here is that I don't see where you define the symbols you use to represent these parameters in Table 6. It would also be worth explaining that the parameters of the log-normal are mean and standard deviation of log transformed data. I guess you then transform them back for table 6?*

Thank you. Indeed, the values from 6 are log-transformed when applied in the code of SlideforMap. The properties (mean and standard deviation) of the log-normal are the same as that of the originally used normal distribution. For clarity and openness on the method, we added the code transforming a mean and standard deviation into parameters of the log-normal distribution in the supplementary material.

*(...) I have just a few remaining minor suggestions / queries:L197: "definitive values for soil thickness": I'm not sure what you mean by definitive in this context.L198: It would help to clarify what sets the "initial thickness" if you added "which is sampled from a log-normal distribution".L202: Why do the coefficients used to find mu1 and sigma1 from mu_h and sigma_h take these particular values (1.35 and 0.75 respectively)? Did you establish them by trial and error based on your perceptual model for how soil depth varies with slope in your study areas?*

L.198. We agree by adding "sampled from a log-normal distribution and dedicated a more specific section to the soil thickness (L202-205)
L.202. The coefficients of 1.35 and 0.75 were from manual fitting of our available Slope - soil depth data. Upon reconsideration we perform a numerical fit to compute the coefficients. This results in values of 1.47 and 0.5 respectively. We recomputed our results, adjusted the text accordingly (L348-351) and added a plot in the appendix.

*Hydrology*
*(...)*
*1)    L239: TOPOG should not be cited as Montgomery and Dietrich (1994), because they didn't develop it, but should instead cite O'loughlin (1986).*

*2)  L241: Pack et al. (1998) is the incorrect citation for SHALSTAB, this should be Montgomery and Dietrich (1994).*
*3)  I agree though that Pack et al. (1998) should be cited here because the identical model is used in SHALSTAB and SINMAP and the latter is Pack et al.'s model.*
*4)  L478: 'application of the TOPOG/TOPMODEL approach'. I suggest you cut reference to TOPMODEL here because you aren't using a TOPMODEL approach.*

1. Thank you, changed it (L248)
2. changed it (L250)
3. We added SINMAP reference (L250-251)
4. Changed it (L491), thank you

*(...) L255: "Based on the literature data discussed in the introduction": I think the literature you refer to needs citing again here, ideally with a sentence that explains the basis for your claim that the time to equilibrium is 1 hour. Li et al. (2013) aren't much help here because they are talking about the time to equilibrium for vertical infiltration only, there is no lateral flow in their model as I understand it. Montgomery et al. (2002, 2004) and Iverson (2000) are counter examples because while they disagree on many things, they agree that the equilibrium time even in the conductive Coos Bay soil should be much longer than 1 hour. As I said above, I don't think that makes the model inappropriate because (lateral flow sets the antecedent conditions for the triggering burst). But it does mean that R/T should be treated as an index for the propensity of landsliding rather than a quantity that can be directly compared to observed rainfall.If you disagree, and instead think hillslope lateral response times in your study areas are fast enough to allow direct connection to observed rainfall then a sentence or two recognising the debate around these response times would probably be sufficient to make readers aware that you differ with others on this point.*

Upon consideration, we agree with the reviewer. We let go of the 1-hour assumption and indeed assume the P/T ratio as a measure for the propensity of shallow landslide activity (L498-499). In line with other reviewer comments, we added a table (Table 5) of rainfall depth corresponding to different return periods, to analyze both the conditions of a short/high intensity and long/lower intensity event. The full range of corresponding rainfall intensities is used in the qualitative sensitivity analysis.

*Queries on equations:*
*(...) see my earlier comment about a potential problem with Equation 9. If you took my alternative approach then the dimensional mismatch would disappear because Eqn 8 would be integrated over depth while Eqn 9 would be calculated directly from the depth decay function (whether gamma or exponential).*

This is in line with previous comments on the equation. We thank the reviewer for the critical look but propose the keep the equation as is.

*3) (...) Adjusting the units in Eqn 14 fixes the problem there but leads to ambiguity earlier in the paper where you define is one outstanding problem here in that you define DBH earlier in the paper (L214) then use it as a term in equation 8 is DBH also being measured in cm here or is a conversion required? I think it would be better to adjust Eqn 14 so that DBH is expressed in metres there in order to avoid confusion elsewhere.*

Thank you for the suggestion. We updated equation 14 and units throughout the paper making DBH consistently in meters.

*Literature*

Askarinejad, A., Casini, F., Bischof, P., Beck, A., & Springman, S. M. (2012). Rainfall induced instabilities: a field experiment on a silty sand slope in northern Switzerland. Rivista Italiana Di Geotecnica, (3), 50–71. Retrieved from http://www.associazionegeotecnica.it/rig/archivio

Bellugi, D., Milledge, D.G., Dietrich, W.E., Perron, J.T. and McKean, J., 2015. Predicting shallow landslide size and location across a natural landscape: Application of a spectral clustering search algorithm. Journal of Geophysical Research: Earth Surface, 120(12), pp.2552-2585.

Cislaghi, A., Chiaradia, E.A. and Bischetti, G.B., 2017. Including root reinforcement variability in a probabilistic 3D stability model. Earth Surface Processes and Landforms, 42(12), pp.1789-1806.

Cislaghi, A., Rigon, E., Lenzi, M.A. and Bischetti, G.B., 2018. A probabilistic multidimensional approach to quantify large wood recruitment from hillslopes in mountainous-forested catchments. Geomorphology, 306, pp.108-127.

Hess, D.M., Leshchinsky, B.A., Bunn, M., Mason, H.B. and Olsen, M.J., 2017. A simplified three-dimensional shallow landslide susceptibility framework considering topography and seismicity. Landslides, 14(5), pp.1677- 1697.

Montgomery, D.R., Schmidt, K.M., Greenberg, H.M. and Dietrich, W.E., 2000. Forest clearing and regional landsliding. Geology, 28(4), pp.311-314.
von Ruette, J., Lehmann, P. and Or, D., 2013. Rainfall-triggered shallow landslides at catchment scale: Threshold mechanics-based modeling for abruptness and localization. Water Resources Research, 49(10), pp.6266-6285.

Moos, C., Bebi, P., Graf, F., Mattli, J., Rickli, C., & Schwarz, M. (2016). How does forest structure affect root reinforcement and susceptibility to shallow landslides? Earth Surface Processes and Landforms, 41(7), 951–960. https://doi.org/10.1002/esp.3887

Schwarz, M., Rist, A., Cohen, D., Giadrossich, F., Egorov, P., Büttner, D., … Thormann, J. J. (2015). Root reinforcement of soils under compression. Journal of Geophysical Research F: Earth Surface, 120(10), 2103–2120. https://doi.org/10.1002/2015JF003632

---

## Author Response (AR3)

**nhess-2021-140 -Introducing SlideforMap; a probabilistic finite slope approach for modelling shallow landslide probability in forested situations - van Zadelhoff et al., 2021**

We thank the reviewer for taking the time to rereview our paper. Below, our responses are in black. Red and green are the comments of the reviewer. Results were recomputed to create the AUC curves in Figure 10. Consequently,  due to the probabilistic nature of the model, the results change but the conclusions of the paper do not. We took the comments to our computation of root reinforcement seriously and reformulated how we compute the lateral root reinforcement. We hope this is understandable now.

**Reviewer 1; GENERAL COMMENTS**

 1. On L50, I do not quite agree with the authors' assertion that "statistical" and "probabilistic" models are distinguished by whether physical processes are included. It is entirely possible to have a probabilistic model that does not take physical processes into account.
This is indeed indisputable. We revised the text (L 50).
2. On L119, the authors state that SlideForMAP is intended for application on a "scale of 1-1000 square meters." What exactly does this mean (study area, raster cell size, etc.)? This description is confusing in relation to L188, where the authors indicate that the maximum landslide surface area is 3000 square meters.
This is the scale of application of SlideforMAP in square kilometers. This paragraph aims to explain the reader on what study areas it can be applied and to quantify our definition of 'regional scale'. We revised the text to make this clearer (L 119-120).
 3. On L177-178, the authors are referring to the landslide density, which indicates the number of landslides per unit area. However, it is unclear if the authors mean that there should be more than 1 center of mass per raster cell, or whether it would be sufficient for each cell to be overlapped by more than one HL, which could occur with a landslide density significantly less than 1 center of mass per cell if the landslides were sufficiently large.
Thank you for the detailed reflection. Not every raster cell should have a random landslide center of mass, overlap suffices. Due to this the landslide density can be less than one 1 per raster cell.
 4. In Figure 6, it appears that the number of subsamples retaining the full parameter subset varies. Shouldn't these all have a count of 100?
The total is 1000 samples from a uniform distribution using Latin Hypercube sampling divided over 10 bins. 100 samples per bin is the mean, but minor inaccuracies occur due to sampling.
*TECHNICAL CORRECTIONS*
L45. Change "spatial" to "spatially".
Thank you, we corrected line (L45).

There is some inconsistency throughout the paper in the use of "SlideForMAP" versus "SlideForMap". Please ensure that the name of the model is consistent throughout.

Thank you for pointing out this inconsistency. After decision making with the team, we settled on 'SlideforMAP'. We made this consistent throughout the paper.

There is some inconsistency throughout the paper in referring to parameters using mathematical notation and fonts (for example, "a and b" in L 209). Please ensure that mathematical notation is consistent throughout.

Thank you, we made this consistent throughout the paper.

**Review of van Zadelhoff et al. for NHESS by David Milledge**

*Thank you to the authors for taking the time to respond to my comments, almost all of which have been addressed. However, I have one major outstanding concern. I think that the equations for basal and lateral reinforcement are still incorrect. It is essential that the authors address this prior to publication. I also have three, fairly minor concerns that can be very easily addressed:*

1. The need for a clearer justification for the choice to neglect passive resistance;
2. Some statements in the paper have not been updated to reflect new results and are now incorrect;
3. Discussion of similarities between your 'no vegetation' parameterisation of your model and SINMAP.

*My other concerns from the previous review have been addressed. Each of the remaining concerns has some history in previous reviews I have tried to summarise my concern and the history from previous reviews below.*

Major Comment: Representing basal root reinforcement as a function of lateral reinforcement is incorrect

It is now clear that you use a Gamma pdf and the additional equation is useful, you clarified in your response that you used numerical integration but should update the text to reflect this. However, my main original concern has not been addressed. Eqn10 calculates basal reinforcement as a function of lateral reinforcement. That is incorrect conceptually because if lateral reinforcement is a depth integration of reinforcement per unit area then basal reinforcement should simply be reinforcement per unit area at the basal depth. Your response to my previous comment suggests that you agree (you said "The equation in essence is the same in both equation 8 and 9. Equation 8 however, applies the cumulative value (soil depth in range [0,x]), equation 9 applies the point value soil depth at [x]"). However, your implementation is at odds with this. Eqn 4 in Gehring et al. (2019) is much more consistent with both your response and my expectation because they use of Rmax rather than Rlat. Is it possible there is an error in Eqn 10 in the paper and it should be using Rmax?

We updated the text to explicitly state that we use numerical integration (L245).

Our previous response was indeed not clear. We revised the original equations and now present lateral root reinforcement as a function of maximum lateral root reinforcement (N/m) and vertical root distribution (i.e. maximum lateral root reinforcement is reduced by the assumed root distribution over the soil depth), i.e. it is not reinforcement per unit area but per unit soil depth. The new equations now reads at:

$$RR_{\text{max}} = (c \cdot DBH) \cdot \Gamma_{PDF}\left(\frac{D_{\text{trees}}}{DBH \cdot 18.5}\bigg|\alpha_1, \beta_1\right),$$

The maximal root reinforcement RRmax (N/m) is a function of tree distance $D_{\text{trees}}$ [m] and tree diameter DBH [m]. RRmax refers to the maximum (lateral) force of a root bundle under tension. The factor of a1 and b1 are both dimensionless, rendering the Gamma PDF dimensionless. c has the unit N/m$^2$.

$$R_{\text{lat}} = RR_{\text{max}} \cdot \int_0^{H_{\text{soil}}} \Gamma_{\text{PDF}}\left(H\bigg|\alpha_2, \beta_2\right) dH,$$

The lateral root reinforcement Rlat (N/m) depends on soil depth H (m) and is formulated as an integral of a gamma density function. The factor of a2 is dimensionless and b2 is in [m]. This definition renders the integral dimensionless.

Equation 8 in our paper was identical to Equation 3 in Gehring et al., 2019, except for an addition of a depth correction factor. This was poorly formulated in the paper. We reformulated this by adding RRmax explicitly and redefining the depth correction in a separate subsequent equation (Equation 8 and 10 in the paper). The equation in the previous manuscript version reads as:

$$R_{\text{lat}} = c \cdot \Gamma_{PDF}\left(\frac{D_{\text{trees}}}{DBH \cdot D_{\text{trees,max}}}\bigg|\alpha_1, \beta_1\right) \cdot \int_0^{H_{\text{soil}}} \Gamma_{PDF}(\alpha_2, \beta_2)\, dh$$

The basal reinforcement is defined as the maximum lateral root reinforcement multiplied with the root distribution value at maximum soil depth (with unit 1/m), which results in unit N/m$^2$ or Pa.
In addition to the conceptual problem, your approach introduces two errors:
1.   The dimensional error that you have to correct with the unit coefficient k.
We believe this not to be an error, but to be coherent with an assumption of isotropic root reinforcement. The same assumption is made in Gehring et al., 2019, though not explicitly stated. We reformulated the calculation of lateral root reinforcement, starting with the calculation of RRmax.

2. A relationship between basal reinforcement and depth that is at odds with observations of root density and reinforcement. Your group's previous observations (e.g. Schwarz et al., 2012) and those of other groups all suggest that reinforcement declines with depth, or at least does not increase. Your equations ensure that basal root reinforcement increases with depth. The graph below was generated using your gammaPDF parameters with an Rmax ($Rmax = c\ \Gamma\ (\ Dtrees\ |\alpha 1,\ \beta 1)$) of 1 kN/m2 (to keep things $DBH\ Dtreesmax$ simple). I'm convinced that this representation is incorrect and that it is potentially important because in shallow soils, where basal reinforcement is particularly important, it will result in a very large underestimation of reinforcement.

[Figure]

My second concern having plotted the graph above is that the parameters of your gammaPDF implemented with Gehring et al.'s (2019) parameters implies that root density initially increases rapidly with depth while your own observations of the vertical distribution of roots in Switzerland and many observations from other groups all suggest that root biomass (and thus reinforcement) decays with depth, usually exponentially (see Dazio et al., 2018; Schwarz et al., 2012 and references therein: Abe and Iwamoto, 1990, Abe and Ziemer, 1991, Abernethy and Rutherfurd, 2001, Schmidt et al., 2001, Schenk and Jackson, 2002, Bischetti et al., 2005, Laio et al., 2006, Docker and Hubble, 2009). This suggests that your alpha2 and beta2 parameters are not physically reasonable.

Do you agree that gammaPDF values should be lower at 2 m depth than they are at 0.2 m depth?

We agree with the reviewer that basal root reinforcement should decrease with depth. In addition, we expect that the Rlat should be approximating the value of RRmax much sooner than displayed in the figure provided by the reviewer.

Our hypothesis is, that while creating the figure provided by the reviewer, the values of the coefficients for the calculation of the basal and lateral root reinforcement were mixed up. Alpha 1 and Beta 1 are the coefficients for the Gamma Probability Density function (PDF) of root density in the horizontal direction. The latter is used to compute RRmax.

Alpha 2 and Beta 2 (Table 1) are the Gamma PDF coefficients for the root reinforcement distribution in the vertical direction. That is used to compute lateral root reinforcement and basal root reinforcement from RRmax.

Below we produced figures on the basal and lateral root reinforcement. The upper right figure is our result of the lateral and basal root reinforcement with soil depth, akin to the figure as given by

the reviewer. In our calculations the results are consistent with the examples in the literature, cited by the reviewer. We produced a concise script (in R), provided in the supplementary material, that produces these figures.

[Figure]

upper left: Maximum root reinforcement (RRmax) as a function of horizontal distance from stem.
upper right: Lateral and basal root reinforcement for an assumed RRmax of 1 kN/m as a function of soil thickness. Under this assumption The Gamma PDF is identical to the basal root reinforcement.
lower left: Resulting Lateral and Basal Root force as a function of soil depth. This assumes RRmax of 1 kN/m and a Shallow landslide of 50 m2 surface area.
A minor point, Table 1 would be improved by replacing 'literature' entries with the relevant references.
We agree. We revised Table 1.

Minor Comment 1: The need for a clearer justification for your choice to neglect passive resistance.
I asked that you clarify the term: "second stage of the activation phase"; and provide citations to support your argument that passive resistance can be neglected in this phase.
In response you added the following text:
"The forces assumed in SlideforMap are typical for the second stage of the activation phase: the displacement at which lateral root reinforcement is maximised under tension along the tension crack and at which passive earth pressure, lateral root compression and lateral soil cohesion are assumed to not be fully mobilized (Cohen and Schwarz, 2017) and we choose to neglect these."
It is problematic to neglect the resistance on the basis that it is "not fully mobilised" because that gives no indication of its relative or absolute magnitude. It could mean (in relative terms) that the resistance in this phase is 90% of the maximum (fully mobilised) resistance; alternatively, even if the resistance were only 10% of the maximum this could still be larger than the resistance provided by roots. In fact, Cohen and Schwarz (2017) say: "The largest force that contributes to slope stability is soil compression in the area above the landslide toe (p472)"; and "soil compression, due to its magnitude, dominates and controls the slope stability and its time to failure. (p472)"; their Fig 3 suggests passive earth pressure >5 kPa for most of your second phase displacement range (0.01-0.1 m). I don't think you can use this argument to justify your choice. You might instead neglect passive resistance because it is too difficult or complex or uncertain to model but if that is the case you should say so.  It would also be helpful to explain your choice to model the second activation phase rather than first or third. Why is the second phase the most important phase to model?

We indeed choose to neglect passive resistance. We argue that adding all lateral forces (assuming both activation phases simultaneously) as done in previous models is too optimistic for slope stability. Therefore, we add just one. We argue not all resisting forces are activated at once as shown by the results in Cohen and Schwarz (2017). We focus the modeling on the phase where lateral root reinforcement reaches the maximum values. Our focus on the detailed inclusion of vegetation contributes to this choice. In this phase, we argue that it is more conservative (safer) to neglect passive earth pressure forces, because they are not fully activated and the magnitude considerably changes depending on the stiffness of the landslide material and the dimension of the landslide. Further research and work on this topic is definitely required. We improved this argumentation in the paper (L154-159).

Minor Comment 2: Some statements in the paper have not been updated to reflect new results and are now incorrect. For example you say "The mean angle of internal friction shows a high variation, from a very low value for StA to close to the maximum tested value for Trub" (L498-9); but with the new results, variation is much reduced, Trub is far from maximum and StA is now the second largest value. On L527 you say "This is unique for shallow landslide probability models"; but this is contradicted in your discussion of PRIMULA as a probability model with a treatment of basal and lateral root reinforcement (L607-13).

Thank you for pointing this out: we read the paper thoroughly and believe that all results now correspond to the current submission.

Minor Comment 3: Recognising similarities between your 'no vegetation' model and SINMAP. I suggested that in the discussion of similarities to other models (Section 5.4) you explain that your 'no vegetation' case differs from SINMAP only in: 1) the form of the distributions that you use to sample c and phi (log-normal for you, uniform for SINMAP); and 2) the treatment of soil depth, which is spatially uniform in SINMAP and spatially variable for you. This means that the AUC values for 'no vegetation' are indicative of those that you would expect from SINMAP. This was not addressed in your response but I think it is important that you do add this discussion because it has implications for the trade-off between model complexity and model skill. The no veg case represents a simple (low parameter) infinite slope model very similar to SINMAP that could be widely applied. If it works almost as well as the more complex model then that is an important finding worth reporting.

We thank the reviewer for the suggestion of putting the SlideforMAP "No vegetation" scenario in direct comparison to the well-established SINMAP model. We added an overview of the complexity/model skill trade-off and put this in connection with SINMAP and SlideforMAP (L590-593) as we also believe this would be helpful to users. We did not test SINMAP, therefore we think AUC similarities are speculative. The differences between the model results due to parametrizations from different distribution and the uniform vs. adapted soil depth can be significant. It is not the aim of this paper to compare in detail the results of other models.

A summary of each of your vegetation scenario models, and what they mean for model structure would slot nicely into Section 3.4.3 'Vegetation parameter scenarios' (L425-30). Section 5.4 would be the place to add any interpretation of the performance of different parameter scenarios in terms of their implications for the value of different model structures i.e. what is gained by representing lateral reinforcement in general then what further gains are available when you use single tree detection. In Section 5.6 (L601) you say: "In comparison to more simple models based on infinite slope calculations (Pack et al., 1998; Montgomery and Dietrich, 1994, SINMAP,SHALTAB), SlideforMAP considers the effect of lateral root reinforcement". This is the perfect place to point out that SfM in the 'no veg' case becomes an infinite slope calculation that does not consider the effect of lateral root reinforcement. You could then report performance of this version (which neglects lateral reinforcement) and comment on the performance improvement associated with representing lateral reinforcement. My suggestions in relation to Minor Comment 3 are only suggestions though, not requirements, I'm happy for you to simply say no thankyou we don't want to do that.

We thank the reviewer for pointing this out. Our idea was to indicate the benefits of the inclusion of lateral root reinforcement over infinite slope calculation, but we agree that we can state this more explicitly, both in the vegetation scenarios and the discussion. We revised the lines accordingly (L531 - 542).

*Line specific comments:*

L473: Why did "10 repetitions" of the global uniform veg produce such different mean AUC values (Table 8) to the AUC recovered during calibration (Table 7)? Global mean AUCs for each catchment (Table 8) are 4-7% lower than calibrated values (Table 7). The calibrated value is 1.8-3.9 standard deviations from the mean depending on catchment. Perhaps this is because the calibrated AUC takes the maximum from a large number of random realisations. However, in that case the calibrated AUCs (and associated ROC curves, Fig 8) are an inflated values, which are not representative and should not be reported elsewhere in the paper (e.g. L15, L636). The AUCs in Table 8 are much more representative and should be reported instead. You should consider entirely removing the AUC row from Table 7.

You are right that the AUCs in table 7 are inflated values. They are the maximum of the large number of random realisations and in that sense are 'lucky guesses'. We agree and removed any mention of these values outside of Table 7. We updated the ROC curves to correspond to the 10 repetitions as presented in Table 8.

L475: "compared to the reference scenario used for calibration, the AUC remains almost unchanged or slightly increases for the vegetation based on single tree detection.": I have three points here: 1) this text appears to contradict later claims that single tree detection improves model performance (I think this problem will go away when you address my second point); 2) you shouldn't compare to the AUC for your reference because this is the maximum of a much larger number of trials (see previous comment) so the comparison is not a fair one; 3) Can you be more specific? This is a key result! it is worth adding a little more detail on the relative improvement for each site and on average. From a quick look, I think that in 2 of 3 cases AUC is increased by 1- 4% in the other it is reduced by 4% and on average there is a 1% increase in AUC.

Thank you for pointing this out as well as for the suggestions for improvement:

1. we removed the sentence and formulated these results in more detail in line with other comments by the reviewer.
2. In line with previous comments we revised this part and used the results of Table 8 as central to our results and discussion.
3. We added the relative change in AUC of the single tree detection compared to the other vegetation scenarios (L536-537). This combined with Table 9 with the reliability of the difference is presented as a key result (L543-551).

L549: "relative gain of 2%": It isn't clear what this comparison is referring to, which two vegetation parameterisations are being compared? How significant is the change? You can now comment on this using Table 9.

We improved the description and included a reference to Table 9. We argue that though a seemingly small improvement in AUC, the concept of diminishing returns, makes this a valuable improvement. (L537, L547-548)

L551: "single tree detection is the best performing scenario": This is a key result. Can you give more detail here? What about StA and what about on average? How much better is it than the next best scenario? Is it significantly better than all the other scenarios or only a subset? Again this is a suggestion, not critical for publication.

Thank you for the suggestion. We added more details to the description of the key result, relating to the study area and chosen scenario (L531- 543).

Additional reference (not cited in paper)
*Dazio, E., Conedera, M. and Schwarz, M., 2018. Impact of different chestnut coppice managements on root reinforcement and shallow landslide susceptibility. Forest Ecology and Management, 417, pp.63-76.*

---

## Author Response (AR4)

**nhess-2021-140 -Introducing SlideforMap; a probabilistic finite slope approach for modelling shallow landslide probability in forested situations - van Zadelhoff et al., 2021**

We thank the reviewer for his patience and giving our paper another review. Below, our responses to the points that required addressing are in black. Green are the comments of the reviewer.

Third review of van Zadelhoff et al. for NHESS by David Milledge
My major concern with the previous version was in the calculation of basal and lateral root reinforcement. The method of calculating each is now clear with lateral reinforcement a depth integration of reinforcement per unit area then basal reinforcement simply reinforcement per unit area at the basal depth.
There is only one minor outstanding issue here
You were unable to reproduce my depth dependent lateral and basal root reinforcement values though they had been generated using your equations and parameters. You suggested that "while creating the figure provided by the reviewer, the values of the coefficients for the calculation of the basal and lateral root reinforcement were mixed up." However, this is not the case. Instead, I think the mismatch in our root reinforcement-depth functions is related to how we are parameterising the gamma function. I have previously implemented your equations in Excel and have now repeated the exercise using the Matlab function (gampdf(x,alpha,beta)) using exactly the parameter values in your R script. In both Excel and Matlab I get the same result but this differs from your result generated using the R function dgamma(x, shape, rate = 1, scale = 1/rate, log = FALSE). I'm not an R user so I can't check this in R, and I had to look the function up online. I found it here: https://stat.ethz.ch/R-manual/R-devel/library/stats/html/GammaDist.html

In all cases (excel, R, Matlab) there is agreement with your equation (9) both in relation to the functional form of the gammapdf and in how the parameters α (shape) and β (scale) are defined. However, I strongly suspect that the syntax involved in calling the dgamma function in R is leading to the scale (β) parameter being miss-defined as the rate ($\lambda = 1/\beta$) parameter in the R function. I suspect this because when I use your beta_2 value as a rate (i.e. $\lambda = 3.688$) then convert to scale (i.e. $\beta = 1/\lambda = 1/3.688 = 0.2711$), I can reproduce the reinforcement curves in your response.

[Figure]

*Fig. 1: a) my excel implementation of your previous equations using α=1.284, β=3.688; b) my Matlab implementation of your current equations using α=1.284, β=3.688; c) your R implementation of your current equations using α=1.284, β=3.688; d) my Matlab implementation of your current equations using α=1.284, β= 0.2711 (i.e. 1/3.688). Note the similarity between GammaPDF in a and basal in b (which is expected given the RRmax coefficient =1) and the similarity between Rlat in a and Lateral in b (which is expected since the functional form is unchanged in the new version). Note also the similarity in both Lateral and Basal curves in c and d when the second parameter is adjusted in Matlab to account for the rate-scale mismatch.*

I'm not a frequent user of gammaPDF functions so I was concerned that I could be getting something wrong but the results I can find online seem to support my implementation and suggest that the error is in the R implementation.

[Figure]

Fig. 2: Gamma distributions generated from an online calculator using: a) your parameters as shape and scale parameters; b) your parameters as shape and rate parameters. The online calculator is available here: https://homepage.divms.uiowa.edu/~mbognar/applets/gamma.html

We agree with the conclusions by the reviewer. We incorrectly stated the parameter value as being the scale where in reality it is the rate parameter. We thank the reviewer for being thorough in identifying this. We updated references to the scale to rate throughout the paper. To be thorough we also updated the formulation of the Gamma distribution in equation 9 (L238). This way the formulation is similar to the dgamma function in R, the application in SlideforMAP and the website referenced by the reviewer (https://homepage.divms.uiowa.edu/~mbognar/applets/gamma.html)

Minor Comment 1:
The need for a clearer justification for your choice to neglect passive resistance. I asked that you clarify the term: "second stage of the activation phase"; and provide citations to support your argument that passive resistance can be neglected in this phase. In response you added the following text: "The forces assumed in SlideforMap are typical for the second stage of the activation phase: the displacement at which lateral root reinforcement is maximised under tension along the tension crack and at which passive earth pressure, lateral root compression and lateral soil cohesion are assumed to not be fully mobilized (Cohen and Schwarz, 2017) and we choose to neglect these." It is problematic to neglect the resistance on the basis that it is "not fully mobilised" because that gives no indication of its relative or absolute magnitude. It could mean (in

relative terms) that the resistance in this phase is 90% of the maximum (fully mobilised) resistance; alternatively, even if the resistance were only 10% of the maximum this could still be larger than the resistance provided by roots. In fact, Cohen and Schwarz (2017) say: "The largest force that contributes to slope stability is soil compression in the area above the landslide toe (p472)"; and "soil compression, due to its magnitude, dominates and controls the slope stability and its time to failure. (p472)"; their Fig 3 suggests passive earth pressure >5 kPa for most of your second phase displacement range (0.01-0.1 m). I don't think you can use this argument to justify your choice. You might instead neglect passive resistance because it is too difficult or complex or uncertain to model but if that is the case you should say so. It would also be helpful to explain your choice to model the second activation phase rather than first or third. Why is the second phase the most important phase to model?

We indeed choose to neglect passive resistance. We argue that adding all lateral forces (assuming both activation phases simultaneously) as done in previous models is too optimistic for slope stability. Therefore, we add just one. We argue not all resisting forces are activated at once as shown by the results in Cohen and Schwarz (2017). We focus the modeling on the phase where lateral root reinforcement reaches the maximum values. Our focus on the detailed inclusion of vegetation contributes to this choice. In this phase, we argue that it is more conservative (safer) to neglect passive earth pressure forces, because they are not fully activated and the magnitude considerably changes depending on the stiffness of the landslide material and the dimension of the landslide. Further research and work on this topic is definitely required. We improved this argumentation in the paper (L154-159).

This is a good response why not include it more completely in the paper? You make several points here very clearly that you don't really make in the paper itself. You recognise here that you are choosing to represent one particular phase of the failure process and choosing to do so partly because that is the part that interests you most as opposed to based on some argument about which stage is the limiting stage in generation of landslides that run out rapidly over long distances causing considerable damage. I would like to see you move this text into the paper in full, I think it will help readers understand your motivation and approach, but that is only a suggestion.

Re-reading the manuscript we found that all the arguments are expressed in lines (L154-159). We made some minor improvements to the text, we think that now it is more clear than before.